# DIP-MS: ultra-deep interaction proteomics for the deconvolution of protein complexes

Fabian Frommelt [1,8] ✉, Andrea Fossati[1,2,3,4,8], Federico Uliana[1,5], Fabian Wendt [6], Peng Xue[1,7], Moritz Heusel[1], Bernd Wollscheid [6], Ruedi Aebersold [1], Rodolfo Ciuffa[1] & Matthias Gstaiger [1] ✉

Most proteins are organized in macromolecular assemblies, which represent key functional units regulating and catalyzing most cellular processes. Affinity purification of the protein of interest combined with liquid chromatography coupled to tandem mass spectrometry (AP–MS) represents the method of choice to identify interacting proteins. The composition of complex isoforms concurrently present in the AP sample can, however, not be resolved from a single AP–MS experiment but requires computational inference from multiple time- and resource-intensive reciprocal AP–MS experiments. Here we introduce deep interactome profiling by mass spectrometry (DIP-MS), which combines AP with blue-native-PAGE separation, data-independent acquisition with mass spectrometry and deep-learning-based signal processing to resolve complex isoforms sharing the same bait protein in a single experiment. We applied DIP-MS to probe the organization of the human prefoldin family of complexes, resolving distinct prefoldin holo- and subcomplex variants, complex–complex interactions and complex isoforms with new subunits that were experimentally validated. Our results demonstrate that DIP-MS can reveal proteome modularity at unprecedented depth and resolution.

Understanding how different proteins are spatially organized into functional modules catalyzing and controlling numerous biochemical and cellular processes underlying distinct phenotypes is one of the main goals of molecular systems biology. Protein complexes, defined here as stable assemblies that can be isolated by biochemical means, are key regulators of cellular functions. Far from being invariant assemblies, protein complexes are contextual and have been shown to adapt to the cellular type or state by changing their subunit composition, stoichiometry, localization and abundance of expression[1–3]. AP–MS[4,5] has been the method of choice for the analysis of protein complexes.

However, AP–MS of a single bait identifies direct as well as indirect interactors that may not belong to the same complex but rather be part of different complexes concurrently present in the AP sample. Therefore, protein–protein interaction (PPI) data from several reciprocal AP–MS experiments are needed to deconvolve MS data into distinct molecular entities[6].

As an alternative to AP–MS protein correlation methods exemplified by size-exclusion chromatography (SEC–MS)[7,8] and blue-native-PAGE (BNP)[9] coupled to MS[10] have been introduced. They fractionate native complexes by their hydrodynamic radius and

[1]Department of Biology, Institute of Molecular Systems Biology, ETH Zurich, Zurich, Switzerland. [2]Quantitative Biosciences Institute (QBI), University of California San Francisco, San Francisco, CA, USA. [3]Department of Cellular and Molecular Pharmacology, University of California San Francisco, San Francisco, CA, USA. [4]J. David Gladstone Institutes, San Francisco, CA, USA. [5]Department of Biology, Institute of Biochemistry, ETH Zurich, Zurich, Switzerland. [6]Department of Health Sciences and Technology (D-HEST), Institute of Translational Medicine (ITM), ETH Zurich, Zurich, Switzerland. [7]Guangzhou National Laboratory, Guang Zhou, China. [8]These authors contributed equally: Fabian Frommelt, Andrea Fossati. ✉e-mail: fabian.frommelt@hotmail.com; matthias.gstaiger@imsb.biol.ethz.ch

size, respectively, and each fraction is subsequently profiled by MS. Resulting cofractionation profiles are then used to define protein complexes. While these approaches can identify concurrent protein complexes involving overlapping proteins, the information is limited by the sensitivity of the analytical instrument, sample loading capacity and resolution of the SEC columns[11,12]. Collectively, these factors limit the general utility of cofractionation-MS methods for detecting (1) complexes present at low abundance, (2) complex components present in substoichiometric amounts and (3) the resolution of different complex instances containing the same core subunits but different accessory proteins.

Here we introduce deep interactome profiling by mass spectrometry (DIP-MS), which combines the capacity of affinity purification (AP) to enrich the interactome of a target protein with the ability of native BNP fractionation-MS to resolve different complexes sharing the same target protein. In addition to introducing a high-throughput protocol, we developed PPIprophet, a data-driven neural network-based protein–complex deconvolution system.

Compared to the few previous studies that combined AP with fractionation-MS[13–17], DIP-MS provides three critical improvements: (1) a miniaturized sample preparation procedure in a filter plate format that requires ten times less material than traditional chromatography-based separation[18,19] and achieves high reproducibility; (2) a fast data-independent acquisition with mass spectrometry (DIA–MS) scheme with an increased throughput of up to 60 samples per day and (3) a deep-learning framework trained on more than 1.5 million binary interactions from 32 cofractionation datasets, which enables prediction of PPIs, identification of multiple instances of protein complexes and robust deconvolution of complex profiling data into functional modules.

To explore the potential of DIP-MS for large-scale PPI profiling, we analyzed the interactome of human prefoldin proteins. Prefoldins play a central role in cellular proteostasis via stabilizing nascent proteins in interplay with other chaperones[20–22]. They are best known as part of the evolutionarily conserved heterohexameric canonical prefoldin (PFD) complex, a cytosolic roughly 120 kDa ATP-independent chaperone comprising two different roughly 23 kDa α-subunits and four different roughly 15 kDa β-subunits[20,23]. In addition to the prototypical PFD complex, complexes containing prefoldin subunits have been implicated in a range of cellular processes including neurodegeneration[24–26], degradation of misfolded proteins[27] and were detected in different cellular compartments[28,29]. Further, prefoldin and prefoldin-like proteins form the prefoldin-like module (PFDL)[30], and both complexes can assemble in supercomplexes, such as the chaperonin CCT/TRiC-PFD[31] and most prominently the PAQosome, a HSP90 chaperone complex, which has multiple biological chaperone functions, including assisting the assembly and maturation of large RNA-binding protein assemblies[32–36], stabilization of multiple phosphatidylinositol 3-kinase-related kinase complexes[34,35] and interaction with the TSC complex[33,37] (Supplementary Table 1 and Extended Data Fig. 1).

To gain further insights into the landscape of prefoldin complexes we performed DIP-MS using PFDN2 and UXT as bait proteins. Analysis with PPIprophet and comparison of the results with those obtained by AP–MS and size exclusion chromatography coupled to sequential window acquisition of all theoretical fragment ion spectra mass spectrometry (SEC-SWATH) identified most known prefoldin complexes in a single experiment and identified 319 PFD–PFDL-specific interactors. DIP-MS not only recapitulated the composition of all reported PFD complexes but also quantified their stoichiometry and suggested the existence of stable subassemblies and supercomplexes. Further it revealed a previously unknown PFD homolog and deconvolved the PFD and PFDL- complex landscape into multiple complex instances. In summary, we introduce DIP-MS as a method to quantitatively study the organization of the proteome at unprecedented resolution and sensitivity.

## Results

### Overview of the DIP-MS method
The steps of the DIP-MS experimental workflow are illustrated in Fig. 1a. The protein complexes containing the bait protein were affinity purified and subjected to BNP to separate complexes by their apparent molecular weight. The gel was cut into roughly 70 slices of 1 mm width, which were then individually processed using a fast and reproducible filter aided in-gel digestion preparation protocol in a 96-well plate format. Finally, proteolyzed peptides from each fraction were measured with quantitative DIA–MS[38] coupled with a short liquid chromatography gradient[39]. The resulting comigration (from here on coelution) matrices of peptide fragment ion spectra (Fig. 1b) were processed via PPIprophet to infer quantitative protein electrophoretic elution patterns, assembly states and PPIs. Protein profiles showing multiple peaks were further deconvolved to infer multiple assemblies, thereby identifying in a single DIP-MS experiment multiple complexes containing the bait protein.

### A deep-learning framework for PPI prediction and complex inference
The PPIprophet software was specifically developed to extract the following information from DIP-MS data: (1) PPIs, (2) identification of bait-protein complexes, (3) subunit stoichiometry and approximate molecular weight of separated complexes and (4) prey–prey interactions typically invisible to AP-MS.

In developing PPIprophet, we trained a deep neural network (DNN) model (for details, see Methods) for PPI prediction using more than 1.5 million PPIs extracted from databases containing data from different types of cofractionation measurement[40]. By using STRING and BioPlex as ground truth, this DNN model achieved outstanding performance with a receiver-operating characteristic of 0.995 on our independent test set of 335,071 PPIs (Supplementary Table 2 and Methods), showing the flexibility of deep learning for this task compared to previously reported correlation-based approaches[13]. We benchmarked PPIprophet against other cofraction tools (PCprophet, EPIC, PrInCE) and demonstrated its superior performance for the analysis of DIP-MS datasets (Supplementary Results and Supplementary Fig. 1a–c).

To reduce the false discovery rate (FDR) due to spuriously comigrating proteins, PPIprophet performs FDR control using data-generated decoy PPIs. By exhaustively mapping and predicting all PPIs represented by the data, the software tool generates a weighted network (Fig. 1b) used for further protein–complex identification. To distinguish complex components from contaminants, we devised an interaction metric (W score, adapted from CompPASS[41]) that uses specificity and selectivity to filter copurifying proteins, by performing in silico APs. Finally, PPIprophet can be used either in a hypothesis testing mode where the PPI network is deconvolved into complexes by superimposition of available complex knowledge, or in an entirely data-driven mode using MCL clustering[42].

### Benchmarking of DIP-MS against AP–MS and SEC–MS workflows
To evaluate the performance of our methods we compared DIP-MS results using PFDN2 as a bait with data generated by two orthogonal methods: (1) SEC–MS[18] and (2) the reciprocal AP–MS dataset using 11 PFD–PFDL subunits as baits (Supplementary Table 3). The DIP-MS dataset identified 353 interaction partners in total, 187 more than both SEC–MS and AP–MS, recovering roughly 30% of the interactors in public databases (Fig. 2a, for a reference list see Supplementary Table 4). For a comparison of cofractionation data versus DIP-MS, see the Supplementary Information Results section and Supplementary Fig. 2a,b).

Higher benchmark coverage did not substantially affect error rates, as we observed greater recall and similar precision compared to our in-house generated AP–MS data using the manually curated set

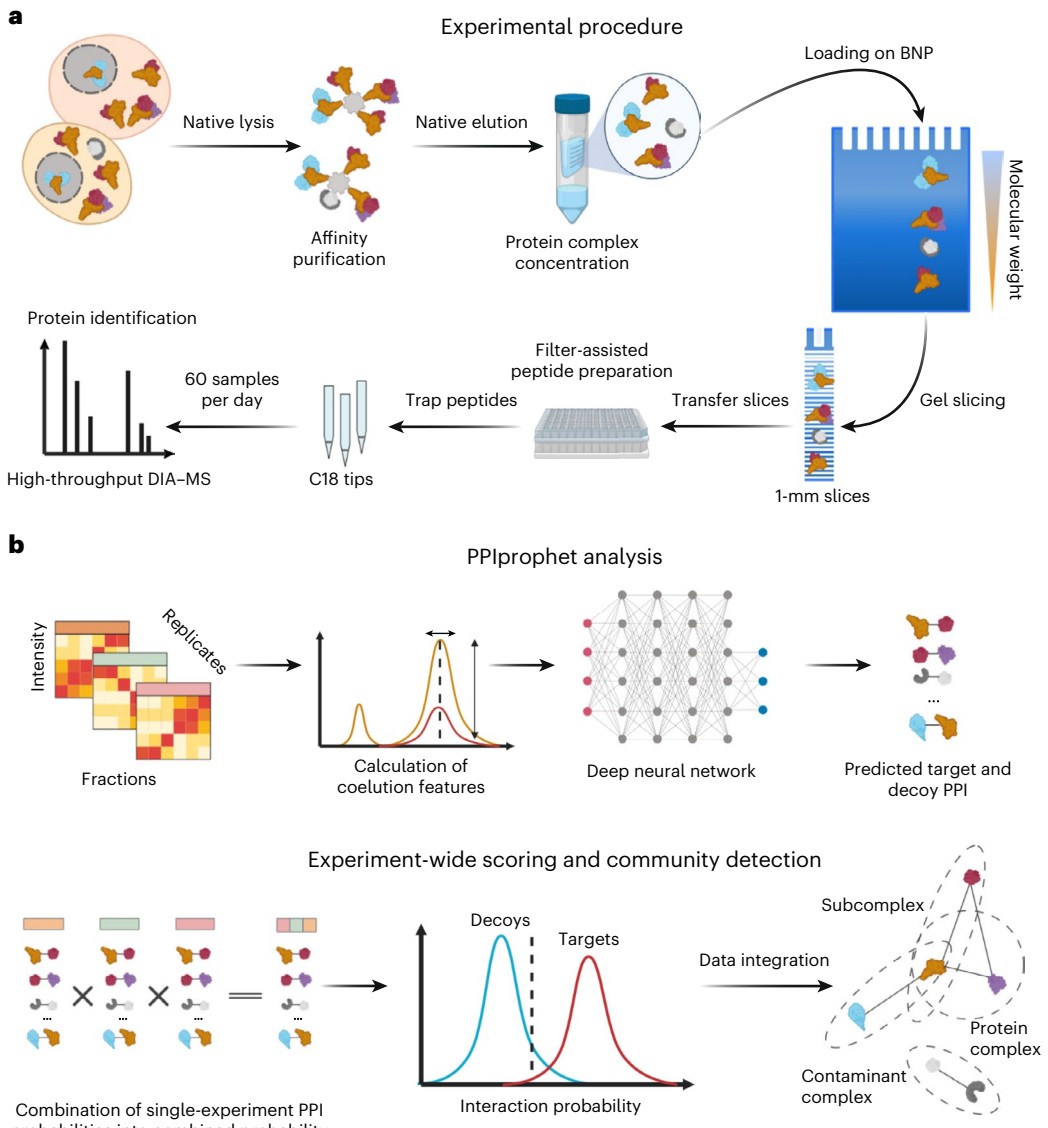

**Fig. 1 | Schematic of experimental and computational DIP-MS workflow. a**, Experimental workflow, including sample preparation, AP, gel-based fractionation and DIA. **b**, Computational framework encompassing, first, generation of all possible PPIs in the data and then prediction of PPI using deep learning. Interaction probabilities are then combined and further filtered using a target–decoy competition approach. Complexes are derived from the resulting interaction network by integration with available databases or by using a data-driven approach.

as ground truth, suggesting that DIP-MS generates much larger and denser interaction networks at no cost of precision (Fig. 2b).

We hypothesized that the high recall rate by DIP-MS is caused by the initial bait-enrichment step. To test this, we compared the MS2 signal intensities of the benchmark proteins as surrogates of protein abundance in the DIP-MS and SEC–MS datasets (Fig. 2c). We found that the signal intensities in the SEC–MS data covered a range of roughly 3.8 logs whereas DIP-MS data covered a dynamic range of roughly 4.4 logs, suggesting increased coverage of low-abundance proteins from the target set. It is important to note that the SEC–MS experiment was measured in a different MS platform compared to our DIP-MS so lower or higher absolute abundance should not be considered as proxy of coverage, while the proportion of signal (that is, the $y$ axis) in the empirical cumulative distribution function plot is magnitude-agnostic allowing comparison between different instruments.

Because the enrichment step lowers the detection threshold for low-abundant proteins and, at the same time, notably reduces the complexity of the sample in each gel fraction compared to lysates

analyzed in SEC–MS, we hypothesized that DIP-MS should also resolve more complexes than SEC–MS. To test this, we applied the same signal processing and peak-picking algorithm to all benchmark proteins identified by both methods ($n = 59$) and indeed, identified more peaks using DIP-MS (1 versus 2, $P = 0.00187$) (Fig. 2d).

Next, we compared topology and connectivity of networks generated by DIP-MS, SEC–MS or AP–MS. To determine which method most closely recapitulated the network topology of a large-scale PPI database, we calculated the graph edit distance (GED) between the subnetworks encompassing all the 475 PFD–PFDL proteins from the target list identified in all three experiments (intersection of quantified proteins across the DIP-MS, SEC–MS and reciprocal AP–MS) and two representative PPI databases (STRING and BioPlex). These 475 proteins represent only the proteins identified, not necessarily predicted as positive interaction partners, hence offer an unbiased metric of algorithm performance in network reconstruction. GED is a measure of topological similarity between networks, taking the value of one for identical graphs. Lower values indicate diverging networks

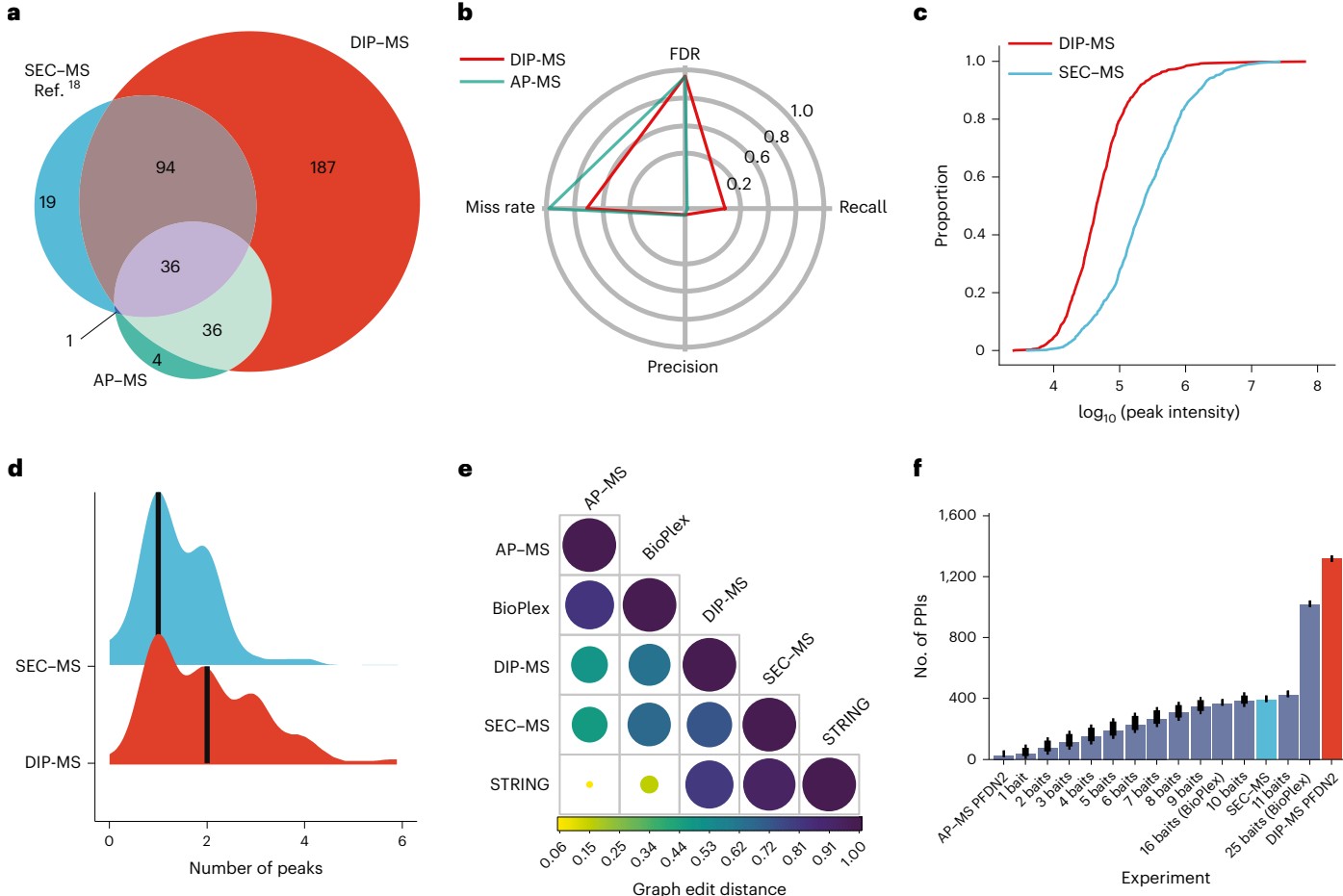

**Fig. 2 | Benchmark of DIP-MS versus other techniques for interactome analysis. a**, Venn diagram showing the overlap in identified PFDN2 interactors in SEC–MS, AP–MS (across 11 core subunits of PFD and PAQosome) and DIP–MS from the curated list of known PFDN2 interactors. **b**, Radarplot showing the performance of DIP-MS over AP-MS. **c**, Empirical cumulative distribution function plot for PFD–PFDL associated proteins identified in the PFDN2 DIP-MS experiment (average across replicates) and the SEC-MS from ref. 18. The $x$ axis represents the $\log_{10}$ MS2 abundance integrated over a protein coelution profile in the SEC-MS dimension, defined here as a protein peak. The $y$ axis represents the proportion of protein peaks at a specific abundance. **d**, Number of coelution peaks per PFD–PFDL associated proteins, identified in both, SEC-MS and DIP-MS.

The solid black line represents the median for SEC-MS (1) and for DIP-MS (2) while the $y$ axis represents the kernel density for the SEC-MS (blue density) and the DIP-MS (red density). Protein peak was defined as a signal having minimal width of three fractions and minimal height above background of 0.2 on a 0–1 signal scale. **e**, GED matrix between different techniques and databases. Color code represents the GED similarity from highly dissimilar (yellow, small circles) to isomorph graphs (dark purple, large circles). **f**, Barplot showing the number interactions recovered by different methods. Error bars represent standard deviation with $n$ being the total number of combinations possible from the correspondent bait number. Different techniques are highlighted by different bar colors. Data are presented as mean values ± s.d.

(Methods for details). As expected, we observed high GED for datasets generated by the same technique. Specifically, the AP–MS derived network was closer to the large-scale AP–MS dataset BioPlex than SEC–MS and DIP-MS. The last networks were more similar to STRING (0.94 and 0.84, respectively), whereas the AP–MS derived network was vastly different from STRING (GED score of 0.078) as shown in Fig. 2e. This indicates that DIP-MS recapitulates the topology of the graph similar to SEC–MS but, critically, does not rely on previous knowledge and therefore allows discovery of new unreported complexes. Finally, we compared the number of PPIs, as direct proxy for the density of the network, generated by DIP-MS, SEC–MS and AP–MS. A single-bait DIP-MS experiment (PFDN2) was sufficient to generate a network with 1,306 PPIs. A subset of the SEC–MS data containing the same proteins identified in DIP-MS from the target list led to a less well-connected network of 386 PPIs. To compare DIP-MS and AP–MS data derived networks, we asked how many AP–MS experiments would be required to reconstruct a network as dense as the one generated by one DIP-MS measurement. To this end, we queried the BioPlex interaction network in human embryonic kidney 393 (HEK293) cells and selected all interactions encompassing either

16 baits (11 PFDN–PFDL-core subunits and five proteins from the R2TP module) or 25 baits (11 PFD–PFDL-core, five from the R2TP module and nine from CCT/TRiC). We found that even using all 25 baits from BioPlex yielded approximately 20% fewer PPIs than the PPIs retrieved by a single DIP-MS experiment (1,011 versus 1,306) (Fig. 2f). In addition, when compared with orthogonal data from published in vivo proximity-dependent biotin identification experiments we found besides method specific interactions 78 PPIs also identified by DIP-MS (Supplementary Fig. 3a,b).

In summary, our benchmarking data indicate that DIP-MS data, when compared to SEC–MS or reciprocal AP–MS data, have a broader dynamic range, capture the separation behavior of proteins at higher resolution, generate more extensive and denser networks and recapitulate a larger portion of the ground truth.

### Global organization of prefoldin and prefoldin-like complexes
We next generated a detailed map of prefoldin and prefoldin-like complexes applying DIP-MS with UXT found in PFDL and PAQosome complexes and PFDN2 a prefoldin subunit common to all known prefoldin assemblies[30,43].

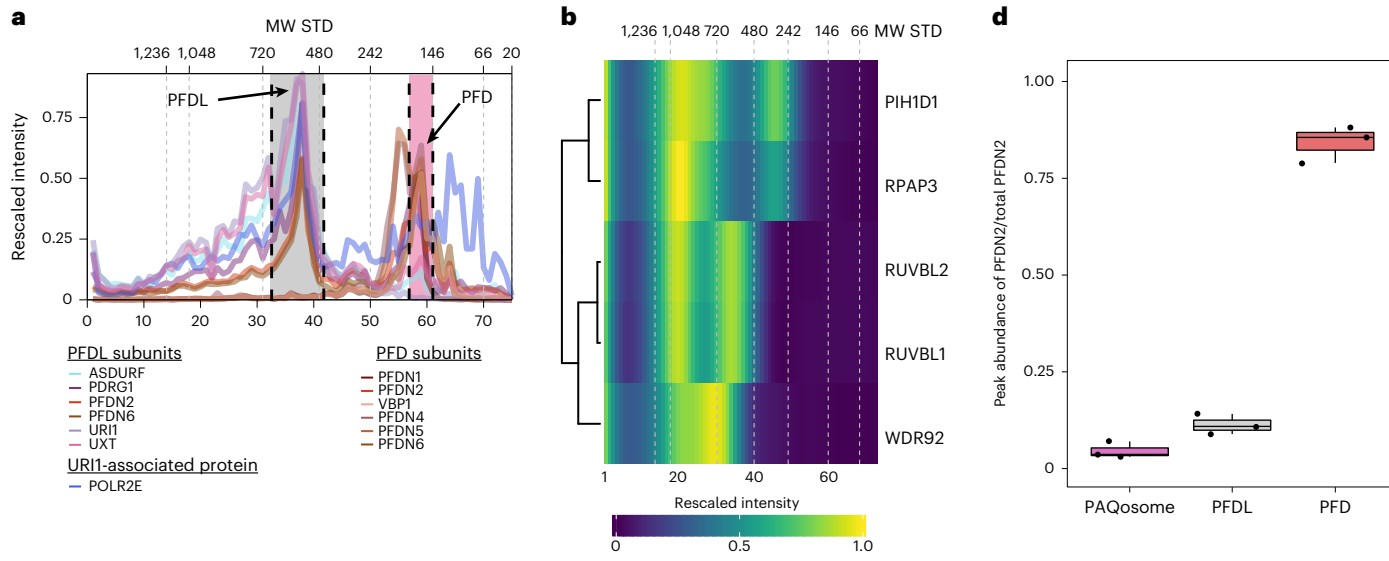

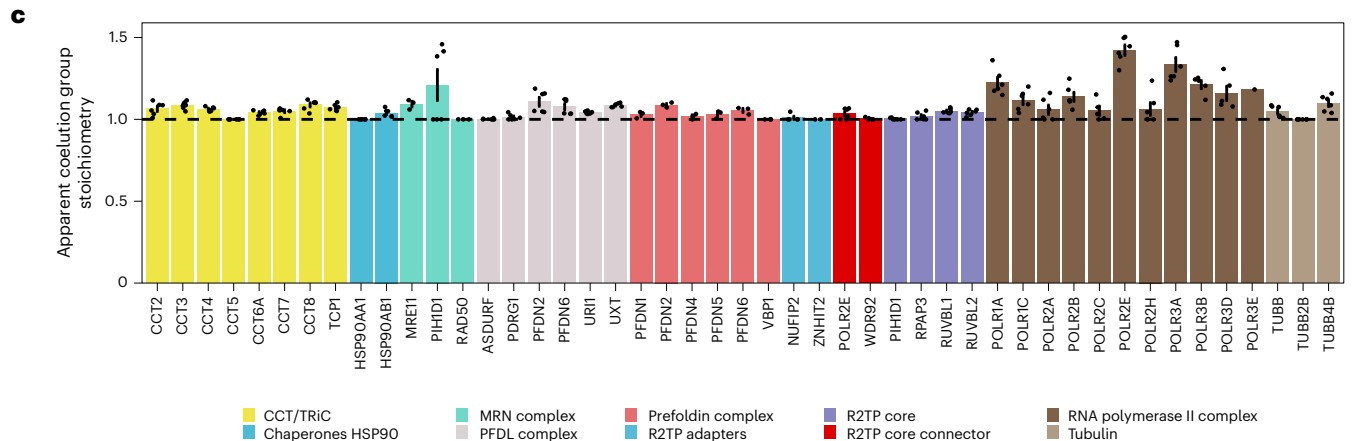

**Fig. 3 | Identification of assemblies from PFDN2 and UXT DIP-MS experiments. a**, Coelution profiles PFD and PFDL complexes in PFDN2 and UXT DIP-MS experiments. The *x* axis represents the gel slice number, while the *y* axis shows the rescaled protein intensities (sum of TOP2 peptides, MS2 based protein intensity, summed up for each protein across all DIP-MS experiments before scaling). Region between the black dotted lines delimits the peak group selected as assembled PFD and PFDL. Molecular weights (MW) of the standards (STD) are reported as an additional *x* axis on top. **b**, Heatmap of the PAQosome subunits Ruvbl1/2, RPAP3, WDR92 and PIH1D1 after Ward's distance-based clustering for the UXT DIP-MS purification. Cell values represent the rescaled intensity averaged across UXT DIP-MS triplicates (*n* = 3 biologically independent replicates). **c**, Barplot showing the stoichiometry obtained from the DIP-MS experiments of UXT and PFDN2 for representative complexes in

the PFD and PFDL interaction network as defined in Supplementary Table 5. Stoichiometry was defined as the ratio between the protein having the lowest full-width at half-maximum peak area in a complex and all the other proteins in the same peak group. The dotted line highlights a stoichiometry of 1.00, which correspond to the literature-reported value for PFDN, PFDL as well as CCT/TRiC. Error bars indicate the standard deviation across six DIP-MS experiments (PFDN2, *n* = 3 biologically independent experiments; and UXT, *n* = 3 biologically independent experiments). Due to missing values, a few proteins have fewer replicates and/or points. Data are presented as mean values ± s.d. **d**, A boxplot showing the ratio of PFDN2 (bait) signal in the PFD, PFDL or PAQosome complex over the total PFDN2 signal. The box shows the interquartile range (IQR) and the whiskers 1.5× IQR. The black line represents the median while each dot shows one independent replicate (*n* = 3).

Across the two DIP-MS experiments, we profiled 1,513 proteins following initial data processing using CCprofiler[18] for sibling peptide correlation-based filtering and conversion of peptide-level features into protein-level features (Supplementary Data 1). Using PPIprophet we detected 6,762 (PFDN2) and 5,682 (UXT) high-confidence (combined probability across replicates greater than or equal to 0.9) binary interactions, resulting in a network containing a total of 11,552 unique interactions and 939 proteins (Supplementary Data 2).

In our combined DIP-MS derived network, we identified all previously reported PFD–PFDL assemblies (Extended Data Fig. 2) including PFD, PFDL and the PFDL containing PAQosome[20,30]. Further we identified coelution groups composed of the R2TP proteins RUVBL1/2, the adapters/regulatory subunits RPAP3 and PIH1D1 and all subunits of the CCT/TRiC-PFD complex.

PFDL subunits interacting with POLR2E, (Fig. 3a) were base-peak resolved and could be readily identified by naïve clustering procedures (Supplementary Fig. 4a). Besides their fractionation in two lower molecular weight peaks, PFD–PFDL subunits also migrated at a high molecular weight indicating that PFD and PFDL partake in two distinct supercomplexes: the PFD-CCT/TRiC supercomplex (Extended Data Fig. 2 coelution group 6), and the PAQosome (Extended Data Fig. 2 coelution group 7) that comprises the fully assembled R2TP and PDFL. Our data captured the reported variant of RUVBL1/2 as a hetero-dodecameric complex (Protein Data Bank (PDB) ID 2XSZ)[44–46] lacking the adapter and/or regulatory proteins RPAP3 and PIH1D1 (Fig. 3b), which are part of the R2TP core. It was proposed that RUVBLs cycle between a double and single ring form, which may give rise to the specialized chaperone function of the R2TP core[47]. For the two

adapter proteins PIH1D1 and RPAP3 (ref. 48), we identified three separate peaks, which suggests the presence of multimeric subassemblies formed by RPAP3 and PIH1D1. We also identified the two R2TP subunits at the PAQosome coelution group. Indeed, in a recently published R2TP structure, RPAP3 binds to PIH1D1, and this assembly is recruited to the RUVBL1/2 hexamer by binding of the C-terminal domain of RPAP3 to the RUVBLs[47,49,50]. A recent structure of a RPAP3–PIH1D1 complex[50] supports our observation of an independent RPAP3–PIH1D1 subassembly. In previous work, RPAP3–PIH1D1 showed high-affinity binding to HSP90, and indeed HSP90 complex subunits coeluted in the lower molecular weight peak group of the adapter peak (Supplementary Fig. 4b).

Of the seven distinct assemblies, only the highly abundant prefoldin complex was identified and resolved in previous SEC–MS experiments[7]. Of note, PFDL and PAQosome subunits were identified in conventional whole cell lysate SEC–MS, but did not show detectable coelution (that is, overlapping peaks at high molecular weight)[18], exemplifying the increased sensitivity and scope for discovery of low-abundant protein complexes using the DIP-MS technology.

Next, we calculated the apparent subunit stoichiometry of the complexes (Supplementary Table 5) and compared it to the reported stoichiometries from structural studies (Fig. 3c). Complexes known to have a 1:1 subunit stoichiometry such as PFD, PFDL, CCT/TRiC–PFD, RNA polymerases (RNAPs) and R2TP were indeed close to their reported stoichiometry even in the case of the PFD complex and the PFDL that ectopically express affinity tagged subunits PFDN2 and UXT, respectively. These results indicate that DIP-MS derived stoichiometry values agree with those derived by structural biology methods and, more broadly, that complex stoichiometry tends to be maintained despite ectopic expression of individual subunits as previously proposed[1]. Then, we calculated the occupancy for both PFDN2 and UXT. Our data indicate that the PFD complex accounts for 84% of the total PFDN2 signal, while only a fractional amount (11%) of PFDN2 was found in the PFDL complex (Fig. 3d) and less than 5% of the PFDN2 signal is in the PAQosome. By contrast, when we performed DIP-MS using the PFDL subunit UXT as bait, we enriched the PFDL peak compared to the PFDN2 DIP-MS experiment by more than sixfold (roughly 74% of the total PFDN2) and the PAQosome peak close to fivefold, while the PFD peak was almost absent and likely represents a contaminant in the UXT purification (Extended Data Fig. 3a). When comparing the intensity for the same complex between the two tested bait proteins, we found enrichment of PFDL in UXT (log$_2$ fold change (FC) 1.88) and depletion of the PFD complex (log$_2$FC −12.4) (Extended Data Fig. 3b). For the PAQosome, we did not observe enrichment (log$_2$FC 0.015) between the two baits. DIP-MS with UXT thus enabled the validation of the PAQosome supercomplex that was enriched to a similar extent in the DIP-MS of PFDN2.

Comparison with previously published SEC–MS data[18] indicates that, as expected, the low expression of PAQosome and PFDL (average of 69 versus 203 normalized transcripts per million for exclusive PFD subunits) results in only a fractional amount of the total signal of $3.3 \times 10^5$ for PFDL and $2.5 \times 10^5$ for the PAQosome in SEC–MS compared to $1.0 \times 10^7$ and $6.7 \times 10^6$ in UXT DIP-MS (Extended Data Fig. 3c–e). This, in turn results in poorer detection of coelution, a problem alleviated by the prefractionation enrichment in DIP-MS.

To highlight the versatility of the method, we performed absolute bait quantification as previously described[51]. Briefly, an external calibration curve was built using a synthetic heavy peptide that corresponds to the tryptic peptide of the affinity tag. Based on this calibration curve, we estimated the absolute amount of PFDN2 in the DIP-MS inputs before separation being roughly 2.24 μg (Supplementary Data 3). Consequently, the PFD peak contains roughly 1.9 μg of PFDN2, while roughly 25 ng are present in the PFDL. The lowest signal measured for the PFDN2 across the DIP-MS gradient was estimated at roughly 22 fmol (average of DIP-MS PFDN2 replicates 1 and 2).

Overall, these two DIP-MS experiments provided an exhaustive account of the organization and architecture of prefoldin complexes. In addition to recalling previous knowledge, we were able to identify two structural variants of PAQosome subunits: the reported RUVBL1/2 heterohexamer and the RPAP3–PIH1D1 subassembly. The absolute quantification of the DIP-MS input allowed us to quantify that around 1–3 μg of purified bait is sufficient to resolve bait-containing protein complexes using DIP-MS.

### Discovery of an alternative PDRG1-containing PFD complex

We next turned our attention to PDRG1 predicted by PPIprophet as a genuine complex component[52]. The PFDN2 DIP-MS data showed that PDRG1 eluted in three separate peaks whereas UXT DIP-MS revealed only two peaks (Extended Data Fig. 4a). The first two PFDN2 DIP-MS peaks corresponded to the previously reported PFDL complex and PAQosome. In the third and lowest molecular weight peak, PDRG1 coeluted with canonical PFD subunits as shown in Fig. 4a. This finding is further supported by results from clustering the interaction probabilities from PPIprophet within the PFDN2 DIP-MS experiments of PDRG1 with PFD subunits and PFDL (Fig. 4b). PDRG1 was originally termed PFDN4-related protein (PFDN4r) due to its sequence homology (30% identity) to the canonical β-PFD subunit PFDN4 (ref. 30) (Extended Data Fig. 5a and Supplementary Data 4 and 5). To further validate our finding, we performed reciprocal AP–MS of four PFD subunits reported to be mutually exclusive for the canonical complex (PFDN1, VBP1, PFDN4, PFDN5) and PDRG1 (Fig. 4c). We recovered PDRG1 as a high-confidence interactor of all four baits used, confirming the interaction with canonical PFD subunits. At the same time, the AP–MS results using PDRG1 as bait protein identified both canonical PFD, PFDL and PFDL containing PAQosome complex members as interactors, consistent with the DIP-MS results (Fig. 4b). When comparing the PFD stoichiometry across AP–MS experiments, we observed a notably lower stoichiometry for PDRG1 in the PFDN4 purification compared to all other purifications, including the control samples (Extended Data Fig. 5b) suggesting that PDRG1 is a contaminant in the PFDN4 purification.

To better understand the organization of the PDRG1-containing prefoldin complex, dubbed PFD homolog (PFDh), we performed structural prediction of PFDh and PFD using ColabFold[53]. Both heterohexameric complexes were predicted with high confidence (weighted score 0.8 for canonical PFD and 0.77 for PFDh), resulting in the 'jellyfish' structure formed by the stacking of the α2β4-prefoldin subunits into the typical β-barrels and the six protruding tentacle-shaped coils[54] (Fig. 4d,e and Extended Data Fig. 5c–f). Multiple structural alignments by US-align[55] of the two predicted structures and an experimental PFD structure showed a large overlap of predicted PFD versus the PFDh (template modeling (TM) score 0.87). Both predicted structures displayed weak similarity to the experimental determined PFD complex (PDB 6NRD) (Fig. 4f), due to the absent tails in the experimental structure. Of note, the PFDN4 to PDRG1 switch allows for the N-terminal tail to extrude from the predicted structure, potentially forming an additional α-helix compared to PFD that may control substrates specificity of PFDh. Comparison of PDRG1 abundance in the PFD–PFDh peak to the abundance all other PFD subunits (Extended Data Fig. 5g) showed that PFD is 28.4 times more abundant than PFDh complex, indicating that only 3.5% of the peak can be attributed to PFDh complex (Extended Data Fig. 5h).

Thus, our data strongly indicate the presence of at least two similarly sized PFD complex isoforms. More work will be needed to identify the molecular function of the newly discovered PFDh (Fig. 4g).

### Identification of core PAQosome and PFDL components

PFDN2 and UXT DIP-MS also identified ASDURF coeluting with PFDL and the PAQosome complex (Extended Data Fig. 6a). In this regard, ASDURF was recently identified as a subunit of the PAQosome[52]. Indeed, reciprocal AP–MS of PAQosome and four PFDL components validated

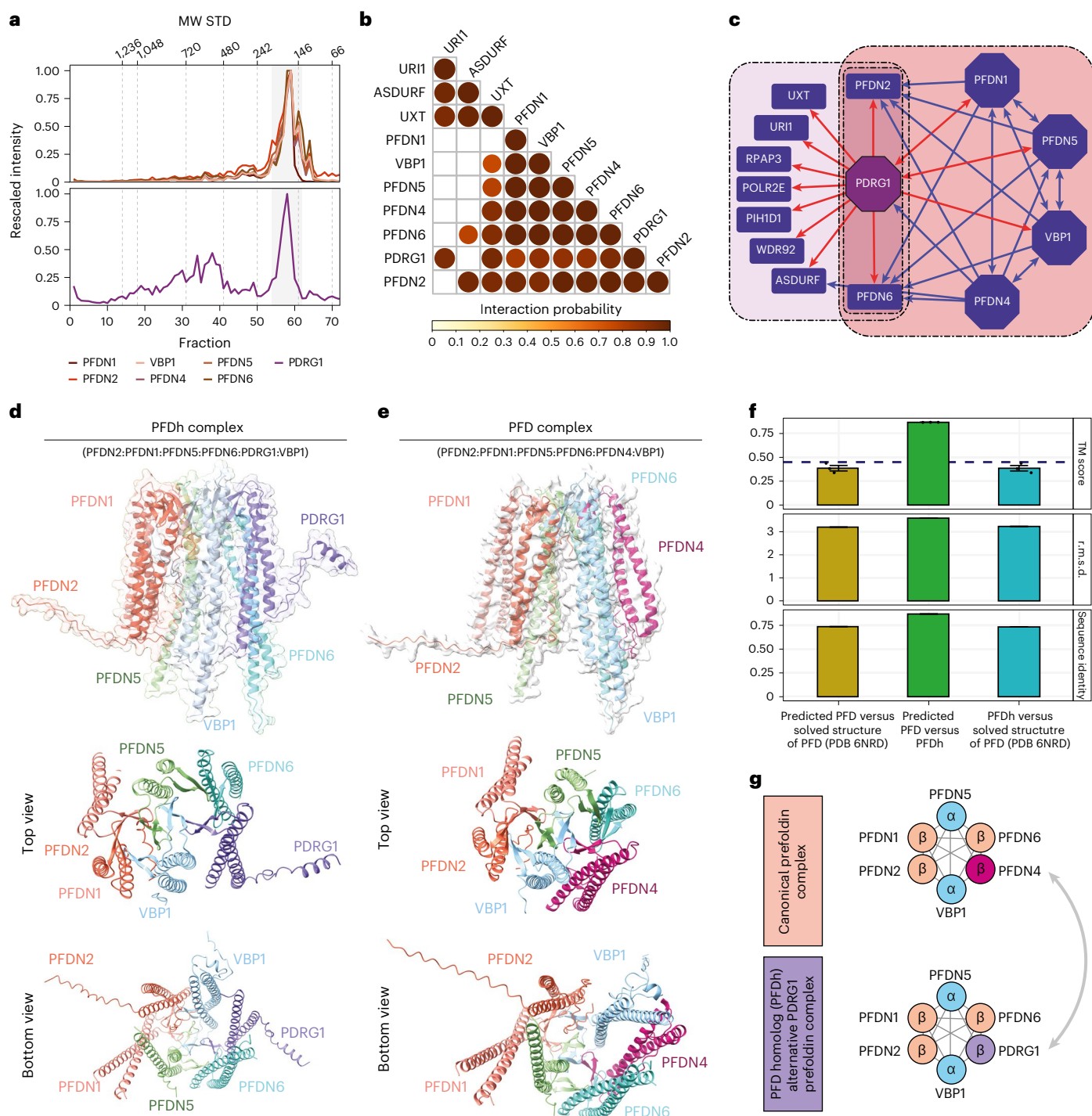

**Fig. 4 | Data-driven identification of alternative assemblies and complexes in the PFD interaction network. a**, PFDN2 DIP-MS coelution profile for canonical PFD complex (top panel) and PDRG1 (bottom panel). The y axis represents the normalized MS2 intensity across the replicates (n = 3 biologically independent experiments), while the x axis shows the fraction number. Molecular weight of the standards is highlighted on top of the plot. **b**, Triangular matrix representing the combined interaction probability derived from PFDN2 DIP-MS experiments (n = 3 biologically independent experiments) for the ten core subunits of the PFD and PFDL complex. **c**, PPI network from AP–MS data, using as bait proteins (octagons) mutually exclusive PFD subunits (PFDN1, VBP1, PFDN4, PFDN5) and PDRG1 (purple). PDRG1 interactions are shown in red. Canonical PFD complex (red background) and PAQosome subunits (pink) are indicated. Shared subunits (purple background) are in the new proposed assemblies PFDN2 and PFDN6 expanded by PDRG1. **d**, Structural model of the PFD homolog complex containing PDRG1 instead of PFDN4 (complex subunits are colored). **e**, Structural model of the canonical PFD complex. **f**, Structural alignment of the two predicted models of canonical PFD and PFD homolog and an experimental PFD (PDB 6NRD). The TM score indicates that the two predicted complexes show structural similarity (TM score >45) and some structural similarity to the experimental structure. TM score was derived by normalization to the length of the two complexes and the average length (n = 3 TM scores from one structural alignment). For r.m.s.d. and sequence identity, the value is the same for all three normalization approaches. Data are presented for TM score as mean values ± s.e.m. and black dots visualize TM scores after different normalizations. **g**, Proposed new PFD homolog complex containing PDRG1 instead of PFDN4.

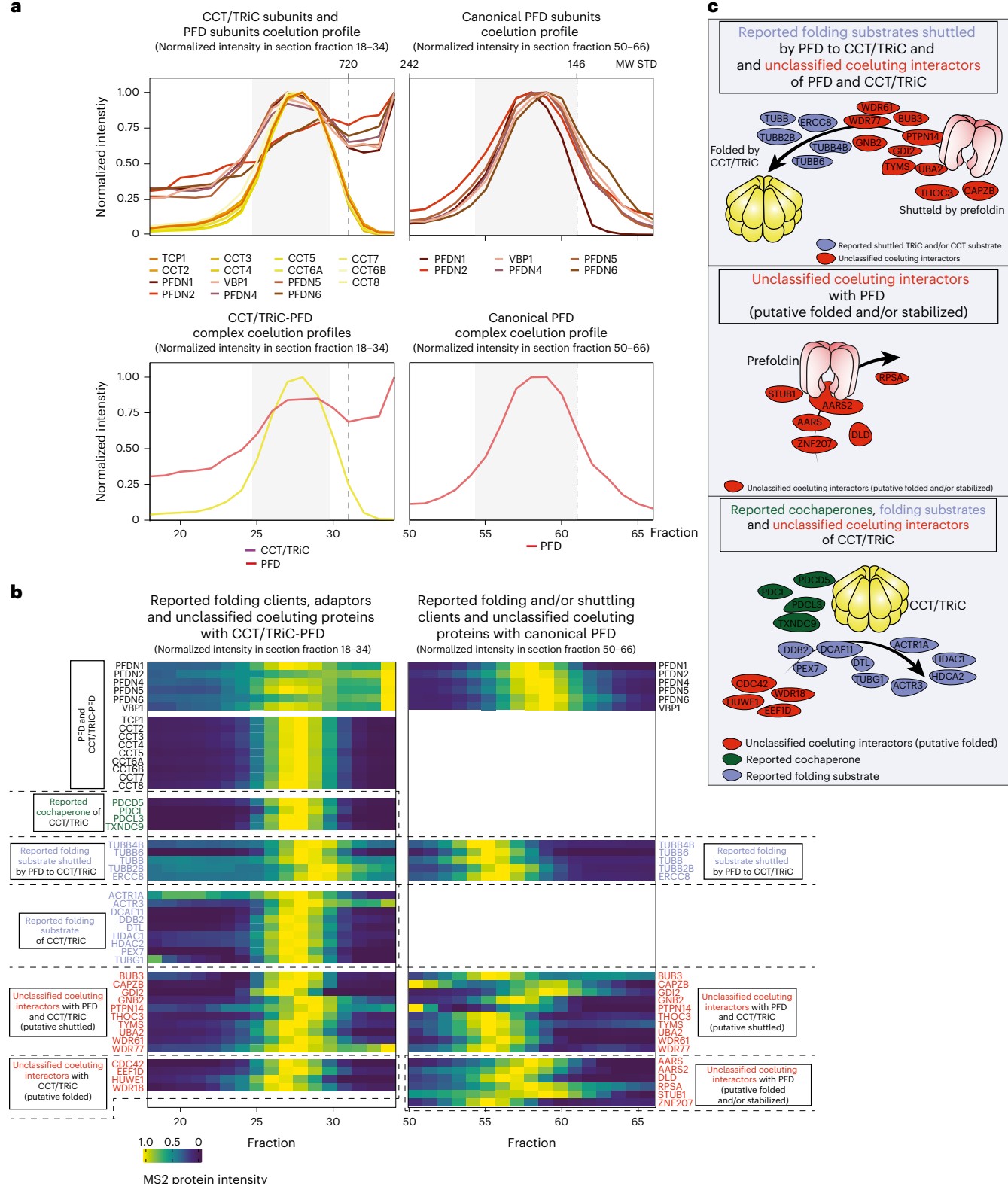

**Fig. 5 | Assignment of coeluting proteins for PFD and CCT/TRiC-PFD chaperone complexes. a**, Coelution profiles of all CCT/TRiC-PFD subunits, the summed CCT/TRiC-PFD complex, the summed PFD complex and the PFD complex subunits. The bottom x axis represents the fraction number, while the top x axis shows the molecular weight from the gel standards. The y axis shows the rescaled protein intensity. **b**, The heatmap showcases reported folding substrates (blue labels) and adapters (dark-green labels) and new unclassified coeluting proteins (red labels). The coeluting proteins are grouped with

CCT/TRiC and PFD (potentially shuttled), PFD alone (potentially folded and/or stabilized by prefoldin) and CCT/TRiC alone (potentially folded by CCT/TRiC). Cell colors of the heatmap represents the MS2 protein intensity rescaled to the maximum intensity of each protein within the section of gel fractions represented. **c**, Graphical representation of the identified coeluting proteins and their interacting complexes. Red circles represent newly identified coeluting proteins, blue circles indicate reported folding substrates and dark-green circles represent reported cochaperones.

our DIP-MS findings (Extended Data Fig. 6b). Binary structural alignments using US-align[55] of predicted structures of ASDURF and the other prefoldin subunits showed a consistently high structural alignment for ASDURF versus β-prefoldin subunits (Extended Data Fig. 6c) (average TM score of 0.563 and a root-mean-squared deviation (r.m.s.d.) of 2.03 Å) but not with R2TP and CCT/TRiC subunits (average TM score of of 0.252 and an r.m.s.d. of 3.68 Å, Extended Data Fig. 6d). To obtain an acceptable structural model of the PFDL complex, we had to cut away the intrinsically disordered C terminus of URI1 (Extended Data Fig. 6e) to model the PFDL complex with a weighted confidence score of 0.75 (Extended Data Fig. 6f and Supplementary Results). Thus, our DIP-MS and AP–MS data, and orthogonal structural prediction, identified ASDURF as PFDL subunit and a component of the PAQosome supercomplex.

Last, the UXT DIP-MS data consistently identified a PPI between the PFDL complex and POLR2E, which was often considered an associated protein for the PFDL-module, but not a core component. Of note, in all PFD2 and UXT DIP-MS experiments we observed the PFDL complex coeluting with POLR2E (Extended Data Fig. 6a). AP–MS analysis of any tagged PFDL-like subunit (URI1, UXT, PDRG1, PFDN2) identified POLR2E, strongly suggesting that POLR2E is constitutively associated with the core subunit of the PFDL complex in vivo. Accordingly, it has been shown that URI contains a dedicated high-affinity POLR2E binding domain, suggesting that the PFDL complex is tightly linked to POLR2E in the absence of other PAQosome members[30]. These results indicate that the in vivo PFDL complex deviates from the hexameric paradigm in current literature[56,57] and suggest an updated hetero-heptameric PFDL complex containing POLR2E.

### Identification of canonical PFD folding clients by DIP-MS

We leveraged the increased sensitivity of DIP-MS to identify reported and putative novel folding substrates interacting with PFD and CCT/TRiC complex.

We used the PPI network derived from PPIprophet (10% FDR threshold) and selected the subnetworks corresponding to proteins interacting with CCT/TRiC and PFD (Fig. 5) or the PAQosome (Fig. 6). Within these subnetworks, we superimposed the available knowledge of the respective complexes to identify complex–complex interactions, while interactions with proteins absent from the complex databases were classified as protein–complex interactions (PCIs). Complex–complex interactions represent potentially high-order assemblies and specialized machines whereas PCIs comprise known canonical substrates and cochaperones as well as unclassified coeluting interactors.

Both canonical PFD and CCT/TRiC-PFD complexes showed excellent coelution of all subunits (Fig. 5a). Proteins and complexes, such as actin, tubulin subunits and heat shock protein multimers, have been reported to shuttle between PFD and CCT/TRiC for folding[20–22]. DIP-MS recapitulated these findings by detecting their coelution with both the PFD and the CCT/TRiC-PFD supercomplex peaks (Fig. 5a,b). Besides PFD and CCT/TRiC subunits we identified 38 additional proteins coeluting with CCT/TRiC-PFD and/or PFD that are known interactors of PFD and CCT/TRiC subunits. These include four known cochaperones, 14 known folding substrates and 20 proteins not classified yet. Among those unclassified, we found four exclusively coeluting with CCT/TRiC, six with PFD and ten with both indicating shuttling between these two complexes, which showcase the high degree of granularity achievable by DIP-MS. Furthermore, 30% of unclassified proteins contain tryptophan-aspartic acid repeats similar to the known substrates we identified (36%). Among the unclassified proteins we found the G-protein GNB2, a previously reported interaction partner of PFD subunits and PDRG1 (refs. 5,58) adding orthogonal evidence for PFDh. GNB2 coeluted with both, the PFD–PFDh complex and the CCT/TRiC-PFD, suggesting that GNB2 shuttles via PFD–PFDh to CCT/TRiC. We validated this interaction by reciprocal AP–MS (Extended Data Fig. 7a) and found additional evidence of the GNB2-PFD PCI in several

large-scale AP-MS datasets[5,58]. GNB1 and other G-proteins are folded by CCT/TRiC in an interplay with phosducin-like cochaperones (PDCL, PDCL3)[59,60]. In line with this, we identified coelution of PDCL and PDCL3 with CCT/TRiC (Fig. 5b,c).

In UXT DIP-MS only two-thirds of the CCT/TRiC-PFD and PFD core subunits were identified and these were present at much lower levels compared to PFD2 DIP-MS (Extended Data Fig. 8a,b), with PFDN2 the CCT/TRiC-PFD subunits being enriched on average by 7.72 $\log_2$FC (Extended Data Fig. 8c), and the corresponding coeluting proteins were more complete in the PFDN2 DIP-MS and enriched 6.26 $\log_2$FC compared to the UXT DIP-MS (Extended Data Fig. 8d–f). Due to its high expression, CCT/TRiC has been repeatedly found as background in APs[61] and we believe that CCT/TRiC is not a specific UXT interactor[61]. Thus, we expect that also CCT/TRiC clients should be either absent in the UXT DIP-MS or less abundant compared to PFDN2 DIP-MS. Indeed, only ten (19%) could be detected in the UXT DIP-MS experiment and were recovered with lower abundance compared to the PFDN2 DIP-MS. Overall, our data suggest a broader role for PFD in the stabilization of unfolded nascent proteins and their transport to CCT/TRiC (Fig. 5c). The presence of additional PFD complexoforms, such as the newly identified PFDh, could be partially responsible in broadening the spectrum of the prefoldin substrates.

### Identification of PAQosome client complexes and clients

Next, we queried UXT and PFDN2 DIP-MS data for client complexes and client proteins of the fully assembled cochaperone PAQosome. The PAQosome is a large multiprotein assembly that assists HSP90 in the assembly of protein complexes, such as small nuclear ribonucleoproteins involved in messenger RNA splicing (small-nuclear ribonucleoproteins (snRNPs) U4 and U5)[62], the three RNAP complexes[43,63] and box C/D small-nucleolar RNP (snoRNP) assemblies[64]. Furthermore, the PAQosome stabilizes phosphatidylinositol 3-kinase-related kinases and many other client complexes[65]. Using previous knowledge on client complexes (75 proteins, 13 complexes, Supplementary Methods), we identified 45 of these 75 clients in the UXT DIP-MS data: 91% of which (n = 41) were scored by PPIprophet as PAQosome interactors. In addition, we could assign dozens of additional proteins including candidate clients to the PAQosome.

Most of these target proteins coeluted with the major PAQosome peak (Fig. 6a) or the adapter peak formed by RPAP3-PIH1D1 (Fig. 3b). Based on coelution and evidence from orthogonal AP–MS datasets, we assigned 92 proteins (organized in 19 complexes) to the PAQosome and 15 cochaperones (organized in four assemblies) to the adapter peak (Fig. 6b).

Among the known PAQosome client complexes, we recovered all three RNAPs[43,63]. Besides these, we found additional polymerase associated proteins such as GPN-loop GTPase 1/3 (GPN1, GPN3) and RPAP2, known assembly factors for RNA Pol II that associate with RNA Pol II before nuclear import[66,67]. We also recovered multiple small-nuclear RNA (snRNA) assemblies (U2, U5 and PRPF19), which together with ZNHIT2 regulate snRNA complex formation[33] (Fig. 6c). Also the box C/D snoRNPs U3, a complex whose assembly is linked to the PAQosome[64] and the KAP1/TRIM28, a transcriptional repressor that interacts with URI1 to recruit PP2A leading to TRIM28 dephosphorylation and repression of TRIM28 regulated retrotransposons were recovered[68]. Additional candidate client complexes not yet reported for the PAQosome include the prohibitin complex or the anaphase-promoting complex (APC/C). Previous studies identified interactions between APC/C subunits and RPAP3 and PFDN2 (refs. 56,69), but not with the entire PAQosome. Whereas AP–MS failed to detect APC/C enrichment, DIP-MS recovered eight PPIs between PAQosome subunits and APC/C subunits suggesting that this is a low-abundant interaction, only accessible through substantial increase in analytical sensitivity.

In the lower molecular weight coelution group (apparent molecular weight of 356 kDA or fraction 45) consisting of the adapter and/or

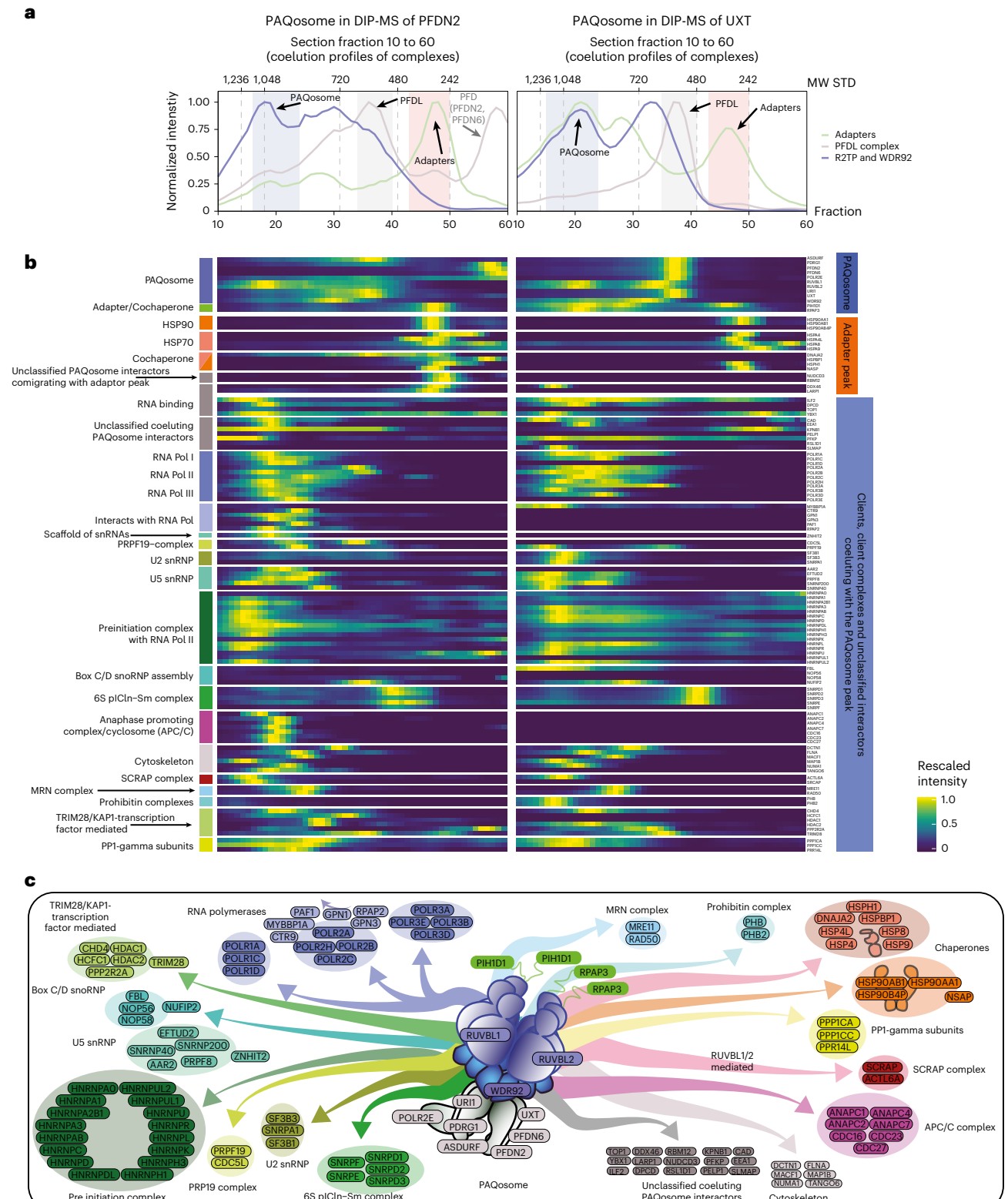

**Fig. 6 | Assignment of PAQosome clients and client protein complexes.**
**a**, Coelution profiles for adapters complex (green line), PFDL complex (gray line) and R2TP module (blue line). The *x* axis represents the fraction number (from 10 to 60) while the *y* axis represents the rescaled intensity. Colored boxes represents either the PAQosome (blue box) or the adapter peak (red box, RPAP3, PIH1D1). The top *x* axis shows the molecular weight from the gel standards. The two facets represent the two bait proteins used. To investigate the peaks within the section, see Extended Data Fig. 2 and the respective coelution group. MW, molecular weight. **b**, Heatmap showcasing the identified clients within the UXT and PFDN2 DIP-MS data. The annotation on the right highlights whether the protein is interacting with either the adapter peak (orange) or the major PAQosome assembly (blue). The clients were further deconvoluted by literature-based knowledge into assemblies, which are indicated on the left-hand side annotation. The color code corresponds with the graphical representation below. Cell color represents the rescaled intensity. **c**, Graphical representation of the identified PAQosome client complexes.

regulatory subunits RPAP3 and PIH1D1 of R2TP, we observed a coelution of HSP90 subunits, which are reported to independently form a complex with RPAP3 at a ratio of 2:1 or 2:2 (ref. 70) (Fig. 6b). Similarly, in the PFDN2 DIP-MS HSP70 subunits coelute, showing additional client chaperones for the two R2TP adapters[50,70]. Quantitative comparison of PAQosome coeluting proteins is shown in Extended Data Fig. 9 and detailed in Supplementary Results. Since DIP-MS can resolve prey–prey interactions, we uncovered an additional 1,117 PPIs between the client complexes, exemplifying the high-density of the DIP-MS generated PPI network[71].

Overall, the identification of a large portion of reported clients for both the CCT/TRiC and the PAQosome as well as novel putative client complexes and client proteins demonstrates the resolution of DIP-MS for dissection of a particular interaction network of interest.

## Discussion

In this study we introduce a high-throughput method dubbed DIP-MS to deconvolute the composition of affinity-enriched protein complexes, yielding insights into contextual protein–complex organization at an unprecedented depth. We benchmarked DIP-MS versus the two state-of-the art MS-based techniques to resolve protein–complex composition and showed that DIP-MS combines the specificity of AP–MS while benefiting from the larger number of interactions recovered from fractionation-based approaches.

As DIP-MS experiments contain various level of information including PPIs, complex–complex interactions, subassemblies and stoichiometry, we developed PPIprophet to facilitate the analysis of DIP-MS datasets. PPIprophet uses deep learning to predict PPIs in DIP-MS data and applies FDR correction using standard target–decoy competition to distinguish true interactors from spurious, coeluting proteins.

Other frameworks for analysis of SEC–MS data such as PCprophet and EPIC[72] may report different results. However, although these approaches rely to a lesser extent on previously reported complexes, they still use previous knowledge for FDR control (PCprophet) or network pruning (EPIC) and are hence not suited for fully knowledge-free searches.

To demonstrate the applicability of DIP-MS to identify all types of interaction (core, accessory, complex–client and complex–complex) we applied DIP-MS to the protein–complex landscape of the prefoldin and prefoldin-like complexes. This system exemplifies the complexities of modular protein organization and its fundamental role in cellular proteostasis.

Our DIP-MS experiments recapitulated two and a half decades of previous prefoldin characterization and cover more than 184 interaction studies. The data recovered subassemblies like the R2TP complex, complex–complex interactions such as the CCT/TRiC-PFD or the PAQosome and discovered an alternative prefoldin complex containing PDRG1 as core subunit. Our data further confirmed recently reported interactions such as PFDL subunits with ASDURF and assigned POLR2E as a constitutive subunit of the PFDL complex.

Overall, DIP-MS identified a large fraction of known PFD and PFDL clients and client complexes and, ultimately, advanced our understanding on the modular organization of this section of the human interactome. While DIP-MS can detect various classes of interacting proteins, their functional classification into clients, adapters or chaperones can only be achieved through literature-based information or additional experiments.

Even though DIP-MS outcompeted reciprocal AP–MS in terms of number of identified interactions, it should be noted the DIP-MS data did not recapitulate all the tested interactions. This may be due to the following reasons: (1) signal dilution, following extensive biochemical fractionation may compromise accurate coelution profiling of low-abundant proteins, (2) true sample and/or state specific differences or (3) the complex stability influenced by the separation conditions used in the gel, which could result in loss of interacting proteins or disassembly of large supercomplexes.

Furthermore, while our scoring approach uses a decoy-based solution to the problem of coeluting proteins, development of more sophisticated statistical frameworks might be beneficial to further filter contaminants and increase specificity in complex identification (for example, based on CRAPome[61]). With an increasing number of DIP-MS experiments analyzed and annotated, contaminant complexes could be more specifically separated from true interactors hence benefiting all interactome studies by providing assembly-state context for common contaminant proteins. While techniques with greater theoretical resolution such as cryo-slicing-BNP[73] have been developed, they require specialized equipment and a great amount of knowhow, thereby being practically impossible to transfer between laboratories.

DIP-MS allows the characterization of all bait-containing protein complexes from roughly roughly 1 µg of bait-protein–complex purified from approximately $6 \times 10^8$ HEK293 cells. For comparison, reciprocal AP–MS study of 11 PFDN–PFDL baits required five time more cells.

Since our high-throughput DIA–MS approach allows complex resolution for a single bait within a day, this cuts acquisition time over previous cofractionation studies by at least threefold[7,74]. Further, the high reproducibility of native-PAGE separation will enable the probing of a selected bait-protein–complex landscape not only at steady state, but across different cellular states, which is difficult to achieve by techniques such reciprocal AP–MS.

We foresee the application of DIP-MS as a valuable approach for high-resolution interactome studies. Once applied under perturbation conditions DIP-MS will increase our understanding of the dynamic nature of modular proteome organization to better understand the functional relationship between proteotype and phenotype.

## Online content

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

## Methods

More detailed information about the methods used is provided in the Supplementary Material and Methods section.

### Reciprocal AP–MS

For 11 core subunits of PFD and PFDL containing PAQosome complexes, cell lines expressing twin-Strep and hemagglutinin (SH)-tagged baits were generated (Supplementary Table 3). For each bait, we performed triplicate experiments. Data acquisition was performed in data dependent acquisition (DDA) mode on the same MS platform as the DIP-MS samples. The exact experimental details are outlined in the Supplementary Methods.

### Purification of complexes for BNP separation

The affinity enrichment of protein complexes for BNP separation followed the protocol for AP–MS samples. For a DIP-MS replicate, 30 confluent 150 mm plates ($6 \times 10^8$ HEK293 cells) were lysed in HNN-lysis buffer (50 mM HEPES, 100 mM NaCl, 50 mM NaF, pH 7.4) supplemented with protease inhibitor cocktail, 1 mM phenylmethylsulfonyl fluoride, 400 nM vanadate, 1.2 µM avidin and 0.5% NP40. Samples were treated with 5,625 U of benzonase and incubated at 10 °C at 500 r.p.m. for 45 min before clarification by centrifugation at 16,000$g$ at 4 °C for 20 min. Then 30 ml of cleared lysate were separated into 7.5 ml aliquots and transferred to four 15 ml falcon tubes. For each tube, 200 µl of equilibrated 50% Strep-Tactin Sepharose beads slurry was added and subsequently incubated on an end-over-end rotator at 12 r.p.m. for 45 min. Cleared lysate was loaded on Bio-Spin chromatography columns. Beads were washed twice with 1 ml of ice-cooled HNN-lysis buffer and three times with 1 ml of HNN buffer without supplements. Purified complexes were eluted three times with 200 µl of 2 mM Biotin buffer. The resulting elution volume of 600 µl per replicate were pooled together and concentrated over a 30 kDa molecular weight cutoff filter at 4 °C and 3,000$g$ to 35–50 µl.

### Separation of copurified complexes by BNP

To separate copurified protein complexes, 35–50 µl of the concentrated sample were loaded on a BNP. The native separation procedure followed previous protocols[13]. First, the concentrated eluate was mixed at a 1:4 ratio with native gel loading buffer by carefully pipetting gently up and down. From this mixture, 45 µl were loaded with gel loader tips on the native-PAGE 3–12% Bis-Tris precast protein gels. For replicate one and two of PFDN2 aliquots of 1 µl and replicate 1 of UXT an aliquot of 0.5 µl was taken away for absolute bait-protein quantification. NativeMARK molecular weight standard was added as the standard. As cathode buffer a Light Blue Cathode Buffer was applied, otherwise the manufacturer's instructions were followed. The BNP was run for 3 h at 4 °C with constant voltage: 120 V for 25 min, 160 V for 2 h and 5 min and 200 V for 30 min (Supplementary Figs. 5a and 6a for BNP images).

### MS-sample preparation of gel slices

The native-PAGE gel was rinsed with deionized $H_2O$ (diH$_2$O) before washing it $3 \times 30$ ml of $H_2O$ for 5 min. Following the initial gel washing step, the gel was stained for 1 h with SimplyBlue SafeStain. After removal of the staining solution, the BNP was rinsed with diH$_2$O before destaining overnight in diH$_2$O. Gels were imaged with a Fusion FX (VILBER), before slicing. The molecular weight standard was noted on a millimeter paper and the line of each replicate was vertically cut to separate each replicate. An in-house designed gel-slicing tool with hundred 1 mm distanced razer blades mounted on a metal frame, was applied to each lane.

The slices were transferred to a 96-well glassfiber filter plate, which contained 200 µl of $H_2O$ (for more detail regarding sample preparation optimization, see Supplementary Fig. 7 and Supplementary Methods). The filter plate was equilibrated by washing twice with 200 µl of 100% acetonitrile (ACN) followed by one wash of 200 µl of 50% methanol (MeOH) in 20 mM ammonium bicarbonate (ABC). The washing solutions were removed by centrifugation at 700$g$ for 5 min at room temperature. The position of the gel slices on the filter plate were randomized. Next, slices were destained by addition of $3 \times 200$ µl 50% MeOH in 20 mM ABC followed by two washes with 200 µl of 100% ACN with 5 min of incubation before each centrifugation step. Reduction was performed by addition of 50 µl of 25 mM TCEP in 20 mM ABC at 90 r.p.m. at 37 °C for 30 min followed by addition of 50 µl of 50 mM IAA in 20 mM of ABC and incubation in the dark at room temperature for 45 min. The slices were washed with 200 µl of 50% ACN in $H_2O$, followed by $2 \times 200$ µl of 100% ACN. To each well, 50 µl of the digestion mix, containing 0.5 µg Trypsin, 0.1 µg lysyl endopeptidase and 0.01% ProteaseMax in 20 mM ABC was added. After 25 min of incubation at 37 °C at 100 r.p.m., an additional 100 µl of 20 mM ABC was added to cover all gel slices. Protein digestion was performed overnight at 37 °C with 100 r.p.m. To avoid evaporation of the digestion mix, the filter plate was closed with parafilm at the bottom and on top with a metal cover lid. Peptides were collected by centrifugation at 700$g$ for 5 min and transferred to LoBind tubes. The filter plate was washed once with 100 µl of 50% ACN in $H_2O$ followed by a wash with 100 µl of 100% ACN. The washing solutions were pooled with the collected peptides. Samples were dried at 45 °C on a vacuum drier and stored at −80 °C until MS acquisition.

### DIA of native-PAGE separated AP samples

The DIP-MS samples were acquired in DIA mode with a Q Exactive Plus Hybrid Quadrupole-Orbitrap Mass Spectrometer interfaced with the EvosepOne system. First, the dried peptides were dissolved in 250 µl of buffer A (0.1% formic acid in $H_2O$) with 1:2,500 (v/v) iRT peptides (Biognoysis). Dried peptides were sonicated for 10 min and centrifuged at 16,000$g$ for 10 min. To avoid loading small gel pieces, which might pass the glassfiber filter, 230 µl of the 250 µl were loaded on equilibrated Evotips. The C18 material of the Evotips was activated with 10 µl of Buffer B (98% ACN and 0.1% formic acid in $H_2O$) and by soaking the tips in Propan-2-ol. Next, the tips were equilibrated by adding 10 µl of buffer A, following by the addition of 230 µl of resuspended peptides per fraction. Loading was completed by centrifugation at 300$g$ for 5 min. To prevent drying of the C18 material, 200 µl of Buffer A were added on top of the tips.

Peptides were separated on a fused silica PicoTip with an inner diameter of 100 µm and 50 µM tip diameter, in-house packed with 8 cm of C18 beads (MAGIC, 3 µm, 200 Å, Michrom BioResources). The peptides were separated using the '60 samples per day' method (24 min gradient for PFDN2 and 21 min gradient for UXT as bait protein) using the EvosepOne system. The mass spectrometer was operated in positive mode with the capillary heated at 275 °C and maintained at 2.5 keV. We used for DIA 22 variable windows with +1 Dalton (Da) overlapping on the upper window boarder, ranging from 350 to 1,650 $m/z$. The full MS1 scan was performed over a mass to charge range of 150 to 2,000 $m/z$ with a high resolution of 70,000 fixed at 200 $m/z$. The automatic gain control target was set to $3 \times 10^6$ with a maximum accumulation time set to 200 ms.

For MS2 scans the resolution was fixed to 17,500 with an automatic gain control target of $2 \times 10^5$ with high collision density fragmentation in stepped mode using collisional energies of 25, 27 and 30%, normalized to 500 $m/z$ at charge state +1. Each MS2 scan was set to 50 ms leading to a total cycle time of 1.3 s. For optimization of the DIA-measurement method, see Supplementary Methods and Supplementary Fig. 8.

### Reciprocal AP–MS data processing

The reciprocal AP–MS samples were analyzed with MaxQuant (v.1.5.2.8) and the built-in search engine Andromeda[75]. Raw files were searched against the human protein database obtained from UniProtKB[76] (downloaded on the 1 December 2019) and supplemented with the protein

sequence of green fluorescent protein (GFP) and the SH-quant peptide (AADITSLYK)[51]. For the search, the MaxQuant contaminant list[75] was included. The peptide identification search was performed with default parameters. Carbamidomethylation on cysteine residues was selected as fixed modification while oxidation on methionine residues and acetylation on the N terminus were used as variable modifications. The maximal number of modifications was limited to five. Furthermore, only trypsin-specific peptides, allowing up to two missed cleavages, was set. For label-free quantification, the default parameters were enabled. Requantification and match between runs were enabled with default parameters. The peptide and protein false discoveries were controlled by a 1% FDR.

### Postanalysis of AP–MS data

The MaxQuant 'proteinGroups' table was filtered (removed contaminants) before SAINTExpress scoring. First decoys that passed the FDR were removed from the results. An additional GFP control originating from a high pH fractionated sample was added (Supplementary Methods)[16]. The final matrix was uploaded to CRAPome[61] to perform SAINTExpress[77] scoring. Each bait was scored independently against GFP controls, as their interactomes are largely overlapping and scoring them together reduces the number of interactions recovered. For SAINTExpress, default parameters were used, with the adaptions that ten virtual controls for FC calculation were applied and 4,000 iterations and normalization for SAINT (Significance Analysis of INTeractome software) score calculation. Next, scored interactions were filtered by applying a $\log_2 FC$ ($FC_A$) $\geq 2$ with a SAINT score greater than or equal to 0.95. Further, only interactors with spectral counts equal to or more than five were kept. Second, preys were filtered against the CRAPome dataset, applying a 30% frequency threshold (excluding from CRAPome some well-characterized PFD–PFDL interactions CCT/TRiC subunits, HSP90 and TUBB2B). This resulted in 407 binary PPIs from 174 interaction partners, which we categorized into 278 high-confidence interactions ($\log_2 FC \geq 5$ and SAINT score $\geq 0.99$) and 140 medium-confidence interactions ($\log_2 FC \geq 2$ and a SAINT score between 0.95 and 0.99) (Supplementary Fig. 9).

### DIP-MS data analysis

The DIA data were searched with Spectronaut (v.13.12.200217.43655, Laika), using library-free directDIA against the human protein FASTA database downloaded from UniProtKB[76] (downloaded on the 1 December 2019) and supplemented with indexed Retention Time (iRTs)[78] and SH-quant peptide[51]. The fasta file contained in total 20,366 entries, which were supplemented by decoy sequences within Spectronaut. The analysis was conducted with default (BGS Factory settings) parameters with minor adaptions. Briefly, the peptide identification search was performed for tryptic peptides, allowing for up to two missed cleavages and a maximum length of 52 and a minimum length of seven amino acids. Carbamidomethylation of Cysteine residues was set as fixed modification (+57 Da) and N-terminal acetylation and oxidation on methionine residues as variable modifications. A maximum of five modifications per peptide was allowed. Precursor and protein $Q$ value cutoff was set to 5%. For quantification, the cross-run normalization and the best $N$ fragments per peptide parameters were disabled. Quantification was performed on MS2 level, and the mean peptide quantity from all quantified fragments per stripped peptide sequence was reported. For PFDN2 and UXT DIP-MS we overall could reconstruct the migration profile of 1,465 proteins and 737 proteins, respectively (Supplementary Data 1). General characteristics of the elution profile are reported in Supplementary Figs. 4a–f and 5a–f.

### Postprocessing of DIP-MS data

From the Spectronaut analysis, protein accessions, stripped peptide sequences and peptide quantities per fraction for each replicate were exported. Each DIP-MS replicate was processed in R using the filtering functions of the CCprofiler[18] R package. We first filtered within each gradient the noisy peptide profiles by applying a consecutive protein ID-based stretch filtering of two fractions, which removed inconsistently quantified peptides. In addition, all nonproteotypic peptides were removed. Next, sibling peptide correlation was performed, to remove peptides that do not show coelution across the separation range. An absolute sibling peptide correlation cutoff of 0.2 was applied. After signal processing on the peptide level, protein quantities were inferred by using the top two highest intense peptides per protein. The protein matrices were used for visualization of protein complexes and served as input for the PPIprophet. These conservative filtering and quantification parameters ensured (1) no noisy single hit wonders were used for PPI-identification or complex mapping, and (2) that the intensities for each protein were comparable against each other.

### PPIprophet implementation

**Quantitative protein matrices preprocessing and feature engineering.** The training set was built using 32 datasets encompassing different separation techniques, number of fractions and organisms for a total of 1,675,356 PPIs[40]. Multiple organisms and separation techniques were used to maximize the model generalization capabilities. Positive PPIs were derived from STRING (STRING combined score >600)[79] while to obtain the negative labels random protein pairs showing weak correlation were used (correlation between −0.3 and 0.3) leading to a balanced dataset between positive and negative interactions. Protein profiles were smoothed using one-dimensional discrete Fourier transformation and missing values were filled with the average value between the two-neighboring fraction. Following data smoothing and missing value imputation, the intensity vector was rescaled in a 0–1 range. To have a fixed-size input for learning we used linear interpolation to rescale the fraction number to 72. For training, two types of continuous feature were calculated, similar to the ones used in our recently introduced PCprophet toolkit[80]. The features used by PPIprophet are: (1) sliding-windows correlation ($w = 6$ fractions) and (2) fraction-wide difference between protein intensity resulting in $2n$ features and 144 features when $n = 72$.

### Deep-learning model construction and training

Following data annotation, a DNN was constructed in Python v.3.8 in Keras (https://keras.io) using Tensorflow2 (https://www.tensorflow.org) as backend. Input layer size was fixed to the number of features (144). For the other three layers, 72 neurons were used with rectified linear unit as activation function. To avoid overfitting, 30% dropout was used for the hidden layers. In the final layer, sigmoid activation was used to classify coeluting and not coeluting PPIs. The model was trained using ADAM (learning rate of 0.001) and binary cross-entropy as loss function. To further mitigate overfitting, label smoothing of 0.1 was applied. The dataset was split into a training and testing set using an 80/20 split and, the training set was further split in training and validation set using 70% of the data for training and 30% for validation. The model was trained for 256 epochs using a batch size of 64. EarlyStopping (patience, 10) was used to avoid learning plateau and the best model was selected based on lowest validation loss, which was calculated after every epoch. Achieved performance metrics on the test set are reported in Supplementary Table 2.

### PPIprophet analysis

For new data, a correlation matrix between all protein pairs is computed and nonnegatively correlating pairs are then used for feature construction and deep-learning prediction. For every protein, a decoy PPI is generated by random selection of protein pairs absent from the target set previously generated. After generation of both target and decoy PPIs, features are calculated as previously described and the DNN model is used to discriminate coeluting and not coeluting PPIs.

Following prediction, PPI probabilities from the DNN model are converted to empirical $P$ values and FDR is controlled using the following formula[81].

$$\text{FDR}\left(\text{PPI}_1, \text{PPI}_2, \text{PPI}_3, \dots, \text{PPI}_k\right) = \min_{i \geq k}\left(\frac{m \times \pi_0}{i} \times P_{(i)}\right)$$

where $\pi_0$ is the probability that a putative discovery is false, $k$ is the total number of selected discoveries and $m$ is the number of putative discoveries, where for discovery is intended a PPI above the current probability interaction threshold. For every experiment, an FDR cutoff of 10% is used. If replicates are present, the prediction probabilities for a particular PPI are combined into a weighted joint probability across replicates under the assumption of independence between the different replicates.

Following joint probabilities calculation, a combined adjacency matrix is generated where every edge is represented as the joint probability for that specific PPI. This combined adjacency matrix can be thought of as a series of in silico purification experiments where every column is a bait and every row is a prey

$$\begin{vmatrix} & \text{bait}_1 & \text{bait}_2 & \text{bait}_3 & \text{bait}_j \\ \text{prey}_1 & X_{1,1} & X_{2,1} & X_{3,1} & X_{j,1} \\ \text{prey}_2 & X_{1,2} & X_{2,2} & X_{3,2} & X_{j,2} \\ \text{prey}_3 & X_{1,3} & X_{2,3} & X_{3,3} & X_{j,3} \\ \text{prey}_j & X_{1,j} & X_{2,j} & X_{3,j} & X_{j,j} \end{vmatrix}$$

The score $W$ for the interaction $\text{bait}_j\text{prey}_j$ is calculated assuming independence of prey and bait interaction from other interactions and is performed in vectorial format for computational efficiency.

$$W = X_{j,} \times \frac{n}{n_0} \times \sqrt{\sum_{1}^{n}(X_{j,} - \mu)^2 + (\mu^2 \times n_1)}$$

where $X_{j,}$ represents the $j$th column, $n$ is the total number of elements in the $j$th column, $n_0$ and $n_1$, respectively, represent the number of negatively predicted interaction (probability less than 0.5) and positive predicted interaction (probability greater than 0.5 and FDR lower than the user-set target FDR) in the $j$th column. The variable $\mu$ represents the average probability in the $j$th column and is defined as:

$$\mu = \frac{\sum_{1}^{n}(X_{j,})}{n}$$

Thereby $\mu$ intrinsically represents the specificity of the bait $j$. The term $\sum_{1}^{n}(X_{j,} - \mu)^2$ represents the square error compared to the bait, which translates into penalizing proteins having similar probability to the average in the column, with the rational that true interactions have low $\mu$ and high square error $\mu$. Following conversion of combined probabilities to scores, a bootstrap procedure is applied to threshold the scores and to further filter the interactions.

### Benchmark versus reciprocal AP–MS and SEC–MS

Each dataset was analyzed separately using SAINTExpress, PPIprophet or CCprofiler using different thresholds for each of the tools. In this regard, we used a strict 0.99 threshold for SAINTExpress as outlined in postanalysis of AP–MS data (Supplementary Methods), a threshold of 10% FDR for PPIprophet and a CCprofiler $Q$ value of less than 1% as reported in our previous study[18]. The CCprofiler derived complexes were converted to a PPI network and used as is while for PPIprophet, positively predicted PPIs were selected and used directly. For comparison of abundances between SEC–MS and DIP–MS, for all proteins from the target list identified in the two experiments we selected the most abundant peak and averaged across replicates if present.

### Data analysis for DIP-MS of PFDN2 and UXT

To calculate ratios between PFD, PFDL and PAQosome complex, the protein elution profile of PFDN2 respective to UXT was used. Following peak selection, we assigned manually the PFD and PFDL peak and calculated the full-width at half-maximum. The peak area was integrated using the trapezoid rule and divided by the entire PFDN2 or UXT signal across the entire fractionation dimension. For all stoichiometry calculations, subunits in each complex were selected and the full-width at half-maximum was calculated. Then, the protein with the lowest area was used as the stoichiometric unit. Each replicate was processed individually, and the barplot shows the data from all DIP-MS experiments ($n = 3$ biologically independent experiments for PFDN2 and UXT). For the Prefoldin stoichiometry calculations, only PFDN2-DIP-MS data were considered since UXT is not part of the canonical PFD complex.

### Sequence alignment and prediction of IDRs

Sequence alignment of PFDN4 and PDRG1 was performed on canonical FASTA sequences obtained from UniProtKB (3 July 2022) with Clustal Omega (EBI, v.2.1)[82] using default parameters (Supplementary Data 4 for identity matrix). For visualization, Jalview (v.2.11.2.0)[83] was used (Supplementary Data 5). For prediction of intrinsic disordered regions of URI1, the tool flDPnn[84] under http://biomine.cs.vcu.edu/servers/flDPnn/ was used, applying default parameters (26 June 2022). Outputs were limited to the relevant data containing predictions for disordered regions and protein binding interface.

### Reagent and software tool resources

A list of all materials and software tools used are detailed in Supplementary Table 6, including company names and catalog numbers of commercial reagents.

### Reporting summary

Further information on research design is available in the Nature Portfolio Reporting Summary linked to this article.

## Data availability

The mass spectrometry proteomics data, and Spectronaut, Skyline and MaxQuant outputs have been deposited to the ProteomeXchange Consortium via the PRIDE partner repository (https://www.ebi.ac.uk/pride/archive/projects/PXD035032/) (ref. 85). Human protein fasta files have been retrieved from UniProtKB (Taxonomic identifier 9606, status reviewed, downloaded on 1 December 2019, https://www.uniprot.org/) and are deposited alongside the MS data. The ColabFold (v.1.3.0) predicted structural models, coelution data and PPIprophet parameters are deposited on Github https://github.com/anfoss/DIP-MS_data (ref. 86). PDB entries 2XSZ (https://doi.org/10.2210/pdb6NRD/pdb)[87] and 6NRD (https://doi.org/10.2210/pdb6NRD/pdb)[88] are accessible via https://www.rcsb.org/. Source data are provided with this paper.

## Code availability

PPIprophet is available freely for academic use under MIT license on GitHub at https://github.com/anfoss/PPIprophet. PPIprophet is deposited for review purposes under CodeOcean capsule with the link https://codeocean.com/capsule/2117766.

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

## Acknowledgements

Figure 1 was created with BioRender.com. F.F., F.U. and M.G. were supported by the Innovative Medicines Initiative project ULTRA-DD (FP07/2007-2013, grant no. 115766). M.G. has received support from the EU/EFPIA/OICR/McGill/KTH/Diamond Innovative Medicines Initiative 2 Joint Undertaking (EUbOPEN grant no. 875510). The funders had no role in study design, data collection and analysis, decision to publish or preparation of the manuscript.

## Author contributions

F.F. and M.G. conceived and designed the project. F.F. performed the experiments. M.H., F.U. and B.W. provided critical input to experimental design and biochemical protocols. F.F., A.F., P.X. and F.W. acquired the mass spectrometry data. F.F. and A.F. conducted the data analysis and A.F. developed the software tool for data analysis. F.F. and A.F. generated the figures and wrote the original draft supervised by M.G., R.C. and R.A. All coauthors contributed in reviewing and editing the manuscript. M.G. and R.A. provided funding and resources to support the project.

## Funding

## Competing interests

M.H. was an employee of EVOSEP. The remaining authors declare no competing interests.

## Additional information

**Extended data** is available for this paper at https://doi.org/10.1038/s41592-024-02211-y.

**Correspondence and requests for materials** should be addressed to Fabian Frommelt or Matthias Gstaiger.

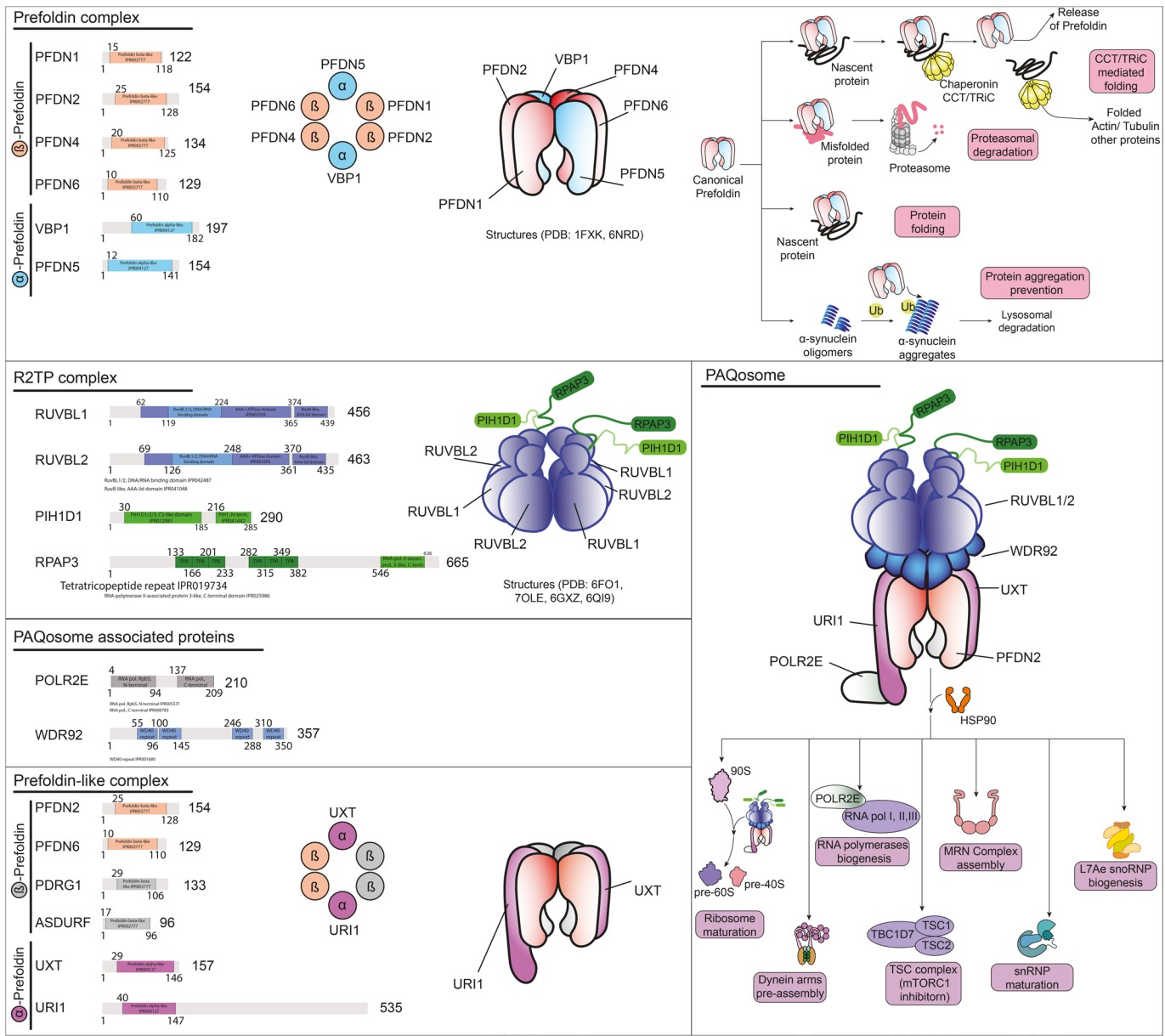

**Extended Data Fig. 1 | Schematic representation of composition of canonical Prefoldin complex and Prefoldin-like/URI protein complexes.** Prefoldin and prefoldin-like protein complexes are reported to be heterohexameric complexes consisting of two α-subunits and four β-subunits. Canonical prefoldin consist of the α-subunits PFDN5 and VBP1 (blue) and the β-subunits PFDN1, PFDN2, PFDN4 and PFDN6 (red). The PFDL complex consists of the shared β-subunits of PFDN2 and PFDN6, the β-subunits of PDRG1, ASDURF (gray) and the α-subunits of UXT URI1 (violet). The R2TP complex, comprises a heterohexameric RUVBL1/RUBL2 ring and one or multiple copies of the adaptor/regulatory subunits RPAP3 and PIH1D1. The R2TP and PFDL-complex are part of the larger PAQosome assembly, which contains two additional subunits, the URI1 associated POLR2E and the WD40 repeat containing protein WDR92, which is likely associated with the R2TP complex. Protein domains were obtained from InterPro. In addition to a sketch of the complexes, the major biological functionalities of canonical Prefoldin and the PAQosome are sketched (see also Lynham et al.[89] for a more detailed review of the PAQosome and PFD complex). The graphical representation of Prefoldin, PFDL and PAQosome components were inspired by graphical representations of Marie-Soleil Gauthier, Philippe Cloutier and Benoit Coulombe.

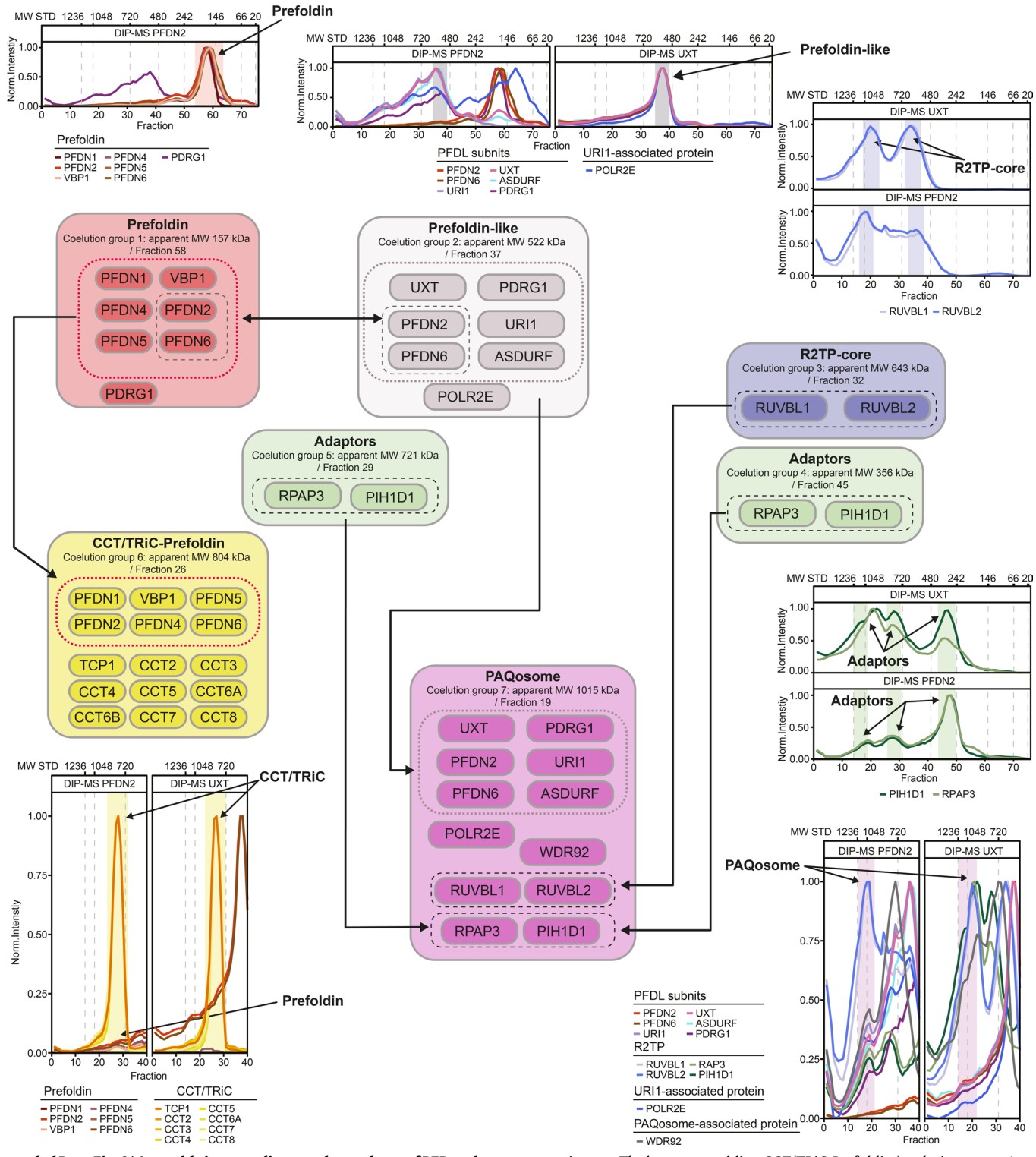

**Extended Data Fig. 2 | Assembly intermediates and complexes of PFD and PFDL subunits identified across DIP–MS experiments.** Across the PFDN2 (n = 3 biologically independent experiments) and UXT (biological replicates: n = 3 biologically independent experiments) DIP–MS assemblies we characterized 7 distinct coelution peak groups of PFD, PFDL, R2TP and CCT/TRiC core-subunits. The canonical PFD complex and PDRG1 eluted at Fraction 58 with an apparent MW of 157 kDa (coelution group 1) and the PFDL-module eluted at Fraction 37 with an apparent MW of 522 kDa (coelution group 2). The R2TP subassembly (coelution group 3 or Fraction 32 apparent MW 643 kDa) and the two adaptor peak subassemblies (apparent MW 356 kDa and 721 kDa or Fraction 45 and 29 assigned as coelution group 4 and 5) were recovered within both DIP-MS

experiments. The larger assemblies, CCT/TRiC-Prefoldin (coelution group 6, apparent MW 804 kDa or Fraction 26) and the PAQosome (coelution group 7, apparent MW of 1015 kDa Fraction 19) are indicated at the bottom. Shared subunits and shared complex modules are indicated by arrows and boxes. Below the subassembly, the apparent MW derived from the separation is indicated. Sub-complex stoichiometry was inferred from literature reported assemblies. Apparent MW calibration curve data is in Supplementary Data 1. Within the DIP-MS of UXT the PFD and PFD-CCT/TRiC was not recovered (only residual CCT/TRiC from background). For the coelution groups profile plots, the y-axis represents the average MS2 protein intensities normalized by the maximum protein intensity across the gel slices (x-axis).

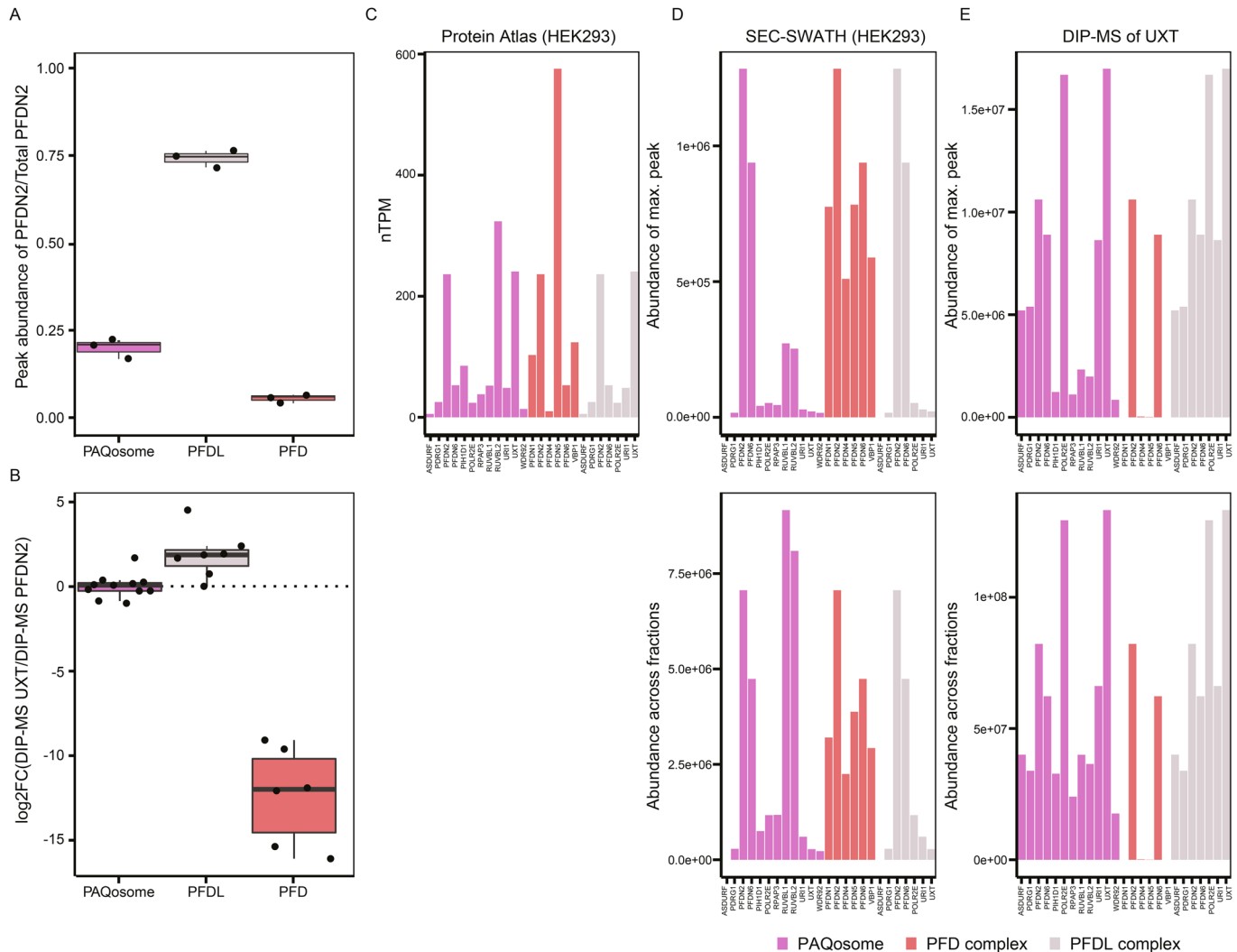

**Extended Data Fig. 3 | Enrichment of PAQosome, PFDL and PFD in DIP-MS of UXT and PFDN2. a**. Relative stoichiometry of PFDN2 for the UXT DIP-MS experiment Boxplot showing the ratio of PFDN2 signal in the three major assemblies PFD, PFDL and PAQosome versus the total PFDN2 signal. Box represents the interquartile range and its whiskers 1.5 X IQR. Black line represents the median and the replicates are shown with black dots (n = 3 biologically independent experiments). **b**. Log$_2$FC of enrichment of PAQosome, PFDL and canonical PFD complexes in the UXT DIP-MS (n = 3 biologically independent experiments) against the PFDN2 DIP-MS (n = 3 biologically independent experiments). The average across the replicates at the peak group fractions of the complexes was used as proxy for the abundance of the complexes. For each subunit within each peak group the log$_2$FC of the MS2 protein intensity in

the UXT against the PFDN2 DIP-MS experiment was calculated. Box represents the interquartile range and its whiskers 1.5 X IQR. Black line represents the median and the black dots represent the average log$_2$FC across the replicates for n = 11 PAQosome subunits, n = 7 PFDL subunits and n = 6 canonical PFD complex subunits at the complex peak. **c**–**e**. Abundance comparison of PFD, PFDL and PAQosome core-subunits within transcriptomics, quantitative SEC-MS and DIP-MS. **c**. Normalized transcripts per million (nTPM) of core-subunits for the PAQosome (pink), PFD (red) and PFDL (gray). **d**. Peak intensity and summed abundance across all fractions for each subunit in a SEC-SWATH (n = 1) experiment and **e**. for DIP-MS of UXT (n = 3 biologically independent experiments). For the DIP-MS the average across all three replicates per experiment was considered.

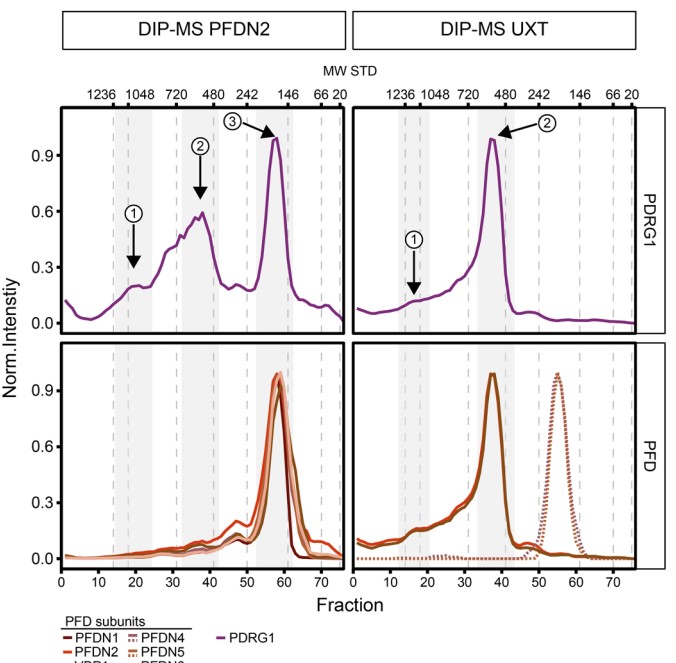

**Extended Data Fig. 4 | PDRG1 coelution profile in the PFDN2 and UXT DIP-MS experiments.** PDRG1 shows three distinct peaks (labelled 1,2,3) in the DIP-MS experiments of PFDN2, whereas in the UXT DIP-MS only two peaks (1,2) are identified. In both DIP-MS experiments the two peaks at the higher MW belong to the PFDL complex (2) and the PFDL containing PAQosome complex (1). In the PFDN2 DIP-MS the third (3) and most intense PDRG1 peak coelutes with canonical PFD. The intensity shows the smoothed average MS2 protein intensity normalized to the maximum protein intensity across all gel-slices. PFDN4 and PFDN5 coeluting at the PFD peak (dashed lines in the UXT DIP-MS) are contaminant signal in the UXT DIP-MS experiment. The signal is of PFDN4 and PFDN5 is three orders of magnitude lower compared to the PFDN2 signal at the PFDL complex.

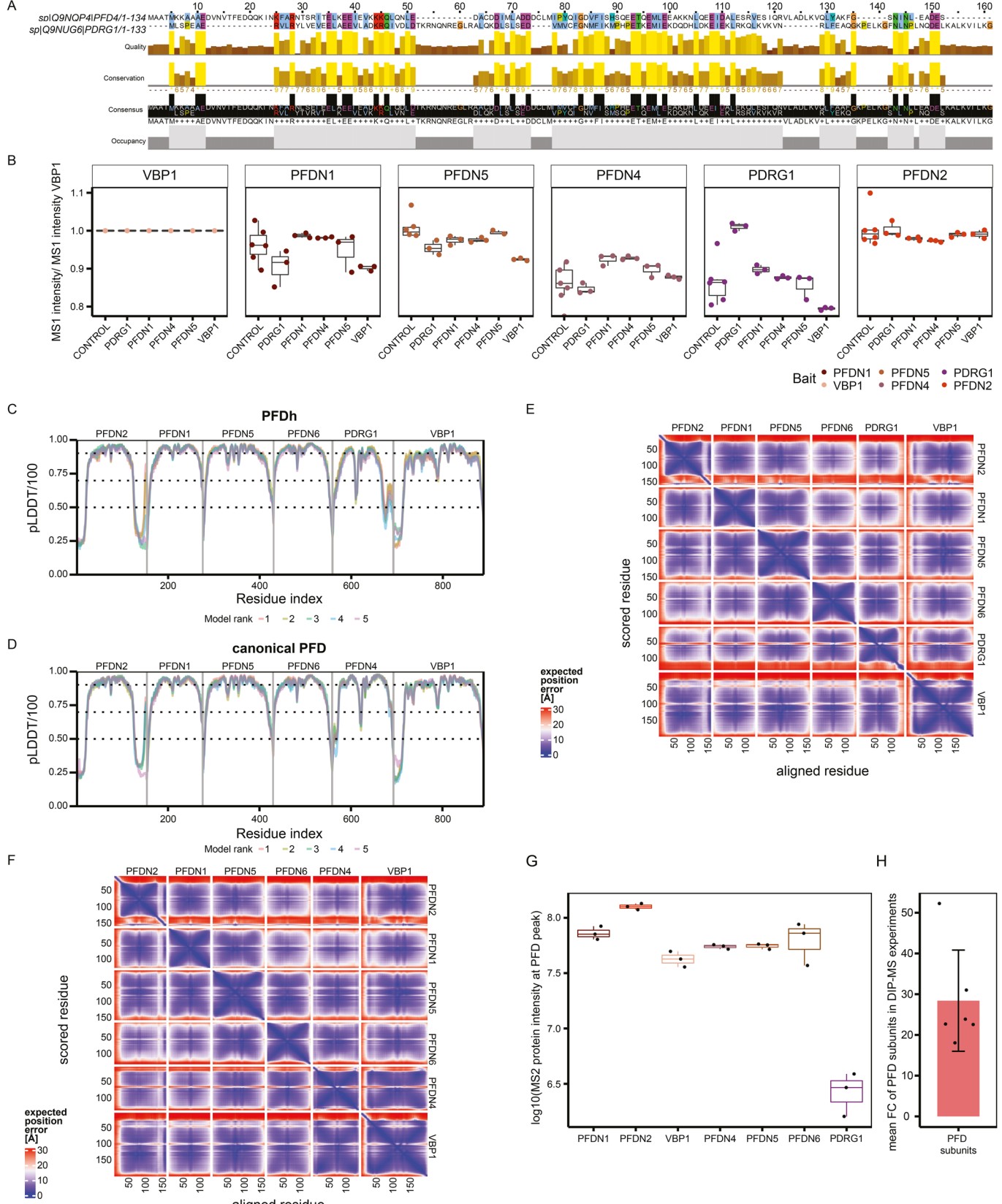

**Extended Data Fig. 5 | See next page for caption.**

**Extended Data Fig. 5 | Identification of PDRG1 as subunit of the alternative PFD homolog complex. a**. Sequence alignment of PDRG1 with PFDN4 indicates large stretches of consensus motifs within both proteins, indicating that PFDN4 is the subunit replaced by PDRG1 in the PFD homolog complex. **b**. Ratio to VBP1 comparison on MS1-intensity protein abundance levels in validation AP-MS experiments across different canonical subunits used as baits showing that PDRG1 ratios were lowest in the AP-MS using PFDN4 as bait, indicative that the recovered PDRG1 interaction with PFDN4 is at the noise level comparable to the level of PFDN4 signal in GFP negative controls. For canonical prefoldin subunits baits: n = 3 biologically independent replicates, Control: n = 6 biologically independent experiments. Solid line represents the median, box limits show the IQR and its whiskers 1.5 x IQR. **c**. AlphaFold2 confidence in pLDDT (predicted Local Distance Difference Test) divided by 100 for 5 models of the PFD homolog complex. **d**. AlphaFold2 confidence in pLDDT/100 for 5 models of the canonical PFD complex. **e**. Inter PAE (Predicted Aligned Error) heatmap for the best ranked model of PFDh complex. **f**. Inter PAE (Predicted Aligned Error) heatmap for the best ranked model of the PFD complex. **g**. Comparison of $\log_{10}$ MS2 protein abundance of canonical PFD subunits and PDRG1 in the PFD coelution peak. Dots represent signals from the replicates of the PFDN2 DIP-MS experiments (n = 3 biologically independent experiments). Solid line represents the median, box limits show the IQR and its whiskers 1.5 x IQR. **h**. Mean FC of each PFD subunit (n = 6) against PDRG1. The mean FC was derived from the MS2 protein abundance of canonical PFD and PDRG1 in the PFD coelution peak for each PFDN2 DIP-MS replicate (n = 3 biologically independent replicate). Each point represents the average FC of a PFD subunits in the PFD-peak against PDRG1. Data are presented as mean values +/- SD.

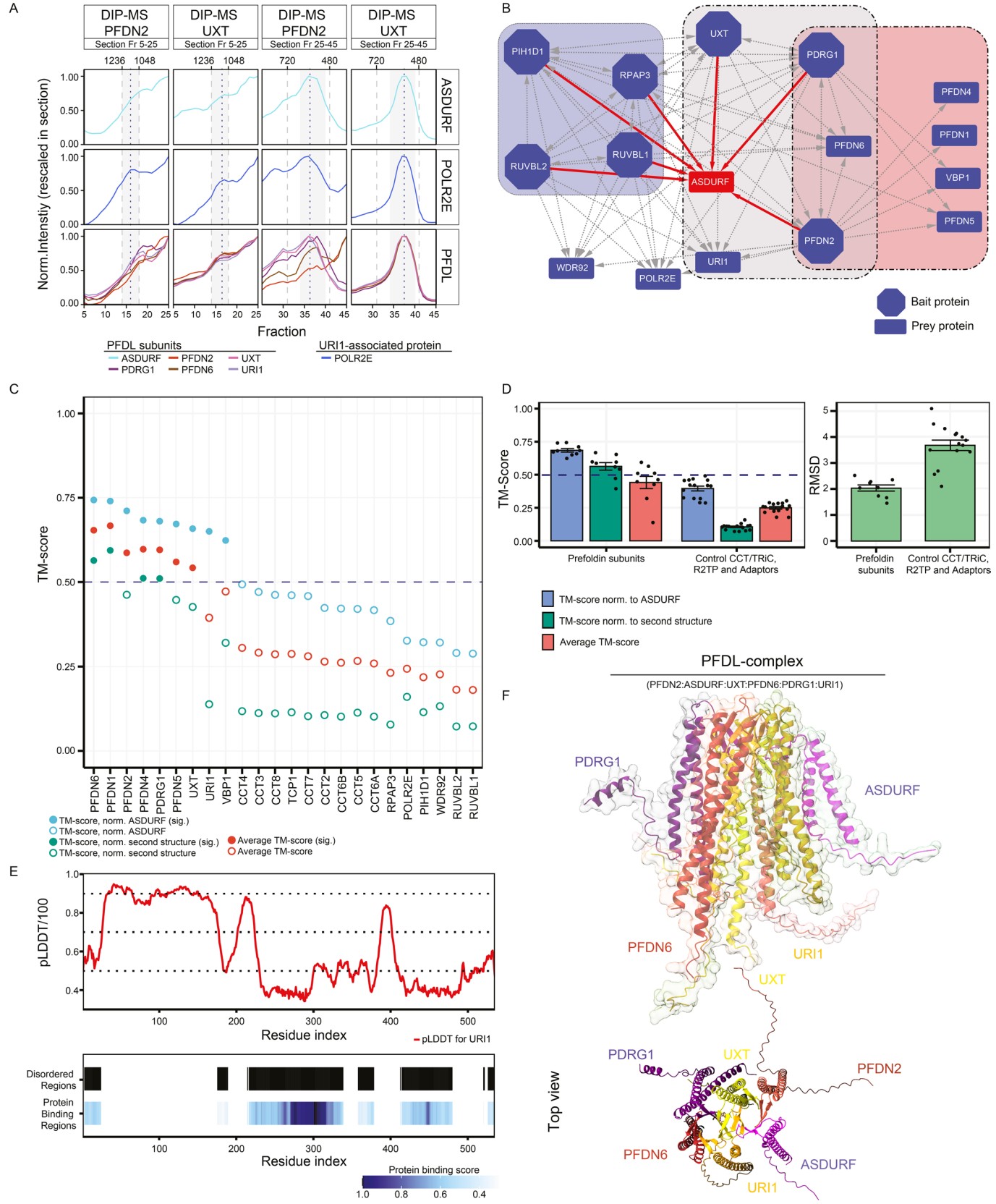

**Extended Data Fig. 6 | See next page for caption.**

**Extended Data Fig. 6 | Identification of ASDURF and POLR2E as constitutive PFDL subunits. a.** Coelution of ASDURF with the PFDL and the PFDL containing PAQosome complex within the PFDN2 and UXT DIP-MS experiments. The MS2 protein intensity was rescaled within the sections (PAQosome section: Fraction 5 – 25 and PFDL section 25 – 45) due to high signal differences between the PFDL and PAQosome complexes. **b.** Interactome derived from AP-MS of core-subunits of the R2TP complex (blue background), PFDL (gray background), and canonical prefoldin (red background). Interactions are filtered only for high-confidence (Log$_2$FC ≥ 5 and Saint score ≥0.99) and showing only interactions between core-subunits. Baits are depicted as octagons, and red edges represent interactions with ASDURF (shown in red). ASDURF was only recovered with subunits of the PFDL and PFDL containing PAQosome complexes. **c.** Structural alignment of ASDURF AlphaFold model to canonical PFD, PFDL, and CCT/TRiC subunits. The TM-score was either normalized to ASDURF (blue circles) or the second structure (green circles). The mean of the two normalized TM-score is reported (red circles). A significant threshold of 0.5 TM-score was applied (dotted line). Filled circles indicate high structural similarity, empty circles indicate lower to no structural similarity. **d.** TM-score and RMSD for Prefoldin subunits (n = 9) and the negative control (subunits of CCT/TRiC, R2TP and adaptor proteins, n = 15). The TM-score for prefoldin subunits indicate strong structural similarities between all PFD subunits and ASDURF. The RMSD of the alignment (2 Å) indicates that the predicted ASDURF model structure shares strong similarities with PFD and PFDL subunits. The structural alignment was performed once for each subunit (n = 1). Data are presented as mean values +/- SEM. **e.** AlphaFold2 pLDDT score divided by 100 of URI1 (AF-O94763-F1) plot by residue. Very low pLDDT/100 score (<0.4) indicate IDRs. At the bottom prediction of IDRs and protein binding regions employing flDPnn indicate large regions IDR within URI1 which explains the poor predictions of these regions in the AlphaFold2 model. **f.** Structural model of the PFDL complex. Subunits are colored. Upper part shows a side view of the complex, the lower part shows a top view, showing the stacked β-sheets.

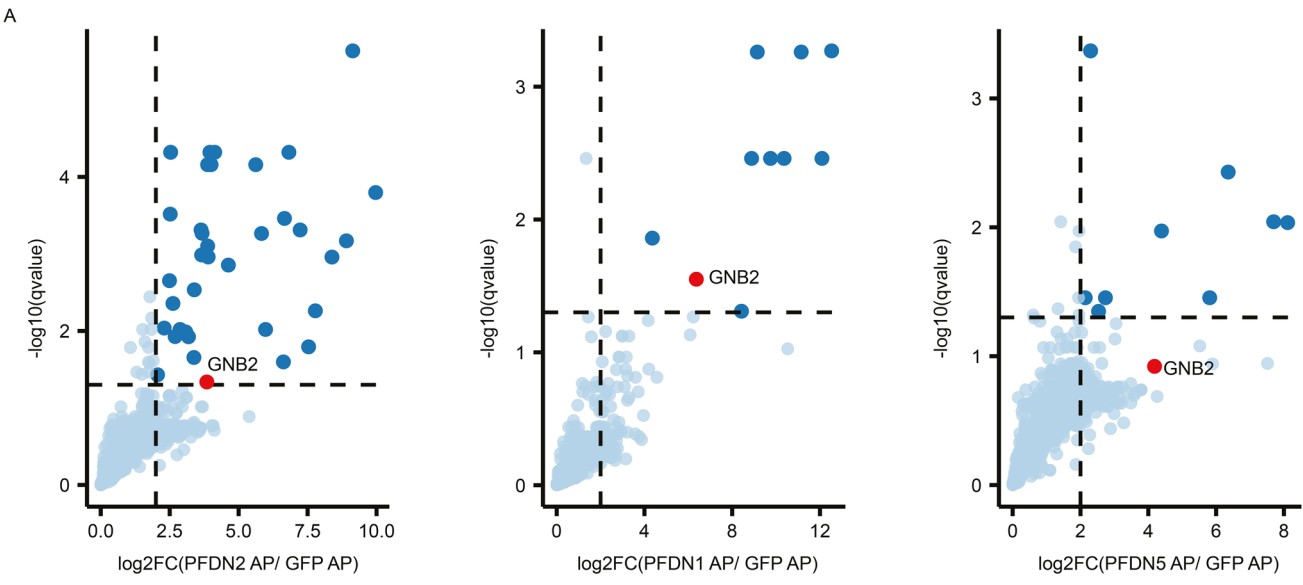

**Extended Data Fig. 7 | Validation of GNB2 as PFD subunit by reciprocal AP-MS. a.** Volcano plot for exclusive PFD subunits (PFDN1, PFDN5) and PFDN2, highlighting the recovery of GNB2 as enriched protein versus GFP purification. Dotted lines correspond to a Log$_2$FC = 2 and a q-value = 5%. **b.** Subnetwork of proteins interacting with GNB2 (red octagon) extracted from BioPlex 3.0 (for HEK293T cells). The reported interactions cover protein-protein interactions between GNB2 and its interactors. The network supports our findings, that GNB2 is interacting with Prefoldin subunits (red) and CCT/TRiC subunits (yellow). The direction of interaction is indicated by arrows, and hexagons indicates bait proteins within BioPlex 3.0.

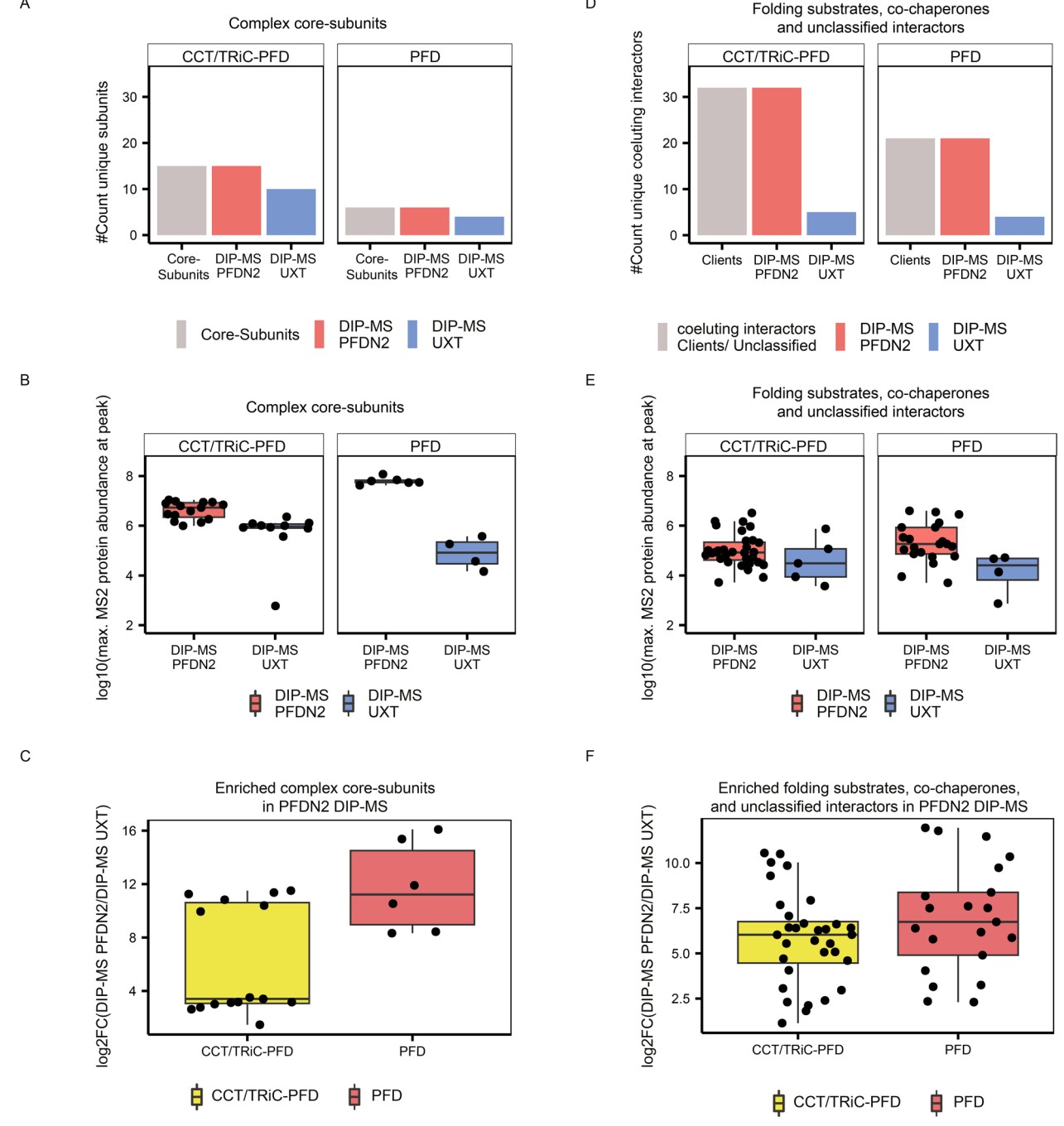

**Extended Data Fig. 8 | Quantitative comparison of core-subunits, substrates, adaptors and unclassified proteins coeluting with CCT/TRiC-PFD and PFD complexes in the PFDN2 and UXT DIP-MS experiments. a**. Recovery of CCT/TRiC-PFD and PFD subunits in the DIP-MS experiments of PFDN2 and UXT. **b**. Boxplot of the $\log_{10}$ quantitative protein abundance of core-subunits (max. MS2 signal at the CCT/TRiC-PFD or PFD peaks) in the PFDN2 (n = 3 biologically independent experiments) and UXT DIP-MS (n = 3 biologically independent experiments) experiments. The low abundance of canonical PFD complex in the UXT DIP-MS, indicates that the PFD complex is a background contaminant within this experiment. The solid line represents the median, box limits show the IQR and its whiskers 1.5 x IQR. The dots show the mean signal across DIP-MS replicates for each core-subunit at the coelution peak. The number of dots varies due to missing values. **c**. Boxplot of $\log_2$FC of core-subunits quantified in the PFDN2 DIP-MS (n = 3 biologically independent experiments) against UXT DIP-MS (n = 3 biologically independent experiments) on protein abundance level. The solid line represents the median, box limits show the IQR

and its whiskers 1.5 x IQR. Dots represent the mean $\log_2$FC across all DIP-MS replicates for each core-subunit split by coelution group (CCT/TRiC-PFD: n = 15, and PFD n = 6). **d**. Recovery of CCT/TRiC-PFD and PFD coeluting proteins in the DIP-MS experiments of PFDN2 and UXT. **e**. Boxplot of $\log_{10}$ quantitative protein abundance of coeluting proteins (max. MS2 signal at the CCT/TRiC-PFD or PFD peaks) in the PFDN2 (n = 3 biologically independent experiments) and UXT DIP-MS experiments (n = 3 biologically independent experiments). The solid line represents the median, box limits show the IQR and its whiskers 1.5 x IQR. The dots show the mean signal across DIP-MS replicates for each protein at the coelution peak. The number of dots varies due to missing values. **f**. Boxplot of $\log_2$FC of coeluting proteins quantified in the PFDN2 DIP-MS (n = 3 biologically independent experiments) against UXT DIP-MS (n = 3 biologically independent experiments) on protein abundance level. The solid line represents the median, box limits show the IQR and its whiskers 1.5 x IQR. Dots represent the mean $\log_2$FC across all DIP-MS replicates for each coeluting protein split by coelution group (CCT/TRiC-PFD: n = 32, and PFD n = 21 coeluting proteins).

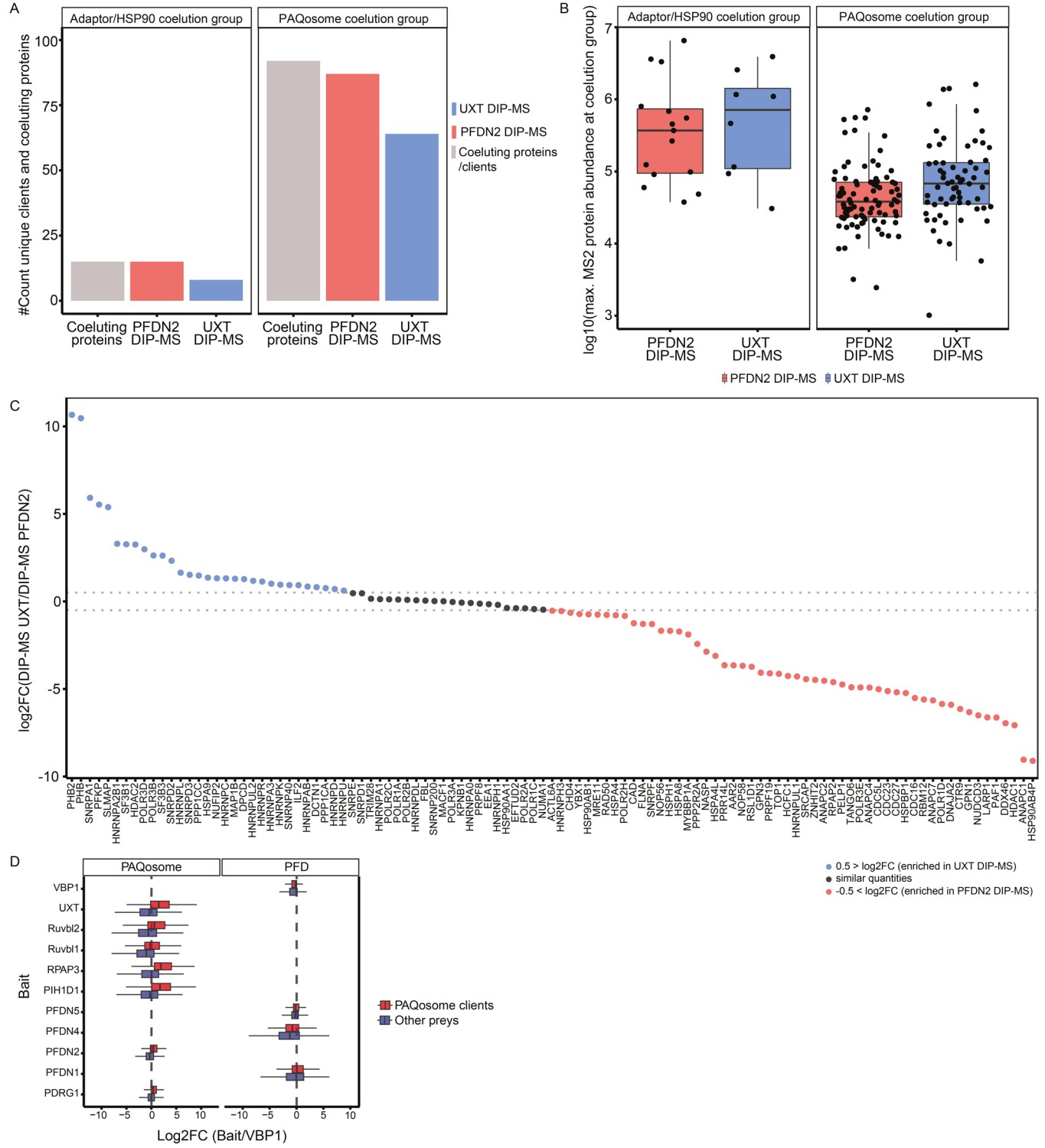

**Extended Data Fig. 9 | See next page for caption.**

**Extended Data Fig. 9 | Quantitative comparison of client complex subunits and proteins coeluting with the PAQosome and the Adaptor/HSP90 assemblies in the PFDN2 and UXT DIP-MS experiments. a**. Recovery of client complex subunits and coeluting proteins in the DIP-MS experiments, separated by the two coelution groups of Adaptor/HSP90 (red) and PAQosome (blue) coelution groups. The combined number of coeluting proteins and clients is reported in gray. **b**. Comparison of coeluting protein and client MS2 protein abundance, averaged maximum intensity across the DIP-MS experiments for PFDN2 (n = 3 biologically independent experiments) and UXT (n = 3 biologically independent experiments) at the coelution groups of the Adaptor/HSP90 (red) and PAQosome (blue) coelution groups. Solid line represents the median, box limits show the IQR and its whiskers 1.5 x IQR. Coelution groups contain the following number of proteins: Adaptor/HSP90 coelution group for PFDN2 DIP-MS n = 15, for UXT DIP-MS n = 8, PAQosome coelution group for PFDN2 DIP-MS n = 87, for UXT DIP-MS n = 64. **c**. Log$_2$FC of coeluting proteins/clients

abundance in UXT DIP-MS compared to the PFDN2 DIP-MS experiment. Missing values were imputed by 1e$^3$ to derive log$_2$FC (indicated in red in the Source Data). Values are ordered from largest to smallest log$_2$FC. Proteins recovered with higher signal in UXT DIP-MS (log2FC > 0.5) are reported in blue dots, whereas proteins quantified higher in the PFDN2 DIP-MS experiment (log$_2$FC < −0.5) are red dots. A group of slightly to unchanged coeluting unclassified protein and clients (log$_2$FC < 0.5 and > −0.5) are reported in black dots. **d**. Boxplot showing the enrichment of PAQosome coeluting proteins/clients (red box) versus the other identified proteins (blue box) across AP-MS. X axis represents the log$_2$FC calculated across bait proteins (Y axis) versus the corresponding protein abundance in VBP1, used here as representative PFD exclusive subunit. Different columns show PAQosome core components or PFD subunits. Solid line represents the median, box limits show the IQR and its whiskers 1.5 x IQR (n = 3 biologically independent replicates per bait).

# Reporting Summary

## Statistics

For all statistical analyses, confirm that the following items are present in the figure legend, table legend, main text, or Methods section.

| n/a | Confirmed | |
|---|---|---|
| ☐ | ☒ | The exact sample size (*n*) for each experimental group/condition, given as a discrete number and unit of measurement |
| ☐ | ☒ | A statement on whether measurements were taken from distinct samples or whether the same sample was measured repeatedly |
| ☐ | ☒ | The statistical test(s) used AND whether they are one- or two-sided<br>*Only common tests should be described solely by name; describe more complex techniques in the Methods section.* |
| ☒ | ☐ | A description of all covariates tested |
| ☐ | ☒ | A description of any assumptions or corrections, such as tests of normality and adjustment for multiple comparisons |
| ☐ | ☒ | A full description of the statistical parameters including central tendency (e.g. means) or other basic estimates (e.g. regression coefficient) AND variation (e.g. standard deviation) or associated estimates of uncertainty (e.g. confidence intervals) |
| ☐ | ☒ | For null hypothesis testing, the test statistic (e.g. *F*, *t*, *r*) with confidence intervals, effect sizes, degrees of freedom and *P* value noted<br>*Give P values as exact values whenever suitable.* |
| ☒ | ☐ | For Bayesian analysis, information on the choice of priors and Markov chain Monte Carlo settings |
| ☒ | ☐ | For hierarchical and complex designs, identification of the appropriate level for tests and full reporting of outcomes |
| ☐ | ☒ | Estimates of effect sizes (e.g. Cohen's *d*, Pearson's *r*), indicating how they were calculated |

*Our web collection on statistics for biologists contains articles on many of the points above.*

## Software and code

Policy information about availability of computer code

| Data collection | Fusion FX6edge (VILBER Lourmat) version 18.02-SN<br>MaxQuant (Max-Planck-Institute for biochemistry) version 1.5.2.8<br>Skyline (MacCoss Lab Software) version 20.1.0.76<br>Spectronaut (Biognosys) version 13.12.200217.43655, Laika) |
|---|---|
| Data analysis | R (The R foundation) version R-3.6.1<br>Python (v3.7.2)<br>PPIprophet (https://github.com/anfoss/PPIprophet) v1<br>Clustal Omega (https://www.ebi.ac.uk/Tools/msa/clustalo/) v2.1<br>ColabFold (https://colab.research.google.com/github/sokrypton/ ColabFold/blob/v1.3.0/AlphaFold2.ipynb) version 1.3.0<br>Jalview https://www.jalview.org/ version 2.11.2.0<br>Cytoscape Version: 3.8.2<br>US-align (Zhang group; https://zhanggroup.org/US-align/) Version 20220511<br>flDPnn (http://biomine.cs.vcu.edu/servers/flDPnn/) Version: December2021<br>UCSF ChimeraX: Structure visualization for researchers, educators, and developers. (UCSF) version 1.4<br>PPIprophet was developed for this study and is freely accessible under https://github.com/anfoss/PPIprophet . |

For manuscripts utilizing custom algorithms or software that are central to the research but not yet described in published literature, software must be made available to editors and reviewers. We strongly encourage code deposition in a community repository (e.g. GitHub). See the Nature Portfolio guidelines for submitting code & software for further information.

## Data

Policy information about availability of data

All manuscripts must include a data availability statement. This statement should provide the following information, where applicable:
- Accession codes, unique identifiers, or web links for publicly available datasets
- A description of any restrictions on data availability
- For clinical datasets or third party data, please ensure that the statement adheres to our policy

The mass spectrometry proteomics data and Spectronaut, Skyline and MaxQuant outputs have been deposited to the ProteomeXchange Consortium via the PRIDE partner repository with the dataset identifier PXD035032. Human protein fasta files have been retrieved from UniProtKB (Taxonomic identifier 9606, status reviewed, downloaded on the 01.12.2019, https://www.uniprot.org/) and is deposited alongside the MS data. The with ColabFold (version 1.3.0.) predicted structural models, coelution data and PPIprophet parameters are deposited on Github https://github.com/anfoss/DIP-MS_data. PDB entries 2XSZ and 6NRD are accessible via https://www.rcsb.org/.

## Human research participants

Policy information about studies involving human research participants and Sex and Gender in Research.

| Reporting on sex and gender | n/a |
| --- | --- |
| Population characteristics | n/a |
| Recruitment | n/a |
| Ethics oversight | n/a |

Note that full information on the approval of the study protocol must also be provided in the manuscript.

# Field-specific reporting

Please select the one below that is the best fit for your research. If you are not sure, read the appropriate sections before making your selection.

☒ Life sciences   ☐ Behavioural & social sciences   ☐ Ecological, evolutionary & environmental sciences

For a reference copy of the document with all sections, see nature.com/documents/nr-reporting-summary-flat.pdf

# Life sciences study design

All studies must disclose on these points even when the disclosure is negative.

| Sample size | No prior sample size calculation was performed. Triplicate experiments were performed as routine in co-fractionation MS experiments. |
| --- | --- |
| Data exclusions | The WDR92-SH tagged AP-MS was excluded from the analysis due to low expression levels of the bait protein, resulting in low abundance and low sequence coverage. |
| Replication | All DIP-MS experiments (native co-fractionation dataset) were conducted in biological triplicates and the replications were successful. For all reciprocal AP-MS biological triplicates were performed, and all replicates were successful. As mentioned in the data exclusion section the AP-MS results of WDR92-SH were excluded due to low abundance and sequence coverage of the bait protein. For the estimation of the absolute amount of PFDN2 and UXT in the samples, external calibration curve and AP-inputs were injected once. Sample preparation optimization for different filter plates were either performed in triplicates or duplicates. DIA-method optimization was performed in triplicates. |
| Randomization | For sample preparation of PFDN2 and UXT DIP-MS fractions we optimized a 96-well format procedure with a randomization scheme. All samples have been LC-MS measured in sequential order for all experiments, including also the for the sample preparation randomized DIP-MS experiments. For AP-MS samples, no randomization was performed as standard in this type of experiments. |
| Blinding | Blinding was not performed due to the need to analyze samples sequentially and the randomization already performed at the sample preparation stage. |

# Reporting for specific materials, systems and methods

We require information from authors about some types of materials, experimental systems and methods used in many studies. Here, indicate whether each material, system or method listed is relevant to your study. If you are not sure if a list item applies to your research, read the appropriate section before selecting a response.

## Materials & experimental systems

| n/a | Involved in the study |
|-----|----------------------|
| ☒ ☐ | Antibodies |
| ☐ ☒ | Eukaryotic cell lines |
| ☒ ☐ | Palaeontology and archaeology |
| ☒ ☐ | Animals and other organisms |
| ☒ ☐ | Clinical data |
| ☒ ☐ | Dual use research of concern |

## Methods

| n/a | Involved in the study |
|-----|----------------------|
| ☒ ☐ | ChIP-seq |
| ☒ ☐ | Flow cytometry |
| ☒ ☐ | MRI-based neuroimaging |

# Eukaryotic cell lines

Policy information about cell lines and Sex and Gender in Research

| | |
|---|---|
| Cell line source(s) | HEK293 WT cell line (Thermo Fisher Scientific), (Invitrogen) R70507<br>For cell line generation of the strep-HA expressing bait proteins the Flp-In HEK293 T-REx cells line (Thermo Fisher Scientific) (Invitrogen) R78007 were used. |
| Authentication | The cell lines were not authenticated. |
| Mycoplasma contamination | The cell lines were not tested for mycoplasma contamination. |
| Commonly misidentified lines<br>(See ICLAC register) | No commonly misidentified cell lines were used in this study. |

