## [Peer Review File · Nature Methods]

Peer Review Information

Manuscript Title: DIP-MS: A novel ultra-deep interaction proteomics for the deconvolution of protein complexes

Corresponding author name(s): Matthias Gstaiger, Fabian Frommelt

Editorial Notes: None

Reviewer Comments & Decisions:

Decision Letter, initial version:

Dear Fabian,

Your Article entitled "DIP-MS: A novel ultra-deep interaction proteomics for the deconvolution of protein complexes" has now been seen by 2 reviewers, whose comments are attached. In the light of their advice we have decided that we cannot offer to publish your manuscript in Nature Methods.

You will see that, while one referee is very positive about the impact of the work, the other referee finds the work to have limited methodological advance. When we review papers we look for both a lack of technical criticisms as well as enthusiasm that the method will enable new biological discovery and be of interest to a broad audience. Based on the feedback we received and our editorial discussion, we do not think the paper is appropriate for Nature Methods. However, we think the work could be presented as a results forward paper and published at another appropriate journal.

You may want to consider *Nature Communications* as a potential venue for publication of your manuscript. I see that you have opted out of transfer consultations. However, if you wish, I would be happy to consult with my colleagues at Nature Communications about the possibility of a transfer. *Nature Communications* publishes high quality and influential research across the full spectrum of the natural sciences. More information on the journal, the potential benefits of transfer, and a link to transfer your paper, can be found below. Please note that the editorial team at Nature Communications will consider your manuscript independently of our suggestion to transfer.

I am sorry that we cannot be more positive on this occasion but hope that you find the reviewers' comments helpful when preparing your paper for submission elsewhere.

Sincerely,
Arunima

Arunima Singh, Ph.D.

Senior Editor
Nature Methods

** *Nature Communications* is the Nature Portfolio flagship Open Access journal. If you would like this work to be considered for publication there, you can easily transfer the manuscript by following the instructions below. It is not necessary to reformat your paper. Once all files are received, the editors at *Nature Communications* will assess your manuscript's suitability for potential publication; they aim to provide feedback quickly, with a median decision time of 8 days for first editorial decisions on suitability. If your paper has been peer reviewed at this journal, the referee reports will also be transferred and assessed by the editorial team. In some cases, papers are accepted without further peer review, providing a rapid path to publication. Nature Communications also offers double blind and transparent peer review options. Please note manuscripts accepted for publication at this journal are subject to an article-processing charge.

For journal metrics, please visit our http://www.nature.com/npg_/company_info/journal_metrics.html (Nature journals metrics page). Our http://www.nature.com/ncomms/open_access/index.html open access pages contain information about article processing charges, open access funding, and advice and support from Springer Nature.

** I suggest that you consider Nature Communications as a suitable venue for your work. To transfer your manuscript there, please use our <https://mts-nmeth.nature.com/cgi-bin/main.plex?el=A4M5WtQ4A1Bdjv1X4A9ftdkGEyslUnoGv1achNNJYGRAZ> manuscript transfer portal. You will not have to re-supply manuscript metadata and files, unless you wish to make modifications, but please note that this link can only be used once and remains active until used. For more information, please see our http://www.nature.com/authors/author_resources/transfer_manuscripts.html?WT.mc_id=EMI_NPG_1511_AUTHORTRANSF&WT.ec_id=AUTHOR manuscript transfer FAQ page.

Note that any decision to opt in to In Review at the original journal is not sent to the receiving journal on transfer. You can opt in to <https://www.nature.com/nature-portfolio/for-authors/in-review> In Review at receiving journals that support this service by choosing to modify your manuscript on transfer. In Review is available for primary research manuscript types only.

Reviewer Comments:

Reviewer #1 (Remarks to the Author):

Protein affinity purification coupled with mass spectrometry (AP-MS) has been used by numerous laboratories to characterize protein complexes. Because a given sample can contain a diversity of complex isoforms (with some varying subunits), the characterization of these complexes usually require reciprocal tagging of several subunits in AP-MS. In this article, Frommelt and colleagues introduce Deep Interactome Profiling by Mass Spectrometry (DIP-MS), a method combining affinity

purification with native-fractionation by PAGE and data independent acquisition mass spectrometry (DIA-MS) and a data analysis software package to resolve complex isoforms sharing the same bait protein in a single experiment. According to the authors, DIP-MS provides three major improvements: (1) a novel miniaturized sample preparation procedure in a filter plate format that requires 10 times less material than traditional chromatography-based separation and achieves high reproducibility; (2) a DIA-MS acquisition scheme coupled with a 24 minutes injection to injection gradient chromatography system that allows for measuring of up to 60 samples per day, thus increasing sample throughput compared to prior methods; (3) a deep-learning based framework trained on more than 1.5 million binary interactions from various datasets, which enabled prediction of protein-protein interactions (PPIs), identification of multiple instances of protein complexes, and ultimately the reliable deconvolution of complex profiling data into functional modules. This new method represents the latest evolution of the AP-MS procedure, with unprecedented resolution and sensitivity.

DIP-MS was tested using the prefoldins (PFD), a group of chaperones that have previously been shown to be components of various complexes in human cells: (1) the canonical PFD complex (2 alpha and 4 beta subunits), (2) the PFDL module (PFD plus PFD-like proteins) and (3) the PAQosome (Particle for Arrangement of Quaternary structure) which combines the PFDL module to the R2TP complex. The PAQosome is a HSP90 chaperone complex which was previously shown to assist in assembly and maturation of large RNA-binding protein assemblies.

Benchmarking experiments comparing classical AP-MS in a reciprocal purification setup, size exclusion chromatography (SEC) coupled with MS and DIP-MS revealed broader dynamic range, captures the separation behaviour of proteins at higher resolution, and generates more extensive and denser networks and recapitulated a significantly larger portion of the gold standard. In the opinion of this reviewer, these improvements are important.

Comparison of the different PFD-containing complexes not only confirmed prior knowledge, but identified two structural variants of PAQosome subunits: the reported RUVBL1/2 heterohexamers and the RPAP3:PIH1D1 subassembly. The data also suggests the existence of previously unidentified alternative PDRG1-containing PFD complex, for which a role remains to be defined. The data confirms that core of the PAQosome PFDL complex contains the RNA polymerase subunit POLR2E. It has to be mentioned that these conclusions have been obtained using only a few experiments, as compared to previous AP-MS methods. In the opinion of this reviewer, this is a major achievement.

Most interestingly, DIP-MS identified novel clients of both the canonical PFD complex and the PAQosome. For the PAQosome, more than 90 proteins organized in about 20 complexes, including many previously identified clients and not yet identified clients such as the Prohibitin complex or the Anaphase-promoting complex (APC/C). Figure 6C nicely summarizes the complexity of this data.

Overall, this is an outstanding manuscript for two main reasons. First, the DIP-MS method provides major improvements over previous methods used for the determination of protein complex composition and architecture, promising to yield significant new data in many fields of biology in the future. Second, the discovery of novel clients (or regulators) for pivotal chaperoning machines of the cell reveals new biology that will be useful for many investigators studying normal and disease conditions. In addition, the figures are well designed, the text easy to follow, and the reference list extensive.

COMMENTS TO BE ADDRESSED:

1 Although DIP-MS recovers and confirms interactors of the PAQosome and other PFD-containing complexes, the method does not discriminate between clients and adaptors. Can DIP-MS be used to make such a difference, which is somehow important to define mechanisms?

2 A recent report indicates that the PAQosome plays a role in ribosomal subunit assembly (J Proteome Res; this reference should be added here). This link appears to be missing in this study. What is the opinion of the authors on this particular point. What explains the fact that ribosomal subunits are not identified here?

3 Please clarify if PAQosome clients are limited to large RNA-binding protein assemblies. If not, is there a link with RNA metabolism?

Reviewer #2 (Remarks to the Author):

In this manuscript the authors claim to have developed an important new approach for studying protein interactions using affinity purification, blue native page gels, standard sample preparation and mass spectrometry, and a neural network based approach for data analysis. They use this combination of established approaches to study human prefolidin complexes. The culmination of this is Figure 6, for example, where a map of protein complexes and protein interactions is presented in a very typical 'ball and stick' like representation. Overall, this manuscript is derivative of much of the work in the field that has been going on for many years, and does not present a compelling case for an advance. All these methods are used commonly, perhaps with the case of blue native page gels, which is a very old technique. In general, most laboratories are pursuing the separation of complexes using more innovative approaches. The major question that arises from this work is exactly what problem are the authors solving that advances the field. All these methods basically exist and have been used in combination before, and the current state of the art methods are capable of significantly greater insight into the assembly of protein interaction networks than the method described here.

Many groups (Marcotte, Emili, Gingras, Rappsilber, Ideker, Krogan, etc.) have made important contributions to the field of protein interactions over the years, and one cannot cover all these groups and all these papers here. In many of the papers from these groups the methods described in this body of work have been used. The Marcotte lab has published on the use of machine learning (Super.Complex: A supervised machine learning pipeline for molecular complex detection in protein-interaction networks | PLOS ONE). Cofractionation using other methods has been described by the Emili group (Scalable multiplex co-fractionation/mass spectrometry platform for accelerated protein interactome discovery | Nature Communications). In addition, the major areas of innovation have included things like proximity analysis, as seen with the Gingras group (A proximity-dependent biotinylation map of a human cell | Nature) and cross linking mass spectrometry and AI approaches (Protein complexes in cells by AI-assisted structural proteomics - PubMed (nih.gov)). Another recent publication demonstrates the power of cross linking mass spectrometry (Cross-linking mass spectrometry discovers, evaluates, and corroborates structures and protein-protein interactions in the human cell | PNAS). Here, the current manuscript does not use any of these continually emerging techniques, like XL-MS in particular, to advance the field. As such it is very difficult to envision the reason that other researchers would adopt the combination of established methods presented in this

manuscript.

As such, the authors should reconsider this manuscript as an analysis of the human prefoldin interactome, but it would need significant orthogonal validation and biological insights for purposes of publication.

Author Rebuttal to Initial comments

Point-by-point response for the manuscript DIP-MS: A novel ultra-deep interaction proteomics for the deconvolution of protein complexes

Reviewer Comments:

Reviewer #1 (Remarks to the Author):

Protein affinity purification coupled with mass spectrometry (AP-MS) has been used by numerous laboratories to characterize protein complexes. Because a given sample can contain a diversity of complex isoforms (with some varying subunits), the characterization of these complexes usually require reciprocal tagging of several subunits in AP-MS. In this article, Frommelt and colleagues introduce Deep Interactome Profiling by Mass Spectrometry (DIP-MS), a method combining affinity purification with native-fractionation by PAGE and data independent acquisition mass spectrometry (DIA-MS) and a data analysis software package to resolve complex isoforms sharing the same bait protein in a single experiment. According to the authors, DIP-MS provides three major improvements: (1) a novel miniaturized sample preparation procedure in a filter plate format that requires 10 times less material than traditional chromatography-based separation and achieves high reproducibility; (2) a DIA-MS acquisition scheme coupled with a 24 minutes injection to injection gradient chromatography system that allows for measuring of up to 60 samples per day, thus increasing sample throughput compared to prior methods; (3) a deep-learning based framework trained on more than 1.5 million binary interactions from various datasets, which enabled prediction of protein-protein interactions (PPIs), identification of multiple instances of protein complexes, and ultimately the reliable deconvolution of complex profiling data into functional modules. This new method represents the latest evolution of the AP-MS procedure, with unprecedented resolution and sensitivity.

DIP-MS was tested using the prefoldins (PFD), a group of chaperones that have previously been shown to be components of various complexes in human cells: (1) the canonical PFD complex (2 alpha and 4 beta subunits), (2) the PFDL module (PFD plus PFD-like proteins) and (3) the PAQosome (Particle for Arrangement of Quaternary structure) which combines the PFDL module to the R2TP complex. The PAQosome is a HSP90 chaperone complex which was previously shown to assist in assembly and maturation of large RNA-binding protein assemblies.

Benchmarking experiments comparing classical AP-MS in a reciprocal purification setup, size exclusion chromatography (SEC) coupled with MS and DIP-MS revealed broader dynamic range, captures the separation behaviour of proteins at higher resolution, and generates more extensive and denser networks and recapitulated a significantly larger portion of the gold standard. In the opinion of this reviewer, these improvements are important.

Comparison of the different PFD-containing complexes not only confirmed prior knowledge, but identified two structural variants of PAQosome subunits: the reported RUVBL1/2 heterohexamer and the RPAP3-PIH1D1 subassembly. The data also suggests the existence of previously unidentified alternative PDRG1-containing PFD complex, for which a role remains to be defined. The data confirms that core of the PAQosome PFDL complex contains the RNA polymerase subunit POLR2E. It has to be mentioned that these conclusions have been obtained using only a few experiments, as compared to previous AP-MS methods. In the opinion of this reviewer, this is a major achievement.

Most interestingly, DIP-MS identified novel clients of both the canonical PFD complex and the PAQosome. For the PAQosome, more than 90 proteins organized in about 20 complexes, including many previously identified clients and not yet identified clients such as the Prohibitin complex or the Anaphase-promoting complex (APC/C). Figure 6C nicely summarizes the complexity of this data.

Overall, this is a outstanding manuscript for two main reasons. First, the DIP-MS method provides major improvements over previous methods used for the determination of protein complex composition and architecture, promising to yield significant new data in many fields of biology in the future. Second, the discovery of novel clients (or regulators) for pivotal chaperoning machines of the cell reveals new biology that will be useful for many investigators studying normal and disease conditions. In addition, the figures are well designed, the text easy to follow, and the reference list extensive.

COMMENTS TO BE ADDRESSED:

- 1 Although DIP-MS recovers and confirms interactors of the PAQosome and other PFD-containing complexes, the method does not discriminate between clients and adaptors. Can DIP-MS be used to make such a difference, which is somehow important to define mechanisms?*
- 2 A recent report indicates that the PAQosome plays a role in ribosomal subunit assembly (J Proteome Res; this reference should be added here). This link appears to be missing in this study. What is the opinion of the authors on this particular point. What explains the fact that ribosomal subunits are not identified here?*
- 3 Please clarify if PAQosome clients are limited to large RNA-binding protein assemblies. If not, is there a link with RNA metabolism?*

Point-by-point response for the manuscript *DIP-MS: A novel ultra-deep interaction proteomics for the deconvolution of protein complexes*

Reviewer #1

We thank the reviewer for the positive comments.

COMMENTS TO BE ADDRESSED:

1. Although DIP-MS recovers and confirms interactors of the PAQosome and other PFD-containing complexes, the method does not discriminate between clients and adaptors. Can DIP-MS be used to make such a difference, which is somehow important to define mechanisms?

DIP-MS data alone cannot distinguish between substrates/clients and adaptor proteins interacting with PAQosome and other PFD containing complexes. Due to the unbiased nature of the approach, it cannot distinguish between different types of interactions, e.g. alternate complex components to form a complex isoform, regulatory proteins, co-factors or folding substrates or assembly clients. These and other types of interactions can be distinguished by relating the data to prior knowledge about the complexes under study or, of course, by specific follow-up experiments. In the case of CCT/TRiC, CCT/TRiC-PFD supercomplex we identified 14 already known CCT/TRiC-PFD substrates in our DIP-MS data (see Supplementary data S12). We also found the known adaptor PDCL and PDCL3 which acts as CCT/TRiC cochaperone for the folding of GNB1. Besides proteins with annotated function, DIP-MS identified a group of interacting proteins not functionally linked yet to CCT/TRiC/PFD complexes. Some of these unclassified proteins have interesting homologies to known substrates and adaptors but need further validation. To avoid confusion, we do now classify these proteins as “unclassified coeluting proteins” of CCT/TRiC-PFD containing complexes instead of “new client candidates” in our revised version (Supplementary table data S12, manuscript text, Figure 5 and extended data Figure 8) and made the following changes to the manuscript to help clarifying this point:

line 517-519: “CCIs represent potentially high-order assemblies and specialized machines whereas PCIs comprise of known canonical substrates and co-chaperones as well as unclassified coeluting interactors (Supplementary Data S12).”

Lines 524-536: “Besides PFD and CCT/TRiC subunits we identified 38 additional proteins coeluting CCT/TRiC-PFD complexes which also can bind to PFD and CCT/TRiC subunits according to public databases. These include 4 known co-chaperones (PDCD5, PDCL, PDCL3, TXNDC9), 14 known folding substrates and 20 proteins not classified yet. Among these unclassified proteins, we found 4 exclusively coeluting with CCT/TRiC, 6 with PFD and 10 with both, indicating shuttling between these two complexes, which demonstrates the great degree of granularity achievable by DIP-MS (see summary in Supplementary Data S12). Furthermore, a significant proportion of the unclassified proteins contains WD repeats (30%), which is a structural domain known to be folded by CCT/TRiC, and present in 35% of the known substrates we identified in the DIP-MS data. Among the unclassified proteins we found the G-protein GNB2, which has been previously identified as interaction partner of PFD subunits and PDRG1 (Huttlin et al., Cell, 2021; Taipale et al., Cell, 2014) adding orthogonal evidence to our previous discovery of the PFD homolog complex.”

Likewise, the DIP-MS data alone also do not support the differentiation between client or adaptor interaction for the PAQosome complex - this is also not the goal of the method. The PAQosome assists in assembling multi subunit complexes. We can thus expect that both single proteins or entire complexes are potential clients of PAQosome-HSP90 mediated folding (see Figure 6c for an overview, and Supplementary Table S14). To categorize the PAQosome interactions (line 575 – 579) into chaperones and

Point-by-point response for the manuscript DIP-MS: A novel ultra-deep interaction proteomics for the deconvolution of protein complexes

client-complexes we therefore again built on prior knowledge. In the case of the PAQosome, the large number of detected interactors makes it unlikely that they are all part of the same molecular entity. The data therefore point to a family of related client-complex entities as would be expected from the interactions of the core complex with client proteins.

To address the comment of the reviewer to more clearly state that prior knowledge or verification experiments are required to differentiate between different classes of interactors, we added the following sentence to the discussion (line 704-707): "While DIP-MS can detect various classes of interacting proteins, their functional classification into clients, adaptors or chaperones can only be achieved through literature-based information or additional experiments and cannot be expected from the acquired data alone."

To avoid confusion in defining folding substrates, adaptors and unclassified coeluting proteins for CCT/TRIC, we changed Figure 5 and extended data Figure 8. The corresponding Supplementary data 12 and 13 were modified accordingly. A minor change was made in Fig.6 by changing from 'ungrouped PAQosome clients' to "Unclassified coeluting PAQosome interactors". For consistency the new terminology was also updated in Extended Data Figure 9 and in Supplementary data set 14 and 15.

2. A recent report indicates that the PAQosome plays a role in ribosomal subunit assembly (J Proteome Res; this reference should be added here). This link appears to be missing in this study. What is the opinion of the authors on this particular point. What explains the fact that ribosomal subunits are not identified here?

The study of Pinard et al. found an increased interaction with ribosomal proteins in AP-MS with PAQosome subunit RPAP3 phospho-null mutants compared to wild-type RPAP3, suggesting this interaction is a transient and sub-stoichiometric RPAP3 interaction. The authors fail to recover any PAQosomal subunits in the reciprocal AP-MS with RRP1B and PDCD11 identified in the phospho-null mutant AP-MS and the authors therefore suggest in the discussion "However, not being able to pull down PAQosome subunits during AP-MS experiment suggests that the co-chaperone might act as a transient reaction during ribosomal maturation and a cause of AP-MS limitation".

It is important to note that AP-MS based identification of interactions with high abundant ribosomal subunits are particularly challenging, given their unspecific copurification and identification in the majority of AP-MS data. As example of this, the CRAPome database (Mellacheruvu et al., Nature Methods, 2013) reports an approximate frequency of 50% across all control experiments for both 40S and 60S ribosomal subunits.

When evaluating the recovery of RPAP3 binders from Pinard et al., across all our AP-MS validation experiments (which encompass both PAQosome and PFD exclusive subunits as baits), a small portion (7/102) were scored as specific interactors of which 5 were identified as RPAP3 interactors (DHX30, HSPA4, HSPH1, RSL1D1, WDR36). The vast majority of the other RPAP3 binders by Pinard et al., (65/102, ~64%) were found in our GFP negative control and therefore scored as unspecific interactions in our study (reviewer Figure 1).

Point-by-point response for the manuscript *DIP-MS: A novel ultra-deep interaction proteomics for the deconvolution of protein complexes*

Reviewer Fig. 1: Recovery of Pinard et al. RPAP3 interactors across our in-house AP-MS dataset. Y axis shows the number of proteins identified (grey) and scored as interactors (red).

Our combined AP-MS and DIP-MS data supports the PAQosome interaction with one ribosomal protein (RSL1D1), thereby providing evidence that the PAQosome may be involved in ribosomal maturation. To highlight this, we added the following sentence (line 622 – 626): “In a recent study the PAQosome was linked to ribosomal maturation via transient interaction between the unphosphorylated subunit RPAP3 with ribosomal proteins (see Extended Figure 1) [78]. Despite low abundance of this interaction, we confirmed multiple hits from this study including ribosomal protein RSL1D1, thereby providing evidence of a potential role of the PAQosome in ribosomal maturation.”

3. Please clarify if PAQosome clients are limited to large RNA-binding protein assemblies. If not, is there a link with RNA metabolism?

We provided an overview of the identified and proposed functionalities of the PAQosome in the Extended data figure 1, which includes besides the RNA-binding polymerases, snRNP and snoRNP complexes also the pre-assembly of Dynein arms (Fabczak & Osinka, *Int J Mol Sci*, 2019), interaction with the tuberous sclerosis complex (TSC; TSC1, TSC2) (Cloutier et al., *Nature Communications*, 2017; Malinova et al., *J Cell Biol*, 2017) and biogenesis and stabilization of multiple PIKKs complexes including ATM, ATR, DNA-PKcs, mTOR complexes (Horejsi et al., *Molecular Cell*, 2010; Kim et al., *Molecular Cell*, 2013). We mention part of these complex-complex interactions also in the result section (line 573 – 575) but mainly link to the Extended data figure 1 for a summary of the functionality of the PFD and PFDL-containing assemblies.

Point-by-point response for the manuscript DIP-MS: A novel ultra-deep interaction proteomics for the deconvolution of protein complexes

Revision Fig. 2: Reactome term enrichment for identified PAQosome clients. X axis represents the negative log₁₀ p value adjusted for multiple testing

We now extended the description of the multiple roles of the PAQosome in the introduction, which should clarify that the PAQosome not only interacts with RNA-binding proteins. For this we extend the text from lines 122 to 125 by adding other functionalities of this important cellular machinery. "can assemble in supercomplexes, such as the chaperonin CCT/TRIC-PFD [29] and most prominently the PAQosome (also known as R2TP/PFDL or URI complex), a HSP90 chaperone complex, which has multiple biological chaperone functions including assisting in assembly and maturation of large RNA-binding protein assemblies [30-34], stabilization of multiple PIKKs complexes [32, 33] and interaction with the tuberous sclerosis complex (TSC) [31, 35] (see Table 1 and Extended data Fig. 1).

We grouped multiple scored subunits into complexes (see Fig. 6 and Supplementary data S14) which are involved in RNA metabolism including spliceosome components (PRF19-complex), U2 snRNP, U5 snRNP and Box C/D snoRNP. The Reactome pathways enrichment analysis of the 119 proteins scored to interact with the PAQosome or adaptor peak, revealed as shown in Revision Fig. 2 that this set is enriched for mRNA splicing, metabolism of RNA and transcription which encompass several aspects of RNA life-cycle.

Point-by-point response for the manuscript DIP-MS: A novel ultra-deep interaction proteomics for the deconvolution of protein complexes

Reviewer #2 (Remarks to the Author):

In this manuscript the authors claim to have developed an important new approach for studying protein interactions using affinity purification, blue native page gels, standard sample preparation and mass spectrometry, and a neural network based approach for data analysis. They use this combination of established approaches to study human prefoldin [sic] complexes. The culmination of this is Figure 6, for example, where a map of protein complexes and protein interactions is presented in a very typical 'ball and stick' like representation. Overall, this manuscript is derivative of much of the work in the field that has been going on for many years, and does not present a compelling case for an advance. All these methods are used commonly, perhaps with the case of blue native page gels, which is a very old technique. In general, most laboratories are pursuing the separation of complexes using more innovative approaches. The major question that arises from this work is exactly what problem are the authors solving that advances the field. All these methods basically exist and have been used in combination before, and the current state of the art methods are capable of significantly greater insight into the assembly of protein interaction networks than the method described here.

Many groups (Marcotte, Emili, Gingras, Rappsilber, Ideker, Krogan, etc.) have made important contributions to the field of protein interactions over the years, and one cannot cover all these groups and all these papers here. In many of the papers from these groups the methods described in this body of work have been used. The Marcotte lab has published on the use of machine learning (Super.Complex: A supervised machine learning pipeline for molecular complex detection in protein-interaction networks | PLOS ONE). Cofractionation using other methods has been described by the Emili group (Scalable multiplex co-fractionation/mass spectrometry platform for accelerated protein interactome discovery | Nature Communications). In addition, the major areas of innovation have included things like proximity analysis, as seen with the Gingras group (A proximity-dependent biotinylation map of a human cell | Nature) and cross linking mass spectrometry and AI approaches (Protein complexes in cells by AI-assisted structural proteomics - PubMed (nih.gov)). Another recent publication demonstrates the power of cross linking mass spectrometry (Cross-linking mass spectrometry discovers, evaluates, and corroborates structures and protein-protein interactions in the human cell | PNAS). Here, the current manuscript does not use any of these continually emerging techniques, like XL-MS in particular, to advance the field. As such it is very difficult to envision the reason that other researchers would adopt the combination of established methods presented in this manuscript.

As such, the authors should reconsider this manuscript as an analysis of the human prefoldin interactome, but it would need significant orthogonal validation and biological insights for purposes of publication.

Reviewer #2 (Remarks to the Author):

In this manuscript the authors claim to have developed an important new approach for studying protein interactions using affinity purification, blue native page gels, standard sample preparation and mass spectrometry, and a neural network based approach for data analysis. They use this combination of established approaches to study human prefoldin [sic] complexes.

We agree that the individual experimental techniques that constitute the DIP-MS method are not novel and have been applied before in different contexts. However, this is not the point. The DIP-MS method described, to our knowledge, for the first time, the possibility to resolve the population of complexes and proteins present in an affinity purified sample into specific complexes and to describe these resolved complexes with respect to their composition. In essence the DIP-MS method provides an additional level of resolution for the analysis of protein-protein interactions that moves beyond hairball models cited by the reviewer by accurately documenting the modularity of the proteome, identifying the true functional modules and their composition and by substantially reducing the incidence of false positives. All this is achieved from minimal amount of material and at a throughput and cost that vastly outperforms the systematic application of reciprocal AP-MS measurements, the state-of-the-art approach to increase the confidence in complexes identified by AP-MS. Further, we introduce a new computational tool based on deep learning (DL) to derive protein interactions and dissect protein modules from the DIP-MS dataset. This combination of biochemical experimental techniques, coupled with a data-driven DL-approach to derive PPIs is novel and the results it achieves are unprecedented. Finally, we consider the fact that the DIP-MS method builds on simple tools which are accessible to most laboratories as a strength of our

Point-by-point response for the manuscript DIP-MS: A novel ultra-deep interaction proteomics for the deconvolution of protein complexes

approach because this should facilitate its uptake. Compared to other native-PAGE separations (Muller et al., Mol Cell Proteomics, 2016) no special equipment is required.

The culmination of this is Figure 6, for example, where a map of protein complexes and protein interactions is presented in a very typical 'ball and stick' like representation.

We think the reviewer is misinterpreting the Fig 6c as a simple 'pretty plot' rather than the end-point of our analysis of the PAQosome interactome. From the first panel (Figure 6a) we show the co-elution of complexes, and in 6b) co-elution of significantly scored subunits of the PAQosome and associated clients and in c) we represent the fully deconvoluted DIP-MS data of UXT and PFDN2 into their constitute complexes. This represents a clear advancement over the technical limitations of all the techniques listed by the reviewer (AP-MS, Bio-ID/APEX, XL-MS).

Overall, this manuscript is derivative of much of the work in the field that has been going on for many years, and does not present a compelling case for an advance. All these methods are used commonly, perhaps with the case of blue native page gels, which is a very old technique. In general, most laboratories are pursuing the separation of complexes using more innovative approaches. The major question that arises from this work is exactly what problem are the authors solving that advances the field. All these methods basically exist and have been used in combination before, and the current state of the art methods are capable of significantly greater insight into the assembly of protein interaction networks than the method described here.

This is basically the same argument as above repeated in other words and we strongly argue against this point. The characterization of molecular network is a long-standing goal of systems biology, and several groups are pursuing different ways to tackle the intrinsic complexity of the cell interactome. Our manuscript presents a novel solution to the problem of high-resolution mapping of protein complexes within a subnetwork of interest and as result it can be conceptually situated as a high-throughput and high-sensitivity alternative to large-scale reciprocal AP-MS. We proceeded to illustrate and focus on the benchmark versus AP-MS and SEC-MS rather than XL-MS or other techniques due to this. None of the individual methods mentioned by the reviewer can achieve this and the whole point of the DIP-MS method is to move beyond the state of the art represented by the mentioned methods and to move beyond hairball models to a modular representation of the proteome.

Many groups (Marcotte, Emili, Gingras, Rappsilber, Ideker, Krogan, etc.) have made important contributions to the field of protein interactions over the years, and one cannot cover all these groups and all these papers here. In many of the papers from these groups the methods described in this body of work have been used. The Marcotte lab has published on the use of machine learning (Super.Complex: A supervised machine learning pipeline for molecular complex detection in protein-interaction networks | PLOS ONE).

The Marcotte/Emili lab and also our lab (Fossati et al., Nature Methods, 2021; Heusel et al., Molecular Systems Biology, 2019; Rosenberger et al., Cell Syst, 2020) have been driving the application of machine learning and statistical analysis of co-fractionation MS datasets but all developed tools until now (Prince (Stacey, Skinnider, Scott, & Foster, BMC Bioinformatics, 2017), EPIC (Hu et al., Nature Methods, 2019), PCprophet (Fossati et al., Nature Methods, 2021), CCprofiler (Heusel et al., Molecular Systems Biology, 2019), SECAT (Rosenberger et al., Cell Syst, 2020), Super.Complex (Palukuri & Marcotte, PLoS One, 2021), ComplexFinder (Nolte & Langer, Biochim Biophys Acta Bioenerg, 2021)) have not been able to extract

Point-by-point response for the manuscript DIP-MS: A novel ultra-deep interaction proteomics for the deconvolution of protein complexes

multiple subcomplexes for a network of interest, both by technical limitations intrinsic in the separation technique utilized or the computational framework of the various tools.

Beside showing the first utilization of DL for PPI prediction from co-fractionation MS datasets, we also further improved the general workflow of interaction prediction utilized by all ML/DL powered co-fractionation MS datasets, where traditionally the predictions are used as-is without further processing for network building and resulting complex inference (EPIC/ Prince/ ComplexFinder/ SuperComplex).

Specifically, to account for spurious co-purification we modified a well-established AP-MS specific score (WD score, (Sowa, Bennett, Gygi, & Harper, Cell, 2009)) to eliminate the need for control samples as well as utilized target-decoy competition for further refining predictions.

Cofractionation using other methods has been described by the Emili group (Scalable multiplex co-fractionation/mass spectrometry platform for accelerated protein interactome discovery | Nature Communications).

Global co-fractionation MS is a powerful technique which our lab together with many others have been working on for several years (Bludau et al., Nat Protoc, 2020; Bludau et al., J Proteome Res, 2023; Havugimana et al., Cell, 2012; Heusel et al., Molecular Systems Biology, 2019; Heusel et al., Cell Syst, 2020; Pourhaghighi, O'Meara, Cowen, & Emili, Methods Mol Biol, 2019; Samant et al., Molecular & Cellular Proteomics, 2023) and that offers an unbiased view on an organism interactome. However, there is an intrinsic tradeoff between the number of interactions that can be possibly to detected and the sensitivity to recover most complexes associated with a protein of interest and understand its repartition across them. To show this shortcoming of total cell lysate SEC-profiling, we utilized all available co-fractionation studies performed in human from a recent publication (Skinnider & Foster, Nature Methods, 2021) for a well-characterized system (PFDN) and a closely related one (PFDL).

When evaluating the recovery of PFDL across these 32 independent co-fractionation studies, only 2 studies identified all components of this complex, while the majority of the studies (19/32) only recovered half or less (Supplementary Fig. 2b), suggesting that these proteins are too low in abundance to be identified with whole-cell co-fractionation MS approaches regardless of the column chemistry, fractionation scheme, proteomics sample preparation, mass spectrometer, MS acquisition scheme and data processing employed.

We then performed a more in-depth benchmark against a comparable Hek293 full proteome profiling with SWATH-DIA, which showed that total cell lysate co-fractionation did not lead to separated profiles for the PFDL containing complexes (as shown in Supplementary Figure 2a) even when those subunits are identified.

It should also be noted that multiple approaches used crude cellular organellar enrichment in order to boost sensitivity for co-elution profiling approaches, thereby allowing them to access lower-abundance proteomes (Schulte et al., Nature, 2023; Van Strien et al., Bioinformatics, 2019). This makes it very clear that global protein co-fractionation profiling does not have the sensitivity to map all protein complexes or instances of protein complexes for a protein of interest as we argue also on in the result section (line 232 to 258) and in the discussion (line 663 – 671). Our DIP-MS technique aims to fill this crucial gap by enabling to concurrent determination of protein complexes associated with a protein of interest at a resolution not achievable by global co-fraction MS as we further demonstrate in Supplementary Figure 2b.

Point-by-point response for the manuscript DIP-MS: A novel ultra-deep interaction proteomics for the deconvolution of protein complexes

Of note, the study mentioned by reviewer #2 supports our claim that lower abundant complexes are missed (as we mentioned in the manuscript at line 232 to 258, Figure 2c, d and Supplementary Fig. 2a, b). In the following figure, we evaluated the complex completeness for PAQosome, PFD and PFDL complexes (defined as % of subunits) both at the MS identification stage (Fig 4a) and the complex-level analysis stage (Fig 4b, c) for the study cited by the reviewer and our DIP-MS dataset. It is clear that the study mentioned does not identify all subunits for these three complexes (Fig 4a), nor does it provide an effective way of deconvolute these proteins into PFD, PFDL and PAQosome.

Specifically, the study recovers 13 interactions between all PFD, PFDL and PAQosome proteins (Fig 4b) while with DIP-MS we recover 76, which results in a denser network highlighting the clear separation between PFD and PFDL complexes (Fig 4c). Moreover, proteins which were identified like ASDURF and which have been reported by us and others to be core PFDL components (Cloutier et al., Journal of Proteome Research, 2020) were not scored as interactors. This is in agreement with the same trend we previously observed for whole-cell co-fractionation MS, where the low-abundance leads to noisy profiles and does not allow to assign proteins into complexes, which ultimately leads to poor network deconvolution for proteins involved in multiple complexes.

Reviewer Fig3. A. Recovery of PAQosome, PFDL and PFDN subunits from Havugimana et al (2022) or our study, expressed as % of subunit identification. B-C Extracted subnetwork for all PAQosome, PFDL and PFDN proteins from Havugimana et al 2022 (B) or our DIP-MS dataset (C)

In addition, the major areas of innovation have included things like proximity analysis, as seen with the Gingras group (A proximity-dependent biotinylation map of a human cell | Nature)

BioID and Apex are powerful approaches to derive a map of the proteins surrounding the tagged protein of interest, but they come with limitations, which was highlighted by multiple laboratories including the Gingras lab cited by the reviewer. Any interaction found in BioID, TurboID or APEX are based on proximity, and thereby does not reveal any direct interactions or complexes (Cho et al., Nat Protoc, 2020; Go et al., Nature, 2021; Nahle et al., STAR Protoc, 2022) in contrast to Y2H, to AP-MS, co-IP-MS, or DIP-MS. While we agree that the mentioned study of the human cell map (Go et al., Nature, 2021) is an important contribution to the field of proximity interactome, it clearly cannot be scaled to multiple cell lines, conditions or perturbations without a substantial amount of resources needed and extensive computational analysis and the true functional complex modules are not identified.

and cross linking mass spectrometry and AI approaches (Protein complexes in cells by AI-assisted structural proteomics - PubMed (nih.gov)).

Cross-linking MS coupled with AI is an important structural technique which we find to be complementary to our approach given it has several limitations that we feel our DIP-MS workflow can accommodate.

Point-by-point response for the manuscript DIP-MS: A novel ultra-deep interaction proteomics for the deconvolution of protein complexes

- Unlike the identification and quantification of proteins, which can be effectively performed from target peptides across the full protein sequence allowing for the selection of sequences that are well-detected by MS, the identification of cross-linked species, especially inter-linked species, can be hugely variable. This is in part due to the variability in the cross-link reaction itself, with several potential outcomes contributing to the dilution of signal across non-reacted, mono-linked (dead-end), intra-linked (loop-linked), and various inter-linked species for the same peptide. In addition, the detectability of cross-linked species can be variable. Whereas only a percent of the full protein sequence is needed to identify and quantify a protein, the cross-linked species can be difficult to be reliably detected across samples. Several proteome-wide publications have now demonstrated a low reproducibility in cross-link peptide identification across biological replicates. In this way, while XL-MS is incredibly useful for many discovery approaches, it fills a different niche than the DIP-MS/co-fractionation MS method we present here. The substantial loss of sensitivity in XL-MS techniques compared to the use of protein profiles as we use in DIP-MS/co-fractionation MS, can also explain the lower interaction recovery compared to other interactomics techniques.
- The exponential increase in search space from an XL-MS search (derived from searching every K-K link as well as hydrolysed XL from both ends) results in an extremely high false positive rate for identification of cross-linked peptides, which is usually mitigated by using of stringent thresholds, hence further lowering XL identification numbers by a large factor. This computational constrains is exemplified by the observation that the majority of proteome-wide XL studies are performed in organisms with small proteomes (as *B.subtilis* in the paper (O'Reilly et al., Molecular Systems Biology, 2023) cited by the reviewer) rather than human. While cleavable cross-linkers have allowed for full proteome searches, and have provided much higher coverage of the human cell proteome (see comment 1 above), these techniques are still not fully capable of covering the full interactome, and as of now, are limited to binary PPI discovery, with multi-protein complexes being inferred, rather than directly identified.
- The incorporation of XL-restraints into alphafold models is undoubtedly a much-needed addition to increase robustness in AF models, but we argue this is a comment out of scope given that we frame our work as alternative technique to reciprocal AP-MS not to XL-MS. We think XL-MS is a technique which might be incorporated into the DIP-MS workflow to further increase the resolution achievable.

Another recent publication demonstrates the power of cross linking mass spectrometry (Cross-linking mass spectrometry discovers, evaluates, and corroborates structures and protein-protein interactions in the human cell | PNAS).

We don't doubt that XL-MS in the future will take the center stage in combination with other interaction proteomics techniques for prediction of interactions, but many computational challenges need to be addressed to be able to utilize the XL-MS data to its full potential.

Point-by-point response for the manuscript DIP-MS: A novel ultra-deep interaction proteomics for the deconvolution of protein complexes

As example, in the paper cited by the reviewer, the authors recover 2110 PPIs, employing subcellular fractionation and peptide fractionations (SEC, reversed phase), resulting in one of the largest XL-MS dataset to date (Bartolec et al., Proc Natl Acad Sci U S A, 2023).

On the other hand, the largest human AP-MS study to date (Huttlin et al., Cell, 2021) encompass more than 100,000 PPIs and the largest human co-fractionation MS dataset has more than 45,000 interactions (Skinnider & Foster, Nature Methods, 2021). The largest BioID study revealed 35,902 proximity interactions for 192 bait proteins (Go et al., Nature, 2021). While XL-MS could potentially lead to greater confidence interactome compared to a global PPI study, we think of these three techniques as complementary rather than contrasting given the vastly different results and numbers achievable with current instrumentation and computational tools for both approaches.

Here, the current manuscript does not use any of these continually emerging techniques, like XL-MS in particular, to advance the field.

The biological significance of analyzing protein complexes isolated from cells at different states is clearly apparent from the many techniques that have been developed and that are cited by the reviewer. However, they fail to mention or appreciate that none of these methods can determine the actual protein complexes that are present in a sample.

In contrast to the cited methods which determine either molecular proximity or provide an average composition of multiple, concurrently present complexes, DIP-MS further resolves the complexes present in an AP-MS sample and infers the composition of these actually present modules. We highlighted the applicability by performing “in-silico” validation of the identified complexes employing AlphaFold multimer. The reviewer argument could be applied to any of the other above approaches cited as they all focus on interaction networks and not global structural elucidation of protein complexes. This could potentially be achieved by combining DIP-MS with our previously introduced AP-XL workflow (Herzog et al, Science 2012).

As such it is very difficult to envision the reason that other researchers would adopt the combination of established methods presented in this manuscript.

The computational workflow presented here has been utilized in a recent preprint. As mentioned above the accessibility of the equipment ensures that the method can be adapted easy in other laboratories to study complex instances. Also, the recent spread of high-throughput HPLC-systems (such as EVOSEP) along with new high-speed instrumentation will allow further laboratories to employ our proposed acquisition scheme within the DIP-MS.

As such, the authors should reconsider this manuscript as an analysis of the human prefoldin interactome, but it would need significant orthogonal validation and biological insights for purposes of publication.

We argue that the advances introduced in our DIP-MS manuscript at all levels, from developing a low-input purification pipeline to the deep learning and computational framework built are beyond incremental and a much-needed addition to the toolkit of the proteome community interested in protein-interaction and protein complex analysis. It is as of now the most effective approach to identify the true complex modules that are present in an AP sample.

Decision Letter, first revision:

Dear Fabian,

Thank you for your letter detailing how you would respond to the reviewer concerns regarding your Article, "DIP-MS: A novel ultra-deep interaction proteomics for the deconvolution of protein complexes". We have decided to invite you to revise your manuscript as you have outlined, before we reach a final decision on publication.

[Redacted] This URL links to your confidential home page and associated information about manuscripts you may have submitted, or that you are reviewing for us. If you wish to forward this email to co-authors, please delete the link to your homepage.

We hope to receive your revised paper within 6 weeks. If you cannot send it within this time, please let us know. In this event, we will still be happy to reconsider your paper at a later date so long as nothing similar has been accepted for publication at Nature Methods or published elsewhere.

OPEN SCIENCE REQUIREMENTS

REPORTING SUMMARY AND EDITORIAL POLICY CHECKLISTS

IMAGE INTEGRITY

DATA AVAILABILITY

Please include a "Data availability" subsection in the Online Methods. This section should inform readers about the availability of the data used to support the conclusions of your study, including accession codes to public repositories, references to source data that may be published alongside the paper, unique identifiers such as URLs to data repository entries, or data set DOIs, and any other statement about data availability. At a minimum, you should include the following statement: "The data that support the findings of this study are available from the corresponding author upon request", describing which data is available upon request and mentioning any restrictions on availability. If DOIs are provided, please include these in the Reference list (authors, title, publisher (repository name), identifier, year). For more guidance on how to write this section please see:

<http://www.nature.com/authors/policies/data/data-availability-statements-data-citations.pdf>

CODE AVAILABILITY

Please include a "Code Availability" subsection in the Online Methods which details how your custom code is made available. Only in rare cases (where code is not central to the main conclusions of the paper) is the statement "available upon request" allowed (and reasons should be specified).

For more information on our code sharing policy and requirements, please see:
<https://www.nature.com/nature-research/editorial-policies/reporting-standards#availability-of-computer-code>

MATERIALS AVAILABILITY

SUPPLEMENTARY PROTOCOL

To help facilitate reproducibility and uptake of your method, we ask you to prepare a step-by-step Supplementary Protocol for the method described in this paper. We [encourage authors to share their step-by-step experimental protocols](https://www.nature.com/nature-research/editorial-policies/reporting-standards#protocols) on a protocol sharing platform of their choice and report the protocol DOI in the reference list. Nature Portfolio's Protocol Exchange is a free-to-use and open resource for protocols; protocols deposited in Protocol Exchange are citable and can be linked from the published article. More details can found at www.nature.com/protocolexchange/about.

ORCID

Sincerely,
Arunima

Arunima Singh, Ph.D.
Senior Editor
Nature Methods

Reviewers' Comments:

Reviewer #1:

Remarks to the Author:

The response of the authors to my previous comments is generally to my entire satisfaction, in fact beyond expectations. New interesting materials was added to the figures and the text. Let me precise that additions to the revised manuscript are highly relevant and improve significantly the overall quality of the paper. The paper is really outstanding and bring a lot to the understanding of PAQosome function and mechanisms. This point is in my opinion capital because the PAQosome is not an ordinary complex as it regulates assembly and maturation of numerous complexes involved in both normal cell function and disease. More importantly, the method described and used in this paper is in my opinion the only approach existing so far to characterize the PAQosome system with this level of details, defining the organization and function of the distinct modules. This revised manuscript now warrants publication in Nature Methods.

Reviewer #2:

Remarks to the Author:

The revised version of this manuscript does not have any major changes in response to my initial review. Rather, the authors have largely presented a strongly worded counter argument to the comments regarding the first version of this manuscript. I have now reread the manuscript and authors response several times in order to make a concerted effort to understand the authors argument that their work is a significant advance, unfortunately I still do not see it as such. The result of this is that we have a strong difference of opinion, which is largely subjective on both ends and not the ideal approach for determining the appropriateness for a publication, especially in a journal like Nature Methods.

To try to remedy this and perhaps in a further revised manuscript in a productive and scientifically based approach I recommend one of two options. One option is Nature Methods seeks a 3rd adjudicative reviewer, which is often appropriate in situations like this, but is not ideal either. The second option would be to revise the manuscript and provide additional experimental evidence validating the potential power of their approach. As it stands, the manuscript contains proteomics and computational data, but does not validate the proposed biological insights of the study. It also does not provide extensive benchmarking against alternative computational approaches.

Therefore, I recommend two areas of additional research. First, the authors should provide a detailed

alternative computational analysis of their data using alternative algorithms and demonstrate their approach captures information that other approaches do not. However, this then necessitates the biological validation of such a claim. Using cell imaging approaches demonstrating interactions in cells with approaches like a proximity ligation assay or AP-FRET to demonstrate cellular interactions uniquely captured by DIP-MS would be a good approach. Also, potentially very compelling would be to perturb the system that they are studying and use DIP-MS to demonstrate the ability to accurately analyze a perturbed protein interaction network.

Reviewer #3:

Remarks to the Author:

Because proteins tend to assemble into complexes to carry out their biological functions, defining protein interactions and complexes can provide important biological insights. Toward this end, a variety of experimental approaches – many enabled by mass spectrometry – have been developed to profile protein-protein interactions. In this paper, Frommelt et al. present their DIP-MS method, which blends elements of existing MS methods (AP-MS and native complex separations) with modern instrumentation and deep learning to efficiently define protein complexes. They demonstrate their method by characterizing human prefoldin complexes.

While the opinions of the first two reviewers vary, I think the DIP-MS method presented by the authors does represent a significant technical and conceptual advance for the study of protein-protein interactions. Through their thorough characterization of the prefoldin complexes, the authors do a good job showing the potential of their method. And overall, I think the manuscript and supplement explain their method both clearly and completely. I'm also satisfied with the authors' responses to the comments from Reviewers 1 and 2.

The chief concern raised by Reviewer #2 is that DIP-MS lacks novelty, as it is built from elements (affinity purification, complex separations, machine learning) that are well established. I believe the novelty of this approach comes from the unique way the authors have put these pieces together in the context of modern instrumentation to create a platform that can efficiently delineate protein complexes in a single experiment, in a way that isn't really duplicated by other techniques. Though both affinity purification and protein co-fractionation have long histories, they've typically been employed separately – either you do affinity purification and analyze the resulting sample as a single fraction, or you use size exclusion/native gel fractionation to separate complexes from cell lysates or relatively complex mixtures. Combining these biochemical approaches wasn't practical, since affinity purification typically recovers small amounts of protein, while protein complex fractionation has generally required larger sample input and considerable instrument time to attain depth and resolution. Here the authors leverage the speed and sensitivity of modern proteomics technology to combine affinity purification with protein co-fractionation in an efficient integrated platform. Combining the specificity of affinity purification with the complex-level resolution of native gels in this way can reveal in a single experiment the diversity of complexes in which a given protein is found. In my view, this makes DIP-MS an important complement to traditional affinity purification and co-fractionation approaches, as well as other methods the reviewer mentions, such as proximity labeling and XL-MS, which are themselves powerful tools, but don't afford the same direct insights into complex organization that DIP-MS provides.

In their comments, Reviewer #2 also points out that others have previously used machine learning

approaches to identify complexes from protein interaction data, mentioning specifically Marcotte's Super.Complex paper (PLOS). It's important to note that though both are concerned with identifying protein complexes, they are solving different computational problems. While Super.Complex seeks communities within a predefined network of protein-protein interactions based on network connectivity, most of the machine learning described in the current paper is concerned with distinguishing real interactions from background (i.e. non-specific coeluting proteins) and inferring complexes from these proteins' co-fractionation profiles. Though others, including the authors of this work, have previously published methods for extracting interactions and complexes from co-fractionation (SEC) data, their computational approach described here is distinct because it has been designed to account for the unique qualities of the data produced by their hybrid method.

Author Rebuttal, first revision:

Reviewer Comments:

Reviewer #1: Remarks to the Author:

The response of the authors to my previous comments is generally to my entire satisfaction, in fact beyond expectations. New interesting materials was added to the figures and the text. Let me precise that additions to the revised manuscript are highly relevant and improve significantly the overall quality of the paper. The paper is really outstanding and bring a lot to the understanding of PAQosome function and mechanisms. This point is in my opinion capital because the PAQosome is not an ordinary complex as it regulates assembly and maturation of numerous complexes involved in both normal cell function and disease. More importantly, the method described and used in this paper is in my opinion the only approach existing so far to characterize the PAQosome system with this level of details, defining the organization and function of the distinct modules. This revised manuscript now warrants publication in Nature Methods.

Reviewer #2: Remarks to the Author:

The revised version of this manuscript does not have any major changes in response to my initial review. Rather, the authors have largely presented a strongly worded counter argument to the comments regarding the first version of this manuscript. I have now reread the manuscript and authors response several times in order to make a concerted effort to understand the authors argument that their work is a significant advance, unfortunately I still do not see it as such. The result of this is that we have a strong difference of opinion, which is largely subjective on both ends and not the ideal approach for determining the appropriateness for a publication, especially in a journal like Nature Methods.

To try to remedy this and perhaps in a further revised manuscript in a productive and scientifically based approach I recommend one of two options. One option is Nature Methods seeks a 3rd adjudicative reviewer, which is often appropriate in situations like this, but is not ideal either. The second option would be to revise the manuscript and provide additional experimental evidence validating the potential power of their approach. As it stands, the manuscript contains proteomics and computational data, but does not validate the proposed biological insights of the study. It also does not provide extensive benchmarking against alternative computational approaches.

Therefore, I recommend two areas of additional research. First, the authors should provide a detailed alternative computational analysis of their data using alternative algorithms and demonstrate their approach captures information that other approaches do not. However, this then necessitates the biological validation of such a claim. Using cell imaging approaches demonstrating interactions in cells with approaches like a proximity ligation assay or AP-FRET to demonstrate cellular interactions uniquely captured by DIP-MS would be a good approach. Also, potentially very compelling would be to perturb the system that they are studying and use DIP-MS to demonstrate the ability to accurately analyze a perturbed protein interaction network.

Reviewer #3: Remarks to the Author:

Because proteins tend to assemble into complexes to carry out their biological functions, defining protein interactions and complexes can provide important biological insights. Toward this end, a variety of experimental approaches – many enabled by mass spectrometry – have been developed to profile protein-protein interactions. In this paper, Frommelt et al. present their DIP-MS method, which blends elements of existing MS methods (AP-MS and native complex separations) with modern instrumentation and deep learning to efficiently define protein complexes. They demonstrate their method by characterizing human prefoldin complexes.

While the opinions of the first two reviewers vary, I think the DIP-MS method presented by the authors does represent a significant technical and conceptual advance for the study of protein-protein interactions. Through their thorough characterization of the prefoldin complexes, the authors do a good job showing the potential of their method. And overall, I think the manuscript and supplement explain their method both clearly and completely. I'm also satisfied with the authors' responses to the comments from Reviewers 1 and 2.

The chief concern raised by Reviewer #2 is that DIP-MS lacks novelty, as it is built from elements (affinity purification, complex separations, machine learning) that are well established. I believe the novelty of this approach comes from the unique way the authors have put these pieces together in the context of modern instrumentation to create a platform that can efficiently delineate protein complexes in a single experiment, in a way that isn't really duplicated by other techniques. Though both affinity purification and protein co-fractionation have long histories, they've typically been employed separately – either you do affinity purification and analyze the resulting sample as a single fraction, or you use size exclusion/native gel fractionation to separate complexes from cell lysates or relatively complex mixtures. Combining these biochemical approaches wasn't practical, since affinity purification typically recovers small amounts of protein, while protein complex fractionation has generally required larger sample input and considerable instrument time to attain depth and resolution. Here the authors leverage the speed and sensitivity of modern proteomics technology to combine affinity purification with protein co-fractionation in an efficient integrated platform. Combining the specificity of affinity purification with the complex-level resolution of native gels in this way can reveal in a single experiment the diversity of complexes in which a given protein is found. In my view, this makes DIP-MS an important complement to traditional affinity purification and co-fractionation approaches, as well as other methods the reviewer mentions, such as proximity labeling and XL-MS, which are themselves powerful tools, but don't afford the same direct insights into complex organization that DIP-MS provides.

In their comments, Reviewer #2 also points out that others have previously used machine learning approaches to identify complexes from protein interaction data, mentioning specifically Marcotte's Super.Complex paper (PLOS). It's important to note that though both are concerned with identifying protein complexes, they are solving different computational problems. While Super.Complex seeks communities within a predefined network of protein-protein interactions based on network connectivity, most of the machine learning described in the current paper is concerned with distinguishing real interactions from background (i.e. non-specific coeluting proteins) and inferring complexes from these proteins' co-fractionation profiles. Though others, including the authors of this work, have previously published methods for extracting interactions and complexes from co-fractionation (SEC) data, their computational approach described here is distinct because it has been designed to account for the unique qualities of the data produced by their hybrid method.

Reviewer #1:

The response of the authors to my previous comments is generally to my entire satisfaction, in fact beyond expectations. New interesting materials was added to the figures and the text. Let me precise that additions to the revised manuscript are highly relevant and improve significantly the

overall quality of the paper. The paper is really outstanding and bring a lot to the understanding of PAQosome function and mechanisms. This point is in my opinion capital because the PAQosome is not an ordinary complex as it regulates assembly and maturation of numerous complexes involved in both normal cell function and disease. More importantly, the method described and used in this paper is in my opinion the only approach existing so far to characterize the PAQosome system with this level of details, defining the organization and function of the distinct modules. This revised manuscript now warrants publication in Nature Methods.

We thank the Reviewer #1 for their positive remarks regarding the manuscript and the appreciation for the revised manuscript. Particularly we want to thank for the strong endorsement for publication and the appreciation of our biological insights into the Prefoldin and PAQosome complex organization, which were only achieved by employing the novel DIP-MS method.

Reviewer #2:

The revised version of this manuscript does not have any major changes in response to my initial review. Rather, the authors have largely presented a strongly worded counter argument to the comments regarding the first version of this manuscript. I have now reread the manuscript and authors response several times in order to make a concerted effort to understand the authors argument that their work is a significant advance, unfortunately I still do not see it as such. The result of this is that we have a strong difference of opinion, which is largely subjective on both ends and not the ideal approach for determining the appropriateness for a publication, especially in a journal like Nature Methods.

We thank the reviewer for re-considering the presented manuscript. Some of the differences discussed in the first review/revision cycle are partly differences in opinion. E.g. it is a matter of opinion whether the novel combination of methods to generate previously unattainable results is significant or not. It should be noted that reviewers #1 and #3 discuss this point in their answer and, similar to the authors, clearly see a significant improvement over the state of the art.

Other issues raised by reviewer #2 in the previous review and reiterated here are not a matter of opinion but rather factual. It is not a difference in opinion but fact that proximity labeling – a very powerful method- defines spatial proximity but does not resolve protein complexes. Similarly, AP-MS data of a given protein, if properly controlled, identify proteins that interact directly or indirectly with the bait, but in contrast to DIP-MS do not define the organization of precipitated proteins into specific complexes. Even when performed in a time and labor intense reciprocal format, all the methods put forward by reviewer #2 to suggest that the DIP-MS method merely recapitulates results also obtainable with existing methods only allow for indirect computational inference of protein modules from binary interaction network data but do not directly demonstrate complex formation, particularly not from a single experiment.

The chief concern raised by reviewer #2 that DIP-MS lacks novelty, was challenged by the strong statements of both reviewer #3 as well by the reviewer #1 on the current revision. The chief concern is also alleviated by the results obtained from the analysis of the PAQosome which show unprecedented modularity of this complex molecular machine. This is also acknowledged by reviewer #1 in their comment “... the only approach existing so far to characterize the PAQosome system with this level of details, defining the organization and function of the distinct modules”.

To try to remedy this and perhaps in a further revised manuscript in a productive and scientifically based approach I recommend one of two options. One option is Nature Methods seeks a 3rd adjudicative reviewer, which is often appropriate in situations like this, but is not ideal either. The second option would be to revise the manuscript and provide additional experimental evidence validating the potential power of their approach. As it stands, the manuscript contains proteomics and computational data, but does not validate the proposed biological insights of the study. It also does not provide extensive benchmarking against alternative computational approaches.

We agree with the reviewer #2 for the suggestion and are thankful to the editor for the involvement of a third opinion. We are pleased to read that the third reviewer emphatically supports our claims about novelty and significance of the method.

To address the other remaining issues of reviewer #2 we structured our answer in 2 parts. The first is to point out that many of the issues re-raised in the response to the revised paper were in fact extensively addressed in our first revision and comments to reviewers as well as in the submitted manuscript. The extent of additional information provided in the revised paper was also acknowledged by reviewer #1 and reviewer #3. The second part summarizes the results of new data analysis which we conducted for this second revision cycle following the suggestion of the reviewer #2. Further we discuss which part of the results requested by the reviewer #2 we deem beyond scope.

Therefore, I recommend two areas of additional research. First, the authors should provide a detailed alternative computational analysis of their data using alternative algorithms and demonstrate their approach captures information that other approaches do not. However, this then necessitates the biological validation of such a claim. Using cell imaging approaches demonstrating interactions in cells with approaches like a proximity ligation assay or AP-FRET to demonstrate cellular interactions uniquely captured by DIP-MS would be a good approach. Also, potentially very compelling would be to perturb the system that they are studying and use DIP-MS to demonstrate the ability to accurately analyze a perturbed protein interaction network.

Already included/addressed:

In the first as well as in the revised version of our manuscript we already expended considerable efforts to validate the biological findings, employing not only experimental validation by reciprocal AP-MS (which is suggested by reviewer #2 in the previous revision cycle as alternative technique to DIP-MS), but we also

cross-validated our findings by an extensive literature mining. Results from 556 curated publications strongly support findings of the DIP-MS protein complex dataset.

In the following, we briefly summarize the experimental validation work for two particular findings. The work consisted of a combination of experimental and computational validation steps, specifically reciprocal AP-MS, AP-MS data from another large-scale study, in-silico AlphaFold multimer modelling, and enrichment analysis against a second DIP-MS experiment for which CCT/TRiC-PFD was not enriched.

- **Validation of PDRG1 as an alternative component of the PFD complex:**

To validate PDRG1 as a bona fide component of the PFD complex we provide data from reciprocal AP-MS experiments using PFD1, PFD2, VBP1, PFD4 and PFDN5 as baits (see Figure 4C, Extended data Fig. 5B and Supplementary Data S7). In addition, we provide in-silico validation data from AlphaFold multimer modelling of the PFD homolog (PFDh) complex (see Figure 4D-F, Extended data Fig. 5C-F and line 445-461). Finally, we show that the protein-protein interactions derived from the DIP-MS data were confirmed by literature. However, due to the lack of complex resolution of the methods employed in the respective studies, the membership of PDRG1 in an alternative PFD-complex was not conclusively apparent from the public data.

- **Validation of GNB2 as shuttling client of PFDN2 and CCT/TRiC:**

In the PFDN2 DIP-MS data we observed the coelution of GNB2 with the PFD as well as the CCT/TRiC-PFD complexes (Figure 5B,C and summary in Supplementary Data S12). We validated this interaction by using AP-MS of PFD specific subunits (PFDN1, PFDN2 and PFDN5) and by using previously reported interactions with GNB2 (Extended data Fig.7A-B and line 540-543). We additionally mined the BioPlex database for interactions of GNB2. This identified the prefoldin complex as GNB2 interactor, solidifying our claim (see Extended data Fig.7B). We then used data of the UXT DIP-MS experiment to further validate the specificity of the observed enrichment of coeluting proteins in the PFDN2 DIP-MS data (see line 552-566). The core-subunits of PFD and CCT/TRiC-PFD and coeluting proteins were all enriched in PFDN2 DIP-MS when compared to the UXT-DIP-MS data (Extended data Fig. 8). This analysis is conceptually similar to utilizing reciprocal affinity purification, and we used the UXT DIP-MS as a validation for the results obtained by PFDN2 DIP-MS, a technique which is used to a large extent for interaction proteomics studies such as the BioPlex studies (Huttlin et al., Cell, 2021; Huttlin et al., Cell, 2015) or other studies employing CompPASS (Sowa, Bennett, Gygi, & Harper, Cell, 2009) or similar scoring approaches. Accordingly, GNB2 was not quantified at the canonical prefoldin nor the CCT/TRiC co-elution group within the UXT DIP-MS experiment (Supplementary Data S13), thus further solidifying the specificity of the GNB2 interaction with PFD and CCT/TRiC identified by DIP-MS.

In summary, we already provide a substantial amount of experimental and computational validation evidence for some of the newly discovered interactions. The validation data were generated by the most advanced and best suited techniques outside DIP-MS that are presently available.

Additional data analysis and benchmarking of DIP-MS analysis tool *PPI-Prophet*:

To further address reviewer #2 concerns we performed additional computational analyses to benchmark our DIP-MS analysis tool *PPIprophet* versus other tools for analysis of co-fractionation MS-data. To further strengthen the findings of our DIP-MS results for the PFD and PAQosome we compared the DIP-MS generated PPI-network to publicly available *in-vivo* proximity ligation (BioID) databases, a method proposed by reviewer #2 to validate our method. **Both additions are described in more detail below along with the indication where the new results are inserted into the paper.**

I. Additional efforts to validate protein-protein interactions identified by DIP-MS

As suggested by reviewer #2, we performed an additional validation of PPIs found by DIP-MS. To achieve this, we compared the protein interactions for the 16 core-subunits of the PFD, PFDL and the PFDL-containing PAQosome complexes of DIP-MS against the proposed orthogonal interaction proteomic approaches proposed by reviewer #2.

In the manuscript we already used the manually curated list of reported PPIs and complexes of PFD, PFDL and PFDL-containing PAQosome, derived from AP-MS and similar affinity-based technologies such as co-IP-WB (see Supplementary Table S1). As suggested by reviewer #2, we now compared DIP-MS protein interaction data with interactions identified by *in-vivo* proximity ligation techniques and other orthogonal techniques not yet included in the manually curated list of reported PPIs used in the previous version of our manuscript (Supplementary Table S1).

To this end we extracted all proximity PPIs of the core-subunits reported in the humanCell map (Go et al., Nature, 2021), and combined it with the results from a BioID dataset of PIH1D1 and UXT (Cloutier et al., Journal of Proteome Research, 2020). To expand this BioID dataset, we filtered all proximity interactions for with the 16-core subunits from BioGRID (Stark et al., Nucleic Acids Research, 2006) and supplemented it to generate a proximity interaction reference dataset of 424 PPIs (Revision Table 1). We further overlapped our DIP-MS data to a whole cell lysate XL-dataset, comprised of 6,439 unique interactions among 2,484 proteins (Wheat et al., Proc Natl Acad Sci U S A, 2021) filtered to the core-subunits, as also XL-MS was proposed as alternative method to DIP-MS by reviewer #2. Finally, we decided to include a recent intron-tagged endogenous level AP-MS OpenCell map (Cho et al., Science, 2022) dataset. Likewise, the DIP-MS results were filtered for interactions of the core-subunits. To harmonize across all datasets, gene identifiers were curated, duplicated edges between nodes were removed (ignoring directionality), and all node combinations were ordered alphabetically.

We summarized the number of bait proteins and PPI covered within the dataset in Revision Table S1. For most of the datasets the coverage of bait-proteins was very low or none of the PFD, PFDL and PFDL-

containing PAQosome core-subunits were used as bait protein. We then investigated how many PPIs were reported for the core-subunits per dataset. The size of the different datasets varies largely, from 15 PPIs in the XL-MS dataset to 2'395 for DIP-MS experiments.

Next, we checked how many PPIs identified by the different techniques overlap with the manually curated list of reported PPIs and complexes of PFD, PFDL and PFDL-containing PAQosome. The highest recovery was achieved by DIP-MS reaching 14.2% followed by the endogenous AP-MS with 7.9%. As already shown within the manuscript, where we benchmarked against reciprocal AP-MS (see line 263 to 294 and Figure 2F), DIP-MS reached a higher coverage of already reported interactions compared to reciprocal AP-MS.

In summary, this indicates that DIP-MS has a higher sensitivity compared to other interaction technologies. This result is partially biased, as datasets which include a higher number of the core-subunits as bait-protein recover a higher number of PPIs (BioID and endogenous AP-MS).

Revision Table 1: Summary of PPI-datasets used to orthogonally validate DIP-MS results.

Dataset	Count of core subunits used as baits in the dataset	Core-subunits with PPIs in dataset	Count of PPIs	PPIs overlapping with 2'452 curated literature derived PPIs of 16PFD/PFDL/PAQosome core-subunits	Recovery of PPIs per dataset compared to 2'452 PPIs of 16 PFD/PFDL/PAQosome core-subunits [%]
BioID datasets	2	16	424	119	4.85
XL-MS	0	8	15	4	0.163
DIP-MS	2	16	2'395	347	14.2
Endogenous AP-MS	4	16	408	194	7.91

As proposed by reviewer #2, we next compared the PPIs obtained by DIP-MS with the results from in-vivo protein proximity ligation experiments. These results are summarized below and were added as Supplementary Fig. S4 to the manuscript.

To investigate how proximity ligation experiments compare to the presented DIP-MS approach, we assembled a dataset of BioID interactions (see Supplementary Methods). We started by evaluating the overlap of the 424 proximity PPIs with the 2'395 PPIs obtained from DIP-MS experiments of PFDN2 and UXT. In total 78 PPIs (18% of the BioID dataset) overlapped with our DIP-MS derived protein-interaction network (Supplementary Fig. S4A). Of the overlapping PPIs, 66 were already present in the manually curated list of reported PPIs obtained with other techniques, whereas the other 12 PPIs were exclusively

reported by biotin proximity ligation experiments. A large number of PPIs were only found within each dataset, which indicates that these two approaches are orthogonal to each other.

We investigated further the 78 PPIs for their relevance and biological functions. Despite the small overlap of PPIs, 20 protein interactions between PAQosome core-subunits were validated by proximity ligation data. In addition, protein interactions with all 16 core subunits have been identified by both techniques, resulting in a fully connected protein interaction network (Supplementary Fig. S4B). Within the PPI network, well-established PAQosome and canonical Prefoldin interactors are covered with multiple interactions. For the canonical prefoldin, a tubulin subunit was identified to interact with multiple PFD subunits in both datasets. Further, multiple PPIs between the PAQosome and known interactors such as RNA polymerase subunits (10 subunits, 16 PPIs) and PP1-gamma subunits (2 subunits, 16 PPIs) were recovered by DIP-MS and BioID. Of note, also the link between the PAQosome and the dynein-dynactin complex was mapped by both methods with 7 PPIs to 3 dynein-dynactin subunits. The results are summarized in Supplementary Data S18.

These results show that PPIs recovered by DIP-MS are found by orthogonal technologies such as in vivo proximity ligation. The BioID data only overlap partially with DIP-MS data, which is indicative that the different interaction methods are orthogonal to each other as reported when BioID was compared to AP-MS (Lambert JP et al., 2015, Proteomics). Of note within DIP-MS, PPIs are derived from co-elution, and thus interactions should rather be interpreted as complex-complex or complex-protein interactions (also mentioned in the discussion of the manuscript) rather than direct or indirect interactions as obtained by AP-MS and proximity ligation MS.

To this end we updated the following points:

- Updated “Benchmarking of DIP-MS against AP-MS and SEC-MS workflows for interactome analysis” section in the manuscript (Line 294 to 297).
- Updated Supplementary Result section “Orthogonal evidence of DIP-MS derived PPIs by in vivo proximity interaction datasets” (Line 71 to 97).
- Added visualization of the result as Supplementary Fig. S4 and updated corresponding figure caption.
- Updated Supplementary Materials and methods section “Orthogonal evidence of DIP-MS derived PPIs by in vivo proximity interaction datasets” (Line 437 to 452).
- Updated order of Supplementary Figures to fit the outline.
- Added Supplementary Dataset containing the outputs from the different co-fractionation tools, Supplementary Data S18.
- Updated order of Supplementary Figures and Supplementary Data to fit the updated outline.

II. Benchmarking of the *PPIprophet* software

Reviewer #2 proposed that a computational benchmark of *PPIprophet* against other analysis tools should be included in the manuscript. To evaluate the performance of the ML-approach embodied in *PPIprophet*, we performed an additional benchmark against other published co-fractionation analysis tools (*PPIprophet*, PCprophet, EPIC, PrInCE). The results are described below and added to the manuscript as additional supplementary figure (Supplementary Fig. S1).

We re-analyzed all the 3 replicates of the PFDN2 DIP-MS experiment using EPIC, PrInCE and PCprophet, using default parameters (EPIC/PrInCE) and a comparable false-discovery rate (FDR) filter to *PPIprophet* when possible (PCprophet). For EPIC/PrInCE no comparable FDR assessment is possible. It is important to point out that these comparisons are inherently problematic because the respective tools were developed with different objectives in mind as already mentioned by reviewer #3 in their reply. In these comparisons, we focused on PPIs because both, PrInCE and EPIC utilize third-party software for graph partitioning and they further prune the graph using functional data, which might obfuscate raw prediction performance. As PCprophet does predict complexes directly and not protein interactions *per se*, we decided to derive networks from the predicted complexes assuming fully connected complexes. It is important to keep the latter point in mind as it results in a large number of interactions for a single complex as the software needs to predict only 1 positive (the complex) versus all the single interactions, which inflates true positives for specific analysis (i.e for example the STRING recovery presented in Revision Fig. 1C).

We started by evaluating the total number of protein identifications that were identified as interactors as a proxy for the utilization of co-elution data (or in other words, a larger number of proteins utilized suggests a larger coverage of potential interactions in the data). EPIC identified interactions for every protein id in the PFDN2 dataset (1475/1475), while *PPIprophet* and PrInCE utilized approximately half of the data (52 and 49% respectively). Interestingly, most tools shared a significant fraction of interacting protein (228/1475) and the three PPI-centric tools (PrInCE, *PPIprophet* and EPIC) shared more than 400 proteins, suggesting identification of similar interactions across the dataset (Revision Fig. 1A).

In total, EPIC and PrInCE predicted ~20'000 unique PPIs across the PFDN2 DIP-MS dataset while *PPIprophet* resulted in 3873 PPIs in total (using only high-confidence interactions plus CRAPome filtering of 0.1% for both interactors). PCprophet predicted roughly 10'000 interactions, but it is important to note that large complexes like the ribosome or the proteasome, will result in many PPIs, hence at the PPI-level this number is likely inflated by the assumption of fully connected complexes mentioned above (Revision Fig. 1B).

We then categorized the total number of PPIs detected by the respective tools, depending on whether the interactions were present in STRING or whether there was no prior interaction evidence for them and then calculated the ratio of each of these two classes to the total per each tool. In this analysis, a higher percentage of overlapping STRING interactions suggests a lower false-positive rate, irrespective of the total number of interactions identified. The large number of interactions covered by EPIC or PrInCE did however not translate in a greater recovery of reported interactions in STRING with only 29%

(5629/19247) of the total PPIs from PrInCE being previously reported, potentially suggesting a greater FDR for this tool (Fig. 1C). EPIC identified 47% of previously reported PPIs (9496/20143), on par with *PPIprophet* (47%, 1876/3836) as demonstrated in Revision Fig 1C. PCprophet in this analysis outperformed the other tools (92% of the derived interactions are present in STRING, 2829/3061) due to the over-representation of large complexes (i.e. ribosome, proteasome, etc) which results in a large number of true interactions when converting complexes predicted by PCprophet into protein-protein interactions.

We next focused on evaluating the recovery of known interactions for the canonical PFD and the PAQosome complexes. To achieve so, we extracted for every tool all the interactions for every PFD components (PFDN1, PFDN2, PFDN4, PFDN5, PFDN6 and VBP1) and PAQosome components (PFDN2, PFDN6, URI, UXT, POLR2E, WDR92, RUVBL1, RUVBL2, ASDURF, PDRG1 and PIH1D1) and separated them into true positive (TP) and false positive (FP) interactions using our manually curated list of reported PPIs and complexes of PFD, PFDL and PFDL-containing PAQosome (see Supplementary Table S1).

These two classes were then used to assess the positive predictive value or precision (defined as $TP / (TP + FP)$) for the various tools at the complex-level as the goal of DIP-MS is to reliably identify complexes from their constituent interactions.

When performing this analysis, we identified *PPIprophet* as the tool having the greatest positive predictive value for the PFD interactions (93% precision) and the PAQosome (60% precision) as shown in Revision Fig 1D.

We then turned our attention to evaluating the number of PPIs per protein as a measure of network topology. We observed a large average number of interactions for both PrInCE (53) and EPIC (27), while PCprophet and *PPIprophet* showed a much more modest average number of PPIs per protein (11 and 18 respectively) (Revision Fig. 1E) similar to PPI networks resulting from high confidence AP-MS data ($n=10$; Huttlin et al., 2015).

We then asked whether this larger number of PPIs per protein is due to the use of multiple peaks for a single protein and whether, overall PrInCE and EPIC utilize proteins with a larger number of peaks given both the larger number of total PPIs and the larger number of PPIs per protein. Proteins with large number of peaks could be participating in numerous independent complexes while single-peak proteins are not assumed to be part of multiple complexes. We calculated the number of peaks per proteins using the criteria outlined in the manuscript (line 252-257) and plotted the peak distributions across the different tools as percentage of peaks across tools (Revision Fig. 1F). In this analysis, a higher percentage of proteins having more than one peak shows an efficient peak deconvolution and DIP-MS data usage as we previously have shown that DIP-MS data is characterized by a larger number of peaks per protein compared to SEC-MS datasets. This analysis showed that the highest percentage of interactors detected in multiple peaks was found for PrInCE (54%) which on the other hand had lowest match to known interaction (Revision Figure 1C, 1D) followed by *PPIprophet* (46%), PCprophet (43%) and EPIC (40%).

Finally, we evaluated the abundance distribution of protein interactors detected by the different computational tools. The discovery of proteins of lower abundance would suggest an increase in sensitivity of a specific tool versus the others in detecting interactions for noisier profiles and increased robustness towards noise. To evaluate this, we summed all protein intensities for each PFDN2 replicate individually and averaged across all the three replicates to obtain a single value representing the mean protein intensity. We then used these results to calculate quantiles for this quantity and then compared the distribution of ranked intensities for the interactors identified by the different tools.

It can be appreciated from the data that the PrInCE and PCprophet do have a skew towards more abundant interactors (i.e. a right shift in the ranked distribution plot) while *PPIprophet* detected both, high and low abundance interactors resulting in a binomial ranked distribution plot. For EPIC, as it utilizes all proteins identified as interactors, the resulting average covers the entire intensity range hence having a mean of 0.5 quantile (Revision Fig. 1G).

Overall, the results of these benchmarking analyses suggest a superior performance of *PPIprophet* over other tools to analyze DIP-MS datasets in terms of specificity of recovery of interactions of the PFD and PAQosome complexes from fractionated affinity purifications. It needs to be noted that different tools have been developed for different purposes as mentioned by reviewer #3. The DIP-MS data is substantially different from a global co-fractionation MS dataset as it encompasses few proteins with large number of peaks and abundances spanning several orders of magnitude. Hence, tools relying on global metrics such as global correlation, Euclidean distances etc., exemplified by EPIC/PrInCE, will be at a disadvantage by utilizing a single feature to classify highly convoluted protein-coelution profiles. This is exemplified by the lower precision of PFDN/PAQosome specific interactions which are proteins known to be in multiple peaks and at different abundances. On the other hands, the computational complexity from global co-fractionation MS datasets cannot be readily handled by *PPIprophet* due to the need to construct a global matrix ($N \times N$) which poorly scales with increased protein numbers.

Revision Figure 1: Benchmark of co-fractionation MS tools on DIP-MS data. **A.** Upset plot showing the overlap between the nodes of the interaction networks derived from the different MS tools on the PFDN2- DIP MS data. **B.** Barplot illustrating the total

number of PPIs identified by the various tools. **C.** Recovery of known interactions from STRING (red bar) over the total number of interactions identified, expressed as percentile for the different tools employed. **D.** Positive predictive value (i.e. precision, X-axis) for the canonical PFD (purple) and the PAQosome complex (yellow bar). **E.** Boxplot showing the number of PPIs for every protein (Y-axis) across the different tools (X-axis). Each dot represents a protein, box limits shows the interquartile range (IQR) and its whiskers 1.5xIQR. Solid line represents the mean. **F.** Stacked barplot representing the peak distribution across the different tools. Y-axis shows the number of peaks per proteins represented as percentage. Different peak numbers are represented by the different colors. **G.** Ridgeplot for the intensity distribution of the nodes in the network for the different tools across the three DIP-MS PFDN2 replicates. Solid line represents the mean.

To this end we updated the following points:

- Updated “*PPIprophet*: a deep-learning powered framework for PPI prediction and complex inference” section in the manuscript (Line 181 to 184).
- Updated supplementary result section “Benchmark of co-fractionation-MS tools on DIP-MS data” (Supplementary information, Line 7 to 69).
- Added visualization of the result as Supplementary Fig. S1 and updated corresponding figure caption.
- Updated supplementary materials and methods section “Benchmarking of PPI-prophet against other co-fractionation tools” (Line 417 to 435).
- Added supplementary dataset containing the outputs from the different co-fractionation tools, Supplementary data S17.
- Updated order of supplementary figures and supplementary data to fit the updated outline.

Experimental validation of PPIs by AP-FRET

In contrast to other PPI detection methods such as AP-MS and BioID which have been used in thousands of papers, AP-FRET (acceptor photobleaching fluorescence resonance energy transfer) so far has been used in only three publications to probe RNA-protein and protein-protein interactions (Rehman s. et al., 2014; Kthawala MH et al., 2015; Weems JC et al. 2019). The method has not been used in the last four years in the literature and it therefore remains to be seen how reliable the technique is compared to AP-MS or BioID techniques. Furthermore, because FRET depends on the challenging engineering and empirical optimization of suitably positioned fluorescence emission and acceptor domains, AP-FRET is a very involved and costly technique, features that would limit reasonable validation efforts to one or at most very few selected protein interactions. Importantly, the technical similarity to proximity dependent protein labelling results in binary interactions, and thus does not reveal the presence of defined subcomplexes. We therefore maintain that AP-FRET as proposed by reviewer #2 is neither a suitable nor feasible method to validate DIP-MS results.

Perturbation experiments

Our lab already demonstrated that the related size exclusion chromatography based protein profiling approach allowed deep insights into the dynamic organization of the HeLa proteome during the cell cycle

(Heusel et al., Cell Syst, 2020). Additional work in regard to perturbation co-fractionation-MS experiment has been published by our lab and others (Fossati et al, Nat. Com 2023, Bludau et al., J Proteome Res, 2023; Scott et al., Molecular Systems Biology, 2017). We therefore strongly believe that profiling cell state specific changes in the composition and abundance of protein complexes will also be one of the main strengths of the DIP-MS method. It will be therefore well suited for future studies to gain new insights into protein-complex changes underlying the control of cellular processes. Whereas such results are obviously of high biological significance, we think that inclusion of additional perturbation experiments in our current manuscript would distract from the technological focus and rather be suited for a biological study with a clear focus towards understanding a specific molecular process.

Reviewer #3:

Because proteins tend to assemble into complexes to carry out their biological functions, defining protein interactions and complexes can provide important biological insights. Toward this end, a variety of experimental approaches – many enabled by mass spectrometry – have been developed to profile protein-protein interactions. In this paper, Frommelt et al. present their DIP-MS method, which blends elements of existing MS methods (AP-MS and native complex separations) with modern instrumentation and deep learning to efficiently define protein complexes. They demonstrate their method by characterizing human prefoldin complexes.

While the opinions of the first two reviewers vary, I think the DIP-MS method presented by the authors does represent a significant technical and conceptual advance for the study of protein-protein interactions. Through their thorough characterization of the prefoldin complexes, the authors do a good job showing the potential of their method. And overall, I think the manuscript and supplement explain their method both clearly and completely. I'm also satisfied with the authors' responses to the comments from Reviewers 1 and 2.

The chief concern raised by Reviewer #2 is that DIP-MS lacks novelty, as it is built from elements (affinity purification, complex separations, machine learning) that are well established. I believe the novelty of this approach comes from the unique way the authors have put these pieces together in the context of modern instrumentation to create a platform that can efficiently delineate protein complexes in a single experiment, in a way that isn't really duplicated by other techniques. Though both affinity purification and protein co-fractionation have long histories, they've typically been employed separately – either you do affinity purification and analyze the resulting sample as a single fraction, or you use size exclusion/native gel fractionation to separate complexes from cell lysates or relatively complex mixtures. Combining these biochemical approaches wasn't practical, since affinity purification typically recovers small amounts of protein, while protein complex fractionation has generally required larger sample input and considerable instrument time to attain depth and resolution. Here the authors leverage the speed and sensitivity of modern proteomics

technology to combine affinity purification with protein co-fractionation in an efficient integrated platform. Combining the specificity of affinity purification with the complex-level resolution of native gels in this way can reveal in a single experiment the diversity of complexes in which a given protein is found. In my view, this makes DIP-MS an important complement to traditional affinity purification and co-fractionation approaches, as well as other methods the reviewer mentions, such as proximity labeling and XL-MS, which are themselves powerful tools, but don't afford the same direct insights into complex organization that DIP-MS provides.

In their comments, Reviewer #2 also points out that others have previously used machine learning approaches to identify complexes from protein interaction data, mentioning specifically Marcotte's Super.Complex paper (PLOS). It's important to note that though both are concerned with identifying protein complexes, they are solving different computational problems. While Super.Complex seeks communities within a predefined network of protein-protein interactions based on network connectivity, most of the machine learning described in the current paper is concerned with distinguishing real interactions from background (i.e. non-specific coeluting proteins) and inferring complexes from these proteins' co-fractionation profiles. Though others, including the authors of this work, have previously published methods for extracting interactions and complexes from co-fractionation (SEC) data, their computational approach described here is distinct because it has been designed to account for the unique qualities of the data produced by their hybrid method.

We want to thank Reviewer #3 for their kind remarks and positive feedback.

Decision Letter, second revision:

Dear Fabian,

Thank you for submitting your revised manuscript "DIP-MS: A novel ultra-deep interaction proteomics for the deconvolution of protein complexes" (NMETH-A52080C). It has now been seen by the original referees and their comments are below. The reviewers find that the paper has improved in revision, and therefore we'll be happy in principle to publish it in Nature Methods, pending minor revisions to satisfy the referees' final requests and to comply with our editorial and formatting guidelines.

TRANSPARENT PEER REVIEW

Nature Methods offers a transparent peer review option for new original research manuscripts submitted from 17th February 2021. We encourage increased transparency in peer review by publishing the reviewer comments, author rebuttal letters and editorial decision letters if the authors agree. Such peer review material is made available as a supplementary peer review file. **Please state in the cover letter 'I wish to participate in transparent peer review' if you want to opt in, or 'I do not wish to participate in transparent peer review' if you don't.** Failure to state your

preference will result in delays in accepting your manuscript for publication.

ORCID

Sincerely,
Arunima

Arunima Singh, Ph.D.
Senior Editor
Nature Methods

Reviewer #2 (Remarks to the Author):

I am satisfied with the authors response to my comments and I support publication of this manuscript in Nature Methods.

Reviewer #3 (Remarks to the Author):

After taking a careful look at the latest version of the manuscript as well as their response to the reviewers, I would like to commend the authors for their thoughtful, thorough responses to the points raised by all reviewers. Though I thought the paper was in good shape to begin with, I do agree that the latest additions and edits further strengthen the paper. In their response, the authors have gone to remarkable lengths to address reviewer concerns, particularly by providing experimental validation and extensive computational and literature-based analyses to support their results and by providing additional comparisons with similar software tools for benchmarking. These additional analyses are well done and considerably strengthen their work. While I agree with Reviewer #2 that additional perturbation experiments would be interesting, I agree with the authors that they are beyond the

scope of this paper. Overall, I believe this paper is suitable for publication in Nature Methods in its current form.

Final Decision Letter:

Dear Fabian,

I am pleased to inform you that your Article, "DIP-MS: Ultra-deep interaction proteomics for the deconvolution of protein complexes", has now been accepted for publication in Nature Methods. The received and accepted dates will be March 22, 2023 and February 14, 2024. This note is intended to let you know what to expect from us over the next month or so, and to let you know where to address any further questions.

Over the next few weeks, your paper will be copyedited to ensure that it conforms to Nature Methods style. Once your paper is typeset, you will receive an email with a link to choose the appropriate publishing options for your paper and our Author Services team will be in touch regarding any additional information that may be required. It is extremely important that you let us know now whether you will be difficult to contact over the next month. If this is the case, we ask that you send us the contact information (email, phone and fax) of someone who will be able to check the proofs and deal with any last-minute problems.

Please note that *Nature Methods* is a Transformative Journal (TJ). Authors may publish their research with us through the traditional subscription access route or make their paper immediately open access through payment of an article-processing charge (APC). Authors will not be required to make a final decision about access to their article until it has been accepted. Find out more about Transformative Journals

You may wish to make your media relations office aware of your accepted publication, in case they

consider it appropriate to organize some internal or external publicity. Once your paper has been scheduled you will receive an email confirming the publication details. This is normally 3-4 working days in advance of publication. If you need additional notice of the date and time of publication, please let the production team know when you receive the proof of your article to ensure there is sufficient time to coordinate. Further information on our embargo policies can be found here: <https://www.nature.com/authors/policies/embargo.html>

If you are active on Twitter/X, please e-mail me your and your coauthors' handles so that we may tag you when the paper is published.

Best regards,
Arunima

Arunima Singh, Ph.D.
Senior Editor
Nature Methods